# Learning multi-modal generative models with permutation-invariant encoders and tighter variational bounds

## Abstract

Devising deep latent variable models for multi-modal data has been a long-standing theme in machine learning research. Multi-modal Variational Autoencoders (VAEs) have been a popular generative model class that learns latent representations which jointly explain multiple modalities. Various objective functions for such models have been suggested, often motivated as lower bounds on the multi-modal data log-likelihood or from information-theoretic considerations. In order to encode latent variables from different modality subsets, Product-of-Experts (PoE) or Mixture-of-Experts (MoE) aggregation schemes have been routinely used and shown to yield different trade-offs, for instance, regarding their generative quality or consistency across multiple modalities. In this work, we consider a variational bound that can tightly approximate the data log-likelihood. We develop more flexible aggregation schemes that generalise PoE or MoE approaches by combining encoded features from different modalities based on permutation-invariant neural networks. Our numerical experiments illustrate trade-offs for multi-modal variational bounds and various aggregation schemes. We show that tighter variational bounds and more flexible aggregation models can become beneficial when one wants to approximate the true joint distribution over observed modalities and latent variables in identifiable models.

## 1 Introduction

Multi-modal data sets where each sample has features from distinct sources have grown in recent years. For example, multi-omics data such as genomics, epigenomics, transcriptomics and metabolomics can provide a more comprehensive understanding of biological systems if multiple modalities are analysed in an integrative framework (Argelaguet et al., 2018; Lee and van der Schaar, 2021; Minoura et al., 2021). In neuroscience, multi-modal integration of neural activity and behavioral data can help to learn latent neural dynamics (Zhou and Wei, 2020; Schneider et al., 2023). However, annotations or labels in such data sets are often rare, making unsupervised or semi-supervised generative approaches particularly attractive as such methods can be used in these settings to (i) generate data, such as missing modalities, and (ii) learn latent representations that are useful for down-stream analyses or that are of scientific interest themselves. The availability of heterogenous data for different modalities promises to learn generalizable representations that can capture shared content across multiple modalities in addition to modality-specific information. A promising class of weakly-supervised generative models is multi-modal VAEs (Suzuki et al., 2016; Wu and Goodman, 2019; Shi et al., 2019; Sutter et al., 2021) that combine information across modalities in an often-shared low-dimensional latent representation. A common route for learning the parameters of latent variable models is via maximization of the marginal data likelihood with various lower bounds thereof suggested in previous work.

**Setup.** We consider a set of $M$ random variables $\{X_1, \ldots, X_M\}$ with empirical density $p_d$, where each random variable $X_s$, $s \in \mathcal{M} = \{1, \ldots, M\}$, can be used to model a different data modality taking values in $\mathsf{X}_s$. With some abuse of notation, we write $X = \{X_1, \ldots, X_M\}$ and for any subset $\mathcal{S} \subset \mathcal{M}$, we set $X = (X_{\mathcal{S}}, X_{\backslash \mathcal{S}})$ for two partitions of the random variables into $X_{\mathcal{S}} = \{X_s\}_{s \in \mathcal{S}}$ and $X_{\backslash \mathcal{S}} = \{X_s\}_{s \in \mathcal{M} \backslash \mathcal{S}}$. We pursue a latent variable model setup, analogous to uni-modal VAEs (Kingma and Ba, 2014; Rezende et al., 2014). For a latent variable $Z \in \mathsf{Z}$ with prior density $p_\theta(z)$,

we posit a joint generative model[1] $p_\theta(z, x) = p_\theta(z) \prod_{s=1}^{M} p_\theta(x_s|z)$, where $p_\theta(x_s|z)$ is commonly referred to as the decoding distribution for modality $s$. Observe that all modalities are independent given the latent variable $z$ shared across all modalities. However, one can introduce modality-specific latent variables by making sparsity assumptions for the decoding distribution.

**Multi-modal variational bounds and mutual information.** Popular approaches to train multi-modal models are based on a mixture-based variational bound (Daunhawer et al., 2022; Shi et al., 2019) given by $\mathcal{L}^{\text{Mix}}(\theta, \phi, \beta) = \int \rho(S) \mathcal{L}_S^{\text{Mix}}(x, \theta, \phi, \beta) \mathrm{d}S$, where

$$\mathcal{L}_S^{\text{Mix}}(x, \theta, \phi, \beta) = \int q_\phi(z|x_S) \left[\log p_\theta(x|z)\right] \mathrm{d}z - \beta \mathsf{KL}(q_\phi(z|x_S)|p_\theta(z)) \tag{1}$$

and $\rho$ is some distribution on the power set $\mathcal{P}(\mathcal{M})$ of $\mathcal{M}$ and $\beta > 0$. For $\beta = 1$, one obtains the bound $\mathcal{L}_S^{\text{Mix}}(x, \theta, \phi, \beta) \leq \log p_\theta(x)$. Variations of (1) have been suggested (Sutter et al., 2020), such as by replacing the prior density $p_\theta$ in the KL-term by a weighted product of the prior density $p_\theta$ and the uni-modal encoding distributions $q_\phi(z|x_s)$, for all $s \in \mathcal{M}$. Maximizing $\mathcal{L}_S^{\text{Mix}}$ can be seen as

$$\text{minimizing } \left\{ \mathcal{H}(X|Z_S) + \beta \, \mathrm{I}_{q_\phi}(X_S, Z_S) = \mathcal{H}(X) - \mathrm{I}_{q_\phi}(X, Z_S) + \beta \, \mathrm{I}_{q_\phi}(X_S, Z_S) \right\}, \tag{2}$$

where $I_q(X, Y) = \int q(x, y) \log \frac{q(x,y)}{q(x)q(y)}$ is the mutual information of random variables $X$ and $Y$ having marginal and joint densities $q$, whilst $\mathcal{H}(X|Y) = -\int q(x, y) \log q(x|y)] \mathrm{d}x\mathrm{d}y$ is the conditional entropy of $X$ given $Y$. We occasionally write $Z_S$ instead of $Z$ to emphasize that $Z$ is conditional on $X_S$ under the encoding density $q_\phi$. Likewise, the multi-view variational information bottleneck approach developed in Lee and van der Schaar (2021) for predicting $x_{\setminus S}$ given $x_S$ can be interpreted as minimizing $-\mathrm{I}_{q_\phi}(X_{\setminus S}, Z) + \beta \, \mathrm{I}_{q_\phi}(X_S, Z)$. Hwang et al. (2021) suggested a related bound that aims to maximize the reduction of total correlation of $X$ when conditioned on $Z$. Similar bounds have been suggested in Sutter et al. (2020) and Suzuki et al. (2016) by considering different KL-regularisation terms, see also Suzuki and Matsuo (2022). Shi et al. (2020) add a contrastive term to the maximum likelihood objective and minimize $-\log p_\theta(x) - \beta \, \mathrm{I}_{p_\theta}(X_S, X_{\setminus S})$.

**Multi-modal aggregation schemes.** In order to optimize the variational bounds above or to allow for flexible conditioning at test time, we need to learn encoding distributions $q_\phi(z|x_S)$ for any $S \in \mathcal{P}(\mathcal{M})$. The typical aggregation schemes that are scalable to a large number of modalities are based on a choice of uni-modal encoding distributions $q_{\phi_s}(z|x_s)$ for any $s \in \mathcal{M}$, which are then used to define the multi-modal encoding distributions as follows:

- Mixture of Experts (MoE), see Shi et al. (2019), $q_\phi^{\text{MoE}}(z|x_S) = \frac{1}{|S|} \sum_{s \in S} q_{\phi_s}(z|x_s)$.

- Product of Experts (PoE), see Wu and Goodman (2018), $q_\phi^{\text{PoE}}(z|x_S) \propto p_\theta(z) \prod_{s \in S} q_{\phi_s}(z|x_s)$.

**Contributions.** This paper contributes (i) a new variational bound as an approximate lower bound on the multi-modal log-likelihood (LLH). We avoid a limitation of mixture-based bounds (1) which may not provide tight lower bounds on the joint LLH if there is considerable modality-specific variation (Daunhawer et al., 2022), even for flexible encoding distributions. The novel variational bound contains a lower bound of the marginal LLH $\log p_\theta(x_S)$ and a term approximating the conditional $\log p_\theta(x_{\setminus S}|x_S)$ for any choice of $S \in \mathcal{P}(\mathcal{M})$, provided that we can learn a flexible multi-modal encoding distribution. This paper then contributes (ii) new multi-modal aggregation schemes that yield more expressive multi-modal encoding distributions when compared to MoEs or PoEs. These schemes are motivated by the flexibility of permutation-invariant (PI) architectures such as DeepSets (Zaheer et al., 2017) or attention models (Vaswani et al., 2017; Lee et al., 2019). We illustrate that these innovations (iii) are beneficial when learning identifiable models, aided by using flexible prior and encoding distributions consisting of mixtures and (iv) yield higher LLH in experiments.

**Further related work.** Canonical Correlation Analysis (Hotelling, 1936; Bach and Jordan, 2005) is a classical approach for multi-modal data that aims to find projections of two modalities by maximally correlating, and has been extended to include more than two modalities (Archambeau and

---

[1] We usually denote random variables using upper-case letters, and their realizations by the corresponding lower-case letter. We assume throughout that $\mathsf{Z} = \mathbb{R}^D$, and that $p_\theta(z)$ is a Lebesgue density, although the results can be extended to more general settings such as discrete random variables $Z$ with appropriate adjustments, for instance, regarding the gradient estimators.

Bach, 2008; Tenenhaus and Tenenhaus, 2011) or to allow for non-linear transformations (Akaho, 2001; Hardoon et al., 2004; Wang et al., 2015; Karami and Schuurmans, 2021). Probabilistic CCA can also be seen as multi-battery factor analysis (MBFA) (Browne, 1980; Klami et al., 2013), wherein a shared latent variable models the variation common to all modalities with modality-specific latent variables capturing the remaining variation. Likewise, latent factor regression or classification models (Stock and Watson, 2002) assume that observed features and response are driven jointly by a latent variable. Vedantam et al. (2018) considered a tiple-ELBO for two modalities, while Sutter et al. (2021) introduced a generalised variational bound that involves a summation over all modality subsets. A series of work has developed multi-modal VAEs based on shared and private latent variables (Wang et al., 2016; Lee and Pavlovic, 2021; Lyu and Fu, 2022; Lyu et al., 2021; Vasco et al., 2022; Palumbo et al., 2023). Tsai et al. (2019) proposed a hybrid generative-discriminative objective and minimized an approximation of the Wasserstein distance between the generated and observed multi-modal data. Joy et al. (2021) consider a semi-supervised setup of two modalities that requires no explicit multi-modal aggregation function, while Bounoua et al. (2023) considered a score-based diffusion on auto-encoded latents. Extending the Info-Max principle (Linsker, 1988), maximizing mutual information $I_q(g_1(X_1), g(X_2)) \leq I_q((X_1, X_2), (Z_1, Z_2))$ based on representations $Z_s = g_s(X_s)$ for modality-specific encoders $g_s$ from two modalities has been a motivation for approaches based on (symmetrised) contrastive objectives (Tian et al., 2020; Zhang et al., 2022c; Daunhawer et al., 2023) such as InfoNCE (Oord et al., 2018; Poole et al., 2019; Wang and Isola, 2020) as a variational lower bound on the mutual information between $Z_1$ and $Z_2$.

## 2 A TIGHTER VARIATIONAL BOUND WITH ARBITRARY MODALITY MASKING

For $\mathcal{S} \subset \mathcal{M}$ and $\beta > 0$, we define

$$\mathcal{L}_{\mathcal{S}}(x_{\mathcal{S}}, \theta, \phi, \beta) = \int q_\phi(z|x_{\mathcal{S}}) \left[\log p_\theta(x_{\mathcal{S}}|z)\right] \mathrm{d}z - \beta \mathsf{KL}(q_\phi(z|x_{\mathcal{S}})|p_\theta(z)). \quad (3)$$

This is simply a standard variational lower bound (Jordan et al., 1999; Blei et al., 2017) restricted to the subset $\mathcal{S}$ for $\beta = 1$, and therefore $\mathcal{L}_{\mathcal{S}}(x_{\mathcal{S}}, \theta, \phi, 1) \leq \log p_\theta(x_{\mathcal{S}})$. To obtain a lower bound on the log-likelihood of all modalities, we introduce an (approximate) conditional lower bound

$$\mathcal{L}_{\backslash \mathcal{S}}(x, \theta, \phi, \beta) = \int q_\phi(z|x) \left[\log p_\theta(x_{\backslash \mathcal{S}}|z)\right] \mathrm{d}z - \beta \mathsf{KL}(q_\phi(z|x)|q_\phi(z|x_{\mathcal{S}})). \quad (4)$$

For some fixed density $\rho$ on $\mathcal{P}(\mathcal{M})$, we suggest the overall bound

$$\mathcal{L}(x, \theta, \phi, \beta) = \int \rho(\mathcal{S}) \left[\mathcal{L}_{\mathcal{S}}(x_{\mathcal{S}}, \theta, \phi, \beta) + \mathcal{L}_{\backslash \mathcal{S}}(x, \theta, \phi, \beta)\right] \mathrm{d}\mathcal{S},$$

which is a generalisation of the bound suggested in Wu and Goodman (2019) to an arbitrary number of modalities. This bound can be optimised using standard Monte Carlo techniques, for example, by computing unbiased pathwise gradients (Kingma and Ba, 2014; Rezende et al., 2014; Titsias and Lázaro-Gredilla, 2014) using the reparameterisation trick. For variational families such as Gaussian mixtures[2], one can employ implicit reparameterisation (Figurnov et al., 2018). It is straightforward to adapt variance reduction techniques such as ignoring the score term of the multi-modal encoding densities for pathwise gradients (Roeder et al., 2017), see Algorithm 1 in Appendix K for pseudo-code. Nevertheless, a scalable approach requires an encoding technique that allows to condition on any masked modalities with a computational complexity that does not increase exponentially in $M$.

**Multi-modal distribution matching.** Likelihood-based learning approaches aim to match the model distribution $p_\theta(x)$ to the true data distribution $p_d(x)$. Variational approaches achieve this by matching in the latent space the encoding distribution to the true posterior as well as maximizing a tight lower bound on $\log p_\theta(x)$, see Rosca et al. (2018). We show similar results for the multi-modal variational bound. Consider therefore the densities $p_\theta(z, x) = p_\theta(z)p_\theta(x_{\mathcal{S}}|z)p_\theta(x_{\backslash \mathcal{S}}|z)$ and $q_\phi(z_{\mathcal{S}}, x) = p_d(x_{\mathcal{S}})q_\phi(z_{\mathcal{S}}|x_{\mathcal{S}})$. The latter is the encoding path comprising the encoding density $q_\phi$ conditioned on $x_{\mathcal{S}}$ and the empirical density $p_d$. We set $q_{\phi, \backslash \mathcal{S}}^{\text{agg}}(z|x_{\mathcal{S}}) = \int p_d(x_{\backslash \mathcal{S}}|x_{\mathcal{S}})q_\phi(z|x)\mathrm{d}x_{\backslash \mathcal{S}}$

---

[2]For MoE aggregation schemes, Shi et al. (2019) considered a stratified ELBO estimator as well as a tighter bound based on importance sampling, see also Morningstar et al. (2021), that we do not pursue here for consistency with other aggregation schemes that can likewise be optimised based on importance sampling ideas.

for an aggregated encoder conditioned on $x_\mathcal{S}$. We provide a multi-model ELBO surgery in Appendix A, summarized in Proposition 9. In particular, we show that maximizing $\int p_d(x_\mathcal{S})\mathcal{L}_\mathcal{S}(x_\mathcal{S}, \theta, \phi)\mathrm{d}x_\mathcal{S}$ drives (i) the joint inference distribution $q_\phi(z, x_\mathcal{S}) = p_d(x_\mathcal{S})q_\phi(z|x_\mathcal{S})$ of the $\mathcal{S}$ submodalities to the joint generative distribution $p_\theta(z, x_\mathcal{S}) = p_\theta(z)p_\theta(x_\mathcal{S}|z)$ and (ii) the generative marginal $p_\theta(x_\mathcal{S})$ to its empirical counterpart $p_d(x_\mathcal{S})$. Analogously, maximizing $\int p_d(x_{\setminus\mathcal{S}}|x_\mathcal{S})\mathcal{L}_{\setminus\mathcal{S}}(x, \theta, \phi)\mathrm{d}x_{\setminus\mathcal{S}}$ drives (i) the distribution $p_d(x_{\setminus\mathcal{S}}|x_\mathcal{S})q_\phi(z|x)$ to the distribution $p_\theta(x_{\setminus\mathcal{S}}|z)q_\phi(z|x_\mathcal{S})$ and (ii) the conditional $p_\theta(x_{\setminus\mathcal{S}}|x_\mathcal{S})$ to its empirical counterpart $p_d(x_{\setminus\mathcal{S}}|x_\mathcal{S})$, provided that $q_\phi(z|x_\mathcal{S})$ approximates $p_\theta(z|x_\mathcal{S})$ exactly. In this case, Proposition 9 implies that $\mathcal{L}_{\setminus\mathcal{S}}(x, \theta, \phi)$ is a lower bound of $\log p_\theta(x_{\setminus\mathcal{S}}|x_\mathcal{S})$. Furthermore, it shows that maximizing $\mathcal{L}_{\setminus\mathcal{S}}(x, \theta, \phi)$ minimizes a Bayes-consistency matching term $\mathsf{KL}(q_{\phi,\setminus\mathcal{S}}^{\mathrm{agg}}(z|x_\mathcal{S})|q_\phi(z|x_\mathcal{S}))$ for the multi-modal encoders where a mismatch can yield poor cross-generation, as an analogue of the prior not matching the aggregated posterior (Makhzani et al., 2016) leading to poor unconditional generation, see Remark 10. Our approach recovers meta-learning with (latent) Neural processes (Garnelo et al., 2018b) when one optimizes only $\mathcal{L}_{\setminus\mathcal{S}}$ with $\mathcal{S}$ determined by context-target splits, cf. Appendix B. Our analysis implies that $\mathcal{L}_\mathcal{S} + \mathcal{L}_{\setminus\mathcal{L}}$ is an approximate lower bound on the multi-modal log-likelihood that becomes tight for infinite-capacity encoders and is a true lower bound if $\mathsf{KL}(q_{\phi,\setminus\mathcal{S}}^{\mathrm{agg}}(z|x_\mathcal{S})|q_\phi(z|x_\mathcal{S})) = 0$, see Remarks 12 and 13 for details.

**Corollary 1** (Multi-modal log-likelihood approximation). *For any modality mask $\mathcal{S}$, we have*

$$\int p_d(x)\left[\mathcal{L}_\mathcal{S}(x_\mathcal{S}, \theta, \phi, 1) + \mathcal{L}_{\setminus\mathcal{S}}(x, \theta, \phi, 1)\right]\mathrm{d}x - \int p_d(x)\left[\log p_\theta(x)\right]\mathrm{d}x$$

$$= -\int p_d(x_\mathcal{S})\left[\mathsf{KL}(q_\phi(z|x_\mathcal{S})|p_\theta(z|x_\mathcal{S}))\right]\mathrm{d}x - \int p_d(x)\left[\mathsf{KL}(q_\phi(z|x)|p_\theta(z|x))\right]\mathrm{d}x$$

$$+ \int p_d(x)q_\phi(z|x)\left[\log\frac{q_\phi(z|x_\mathcal{S})}{p_\theta(z|x_\mathcal{S})}\right]\mathrm{d}z\mathrm{d}x.$$

**Information-theoretic perspective.** Beyond generative modelling, $\beta$-VAEs (Higgins et al., 2017) have been popular for representation learning and data reconstruction. Alemi et al. (2018) suggest learning a latent representation that achieves certain mutual information with the data based on upper and lower variational bounds of the mutual information. A Legendre transformation thereof recovers the $\beta$-VAE objective and allows a trade-off between information content or rate versus reconstruction quality or distortion. We show that the proposed variational objective gives rise to an analogous perspective for multiple modalities. Recall that mutual information $\mathrm{I}_{q_\phi}(X_\mathcal{S}, Z)$ can be bounded by standard (Barber and Agakov, 2004; Alemi et al., 2016; 2018) lower and upper bounds:

$$\mathcal{H}_\mathcal{S} - D_\mathcal{S} \leq \mathcal{H}_\mathcal{S} - D_\mathcal{S} + \Delta_1 = \mathrm{I}_{q_\phi}(X_\mathcal{S}, Z) = R_\mathcal{S} - \Delta_2 \leq R_\mathcal{S}, \tag{5}$$

with $\Delta_1, \Delta_2 \geq 0$ for the rate $R_\mathcal{S} = \int p_d(x_\mathcal{S})\mathsf{KL}(q_\phi(z|x_\mathcal{S})|p_\theta(z))\mathrm{d}x_\mathcal{S}$ measuring the information content that is encoded by $q_\phi$ into the latents, and the distortion $D_\mathcal{S} = -\int q_\phi(x_\mathcal{S}, z)\log p_\theta(x_\mathcal{S}|z)\mathrm{d}z\mathrm{d}x_\mathcal{S}$ given as the negative reconstruction log-likelihood. Observe that $-\int p_d(x_\mathcal{S})\mathcal{L}(x_\mathcal{S})\mathrm{d}x_\mathcal{S} = D_\mathcal{S} + \beta R_\mathcal{S}$ and for any $\beta > 0$, it holds that $\mathcal{H}_\mathcal{S} \leq R_\mathcal{S} + D_\mathcal{S}$. To arrive at a similar interpretation for the conditional bound $\mathcal{L}_{\setminus\mathcal{S}}$, we set $R_{\setminus\mathcal{S}} = \int p_d(x)\mathsf{KL}(q_\phi(z|x)|q_\phi(z|x_\mathcal{S}))\mathrm{d}x$ for a conditional or cross rate. Similarly, set $D_{\setminus\mathcal{S}} = -\int p_d(x)q_\phi(z|x)\log p_\theta(x_{\setminus\mathcal{S}}|z)\mathrm{d}z\mathrm{d}x$. One obtains the following bounds, see Appendix A.

**Lemma 2** (Variational bounds on the conditional mutual information). *It holds that* $-\int \mathcal{L}_{\setminus\mathcal{S}}(x, \theta, \phi, \beta)p_d(\mathrm{d}x) = D_{\setminus\mathcal{S}} + \beta R_{\setminus\mathcal{S}}$ *and for $\Delta_{\setminus\mathcal{S},1}, \Delta_{\setminus\mathcal{S},2} \geq 0$,*

$$\mathcal{H}_{\setminus\mathcal{S}} - D_{\setminus\mathcal{S}} + \Delta_{\setminus\mathcal{S},1} = \mathrm{I}_{q_\phi}(X_{\setminus\mathcal{S}}, Z_\mathcal{M}|X_\mathcal{S}) = R_{\setminus\mathcal{S}} - \Delta_{\setminus\mathcal{S},2}.$$

Using the chain rules for entropy, we obtain that the suggested bound can be seen as a relaxation of bounds on marginal and conditional mutual information.

**Corollary 3** (Lagrangian relaxation). *It holds that*

$$\mathcal{H} - D_\mathcal{S} - D_{\setminus\mathcal{S}} \leq \mathrm{I}_{q_\phi}(X_\mathcal{S}, Z_\mathcal{S}) + \mathrm{I}_{q_\phi}(X_{\setminus\mathcal{S}}, Z_\mathcal{M}|X_\mathcal{S}) \leq R_\mathcal{S} + R_{\setminus\mathcal{S}}$$

*and minimizing $\mathcal{L}$ for fixed $\beta = \frac{\partial(D_\mathcal{S} + D_{\setminus\mathcal{S}})}{\partial(R_\mathcal{S} + R_{\setminus\mathcal{S}})}$ minimizes the rates $R_\mathcal{S} + R_{\setminus\mathcal{S}}$ and distortions $D_\mathcal{S} + D_{\setminus\mathcal{S}}$.*

**Remark 4** (Mixture based variational bound). The arguments in Daunhawer et al. (2022) imply that $-\int p_d(\mathrm{d}x)\mathcal{L}_\mathcal{S}^{\mathrm{Mix}}(x) = D_\mathcal{S} + D_{\setminus\mathcal{S}}^{\mathrm{c}} + \beta R_\mathcal{S}$, where $D_{\setminus\mathcal{S}}^{\mathrm{c}} = -\int p_d(x_\mathcal{S})q_\phi(z|x_\mathcal{S})\log p_\theta(x_{\setminus\mathcal{S}}|z)\mathrm{d}z\mathrm{d}x_\mathcal{S}$ is a cross-distortion term. Due to $\mathcal{H}(X_\mathcal{M}|Z_\mathcal{S}) = -\mathcal{H}(X_\mathcal{M}) + \mathrm{I}_{q_\phi}(X_\mathcal{M}, Z_\mathcal{S}) \leq D_\mathcal{S} + D_{\setminus\mathcal{S}}^{\mathrm{c}}$, we can view minimizing $\mathcal{L}_\mathcal{S}^{\mathrm{Mix}}$ as minimizing $\mathcal{H}(X_\mathcal{M}) - \mathrm{I}_{q_\phi}(X_\mathcal{M}, Z_\mathcal{S}) + \beta\,\mathrm{I}_{q_\phi}(X_\mathcal{S}, Z_\mathcal{S})$, see (2).

**Optimal variational distributions.** Consider the annealed likelihood $\tilde{p}_{\beta,\theta}(x_\mathcal{S}|z) \propto p_\theta(x_\mathcal{S}|z)^{1/\beta}$ as well as the adjusted posterior $\tilde{p}_{\beta,\theta}(z|x_\mathcal{S}) \propto \tilde{p}_{\beta,\theta}(x_\mathcal{S}|z)p_\theta(z)$. The minimum of the bound $\int p_d(\mathrm{d}x)\mathcal{L}_\mathcal{S}(x)$ is attained at any $x_\mathcal{S}$ for the variational density

$$q^\star(z|x_\mathcal{S}) \propto \exp\left(\frac{1}{\beta}\left[\log p_\theta(x_\mathcal{S}|z) + \beta \log p_\theta(z)\right]\right) \propto \tilde{p}_{\beta,\theta}(z|x_\mathcal{S}), \tag{6}$$

see also Huang et al. (2020). Similarly, if (6) holds, then it is readily seen that the minimum of the bound $\int p_d(\mathrm{d}x)\mathcal{L}_{\backslash\mathcal{S}}(x)$ is attained at any $x$ for the variational density $q^\star(z|x) = \tilde{p}_{\beta,\theta}(z|x)$. In contrast, as shown in Appendix D, the optimal variational density for the mixture-based (1) multi-modal objective is attained at $q^\star(z|x_\mathcal{S}) \propto \tilde{p}_{\beta,\theta}(z|x_\mathcal{S})\exp\left(\int p_d(x_{\backslash\mathcal{S}}|x_\mathcal{S})\log\tilde{p}_{\beta,\theta}(x_{\backslash\mathcal{S}}|z)\mathrm{d}x_{\backslash\mathcal{S}}\right)$.

## 3  PERMUTATION-INVARIANT MODALITY ENCODING

**Fixed multi-modal aggregation schemes.** Optimizing these multi-modal bounds requires learning variational densities with different conditioning sets. We write $h_{s,\varphi}\colon \mathsf{X}_s \mapsto \mathbb{R}^{D_E}$ for some modality-specific feature function. We recall the following multi-modal encoding functions suggested in previous work where usually $h_{s,\varphi}(x_s) = \left[\mu_{s,\varphi}(x_s)^\top, \mathrm{vec}(\Sigma_{s,\varphi}(x_s))^\top\right]^\top$ with $\mu_{s,\varphi}$ and $\Sigma_{s,\varphi}$ being the mean, respectively the (often diagonal) covariance, of a uni-modal encoder of modality $s$. Accommodating more complex variational families, such as mixture distributions for the uni-modal encoding distributions, can be more challenging for these approaches.

- MoE: $q_\varphi^{\mathrm{MoE}}(z|x_\mathcal{S}) = \frac{1}{|\mathcal{S}|}\sum_{s\in\mathcal{S}} q_\mathcal{N}(z|\mu_{s,\varphi}(x_s), \Sigma_{s,\varphi}(x_s))$, where $q_\mathcal{N}(z|\mu,\Sigma)$ is a Gaussian density with mean $\mu$ and covariance $\Sigma$.

- PoE: $q_\varphi^{\mathrm{PoE}}(z|x_\mathcal{S}) = \frac{1}{\mathcal{Z}} p_\theta(z)\prod_{s\in\mathcal{S}} q_\mathcal{N}(z|\mu_{s,\varphi}(x_s), \Sigma_{s,\varphi}(x_s))$, for some $\mathcal{Z}\in\mathbb{R}$. For Gaussian priors $p_\theta(z) = q_\mathcal{N}(z|\mu_\theta, \Sigma_\theta)$ with mean $\mu_\theta$ and covariance $\Sigma_\theta$, the multi-modal distribution $q_\varphi^{\mathrm{PoE}}(z|x_\mathcal{S})$ is Gaussian with mean $(\mu_\theta\Sigma_\theta + \sum_{s\in\mathcal{S}}\mu_{s,\varphi}(x_s)\Sigma_{s,\varphi}(x_s))(\Sigma_{1,\theta}^{-1} + \sum_{s\in\mathcal{S}}\Sigma_{s,\varphi}(x_s)^{-1})^{-1}$ and covariance $(\Sigma_{1,\theta}^{-1} + \sum_{s\in\mathcal{S}}\Sigma_{s,\varphi}(x_s)^{-1})^{-1}$.

**Learnable multi-modal aggregation schemes.** We aim to learn a more flexible aggregation scheme under the constraint that the encoding distribution is invariant (Bloem-Reddy and Teh, 2020) with respect to the ordering of encoded features of each modality. Put differently, for all $(H_s)_{s\in\mathcal{S}} \in \mathbb{R}^{|\mathcal{S}|\times D_E}$ and all permutations $\pi \in \mathbb{S}_\mathcal{S}$ of $\mathcal{S}$, we assume that the conditional distribution is $\mathbb{S}_\mathcal{S}$-invariant, i.e. $q_\vartheta'(z|h) = q_\vartheta'(z|\pi\cdot h)$ for all $z\in\mathbb{R}^D$, where $\pi$ acts on $H = (H_s)_{s\in\mathcal{S}}$ via $\pi\cdot H = (H_{\pi(s)})_{s\in\mathcal{S}}$. We set $q_\phi(z|x_\mathcal{S}) = q_\vartheta'(z|h_{s,\varphi}(x_s)_{s\in\mathcal{S}})$, $\phi = (\varphi, \vartheta)$ and remark that the encoding distribution is not invariant with respect to the modalities, but becomes only invariant after applying modality-specific encoder functions $h_{s,\varphi}$. Observe that such a constraint is satisfied by the aggregation schemes above for $h_{s,\varphi}$ being the uni-modal encoders.

A variety of invariant (or equivariant) functions along with their approximation properties have been considered previously, see for instance Santoro et al. (2017); Zaheer et al. (2017); Qi et al. (2017); Lee et al. (2019); Segol and Lipman (2019); Murphy et al. (2019); Maron et al. (2019); Sannai et al. (2019); Yun et al. (2019); Bruno et al. (2021); Wagstaff et al. (2022); Zhang et al. (2022b); Li et al. (2022); Bartunov et al. (2022), and applied in different contexts such as meta-learning (Edwards and Storkey, 2016; Garnelo et al., 2018b; Kim et al., 2018; Hewitt et al., 2018; Giannone and Winther, 2022), reinforcement learning (Tang and Ha, 2021; Zhang et al., 2022a) or generative modeling of (uni-modal) sets (Li et al., 2018; 2020; Kim et al., 2021; Biloš and Günnemann, 2021; Li and Oliva, 2021). We can use such constructions to parameterise more flexible encoding distributions. Indeed, the results from Bloem-Reddy and Teh (2020) imply that for an exchangable sequence $H_\mathcal{S} = (H_s)_{s\in\mathcal{S}} \in \mathbb{R}^{|\mathcal{S}|\times D_E}$ and random variable $Z$, the distribution $q'(z|h_\mathcal{S})$ is $\mathbb{S}_\mathcal{S}$-invariant if and only if there is a measurable function[3] $f^\star\colon [0,1]\times\mathcal{M}(\mathbb{R}^{D_E})\to\mathbb{R}^D$ such that

$$(H_\mathcal{S}, Z) \overset{\mathrm{a.s.}}{=} (H_\mathcal{S}, f^\star(\Xi, \mathbb{M}_{H_\mathcal{S}})), \text{ where } \Xi\sim\mathcal{U}[0,1] \text{ and } \Xi \perp\!\!\!\perp H_\mathcal{S}$$

with $\mathbb{M}_{H_\mathcal{S}}(\cdot) = \sum_{s\in\mathcal{S}}\delta_{H_s}(\cdot)$ being the empirical measure of $h_\mathcal{S}$, which retains the values of $h_\mathcal{S}$, but discards their order. For variational densities from a location-scale family such as a Gaussian

---

[3]The function $f^\star$ generally depends on the cardinality of $\mathcal{S}$. Finite-length exchangeable sequences imply a de Finetti latent variable representation only up to approximation errors (Diaconis and Freedman, 1980).

or Laplace distribution, we find it more practical to consider a different reparameterisation in the form $Z = \mu(h_{\mathcal{S}}) + \sigma(h_{\mathcal{S}}) \odot \Xi$, where $\Xi$ is a sample from a parameter-free density $p$ such as a standard Gaussian and Laplace distribution, while $[\mu(h_{\mathcal{S}}), \log \sigma(h_{\mathcal{S}})] = f(h_{\mathcal{S}})$ for a PI function $f \colon \mathbb{R}^{|\mathcal{S}| \times D_E} \to \mathbb{R}^{2D}$. Likewise, for mixture distributions thereof, assume that for a PI function $f$,

$$[\mu_1(h_{\mathcal{S}}), \log \sigma_1(h_{\mathcal{S}}), \ldots, \mu_K(h_{\mathcal{S}}), \log \sigma_K(h_{\mathcal{S}}), \log \omega(h_{\mathcal{S}})] = f(h_{\mathcal{S}}) \in \mathbb{R}^{2DK+K}$$

and $Z = \mu_L(h_{\mathcal{S}}) + \sigma_L(h_{\mathcal{S}}) \odot \Xi$ with $L \sim \mathrm{Cat}(\omega(h_{\mathcal{S}}))$ denoting the sampled mixture component out of $K$ mixtures. For simplicity, we consider here only two examples of PI functions $f$ that have representations with parameter $\vartheta$ in the form $f_\vartheta(h_{\mathcal{S}}) = \rho_\vartheta \left( \sum_{s \in \mathcal{S}} g_\vartheta(h_{\mathcal{S}})_s \right)$ for a function $\rho_\vartheta \colon \mathbb{R}^{D_P} \to \mathbb{R}^{D_O}$ and permutation-equivariant function $g_\vartheta \colon \mathbb{R}^{N \times D_E} \to \mathbb{R}^{N \times D_P}$.

**Example 5** (Sum Pooling Encoders)**.** The Deep Set (Zaheer et al., 2017) construction $f_\vartheta(h_{\mathcal{S}}) = \rho_\vartheta \left( \sum_{s \in \mathcal{S}} \chi_\vartheta(h_s) \right)$ applies the same neural network $\chi_\vartheta \colon \mathbb{R}^{D_E} \to \mathbb{R}^{D_P}$ to each encoded feature $h_s$. We assume that $\chi_\vartheta$ is a feed-forward neural network, and remark that pre-activation ResNets (He et al., 2016) have been advocated in for deeper $\chi_\vartheta$. For exponential family models, the optimal natural parameters of the posterior solve an optimisation problem where the dependence on the generative parameters from the different modalities decomposes as a sum, see Appendix G.

**Example 6** (Set Transformer Encoders)**.** Let $\mathrm{MTB}_\vartheta$ be a multi-head pre-layer-norm transformer block (Wang et al., 2019; Xiong et al., 2020), see Appendix E for precise definitions. For some neural network $\chi_\vartheta \colon \mathbb{R}^{D_E} \to \mathbb{R}^{D_P}$, set $g_{\mathcal{S}}^0 = \chi_\vartheta(h_{\mathcal{S}})$ and for $k \in \{1, \ldots, L\}$, set $g_{\mathcal{S}}^k = \mathrm{MTB}_\vartheta(g_{\mathcal{S}}^{k-1})$. We then consider $f_\vartheta(h_{\mathcal{S}}) = \rho_\vartheta \left( \sum_{s \in \mathcal{S}} g_s^L \right)$. This can be seen as a Set Transformer (Lee et al., 2019; Zhang et al., 2022a) model without any inducing points as for most applications, a computational complexity that scales quadratically in the number of modalities can be acceptable. In our experiments, we use layer normalisation (Ba et al., 2016) within the transformer model, although, for example, set normalisation (Zhang et al., 2022a) could be used alternatively.

**Remark 7** (Pooling expert opinions)**.** Combining expert distributions has a long tradition in decision theory and Bayesian inference, see Genest and Zidek (1986) for early works, with popular schemes being linear pooling (i.e., MoE) or log-linear pooling (i.e., PoE with tempered densities). These are optimal schemes for minimizing different objectives, namely a weighted (forward or reverse) KL-divergence between the pooled distribution and the inidividual experts (Abbas, 2009). Log-linear pooling operators are externally Bayesian, that is, they allow for consistent Bayesian belief updates when each expert updates her belief with the same likelihood function (Genest et al., 1986).

**Permutation-equivariance and private latent variables.** Suppose that the generative model factorises as $p_\theta(z, x) = p(z) \prod_{s \in \mathcal{M}} p_\theta(x_s | z', \tilde{z}_s)$ with $z = (z', \tilde{z}_1, \ldots, \tilde{z}_M)$, where $Z'$ and $\tilde{Z}^s$, $s \in \mathcal{M}$ are shared, resp., private latent variables. For $s \neq t \in [M]$, we have $h_{\varphi,s}(X_s) \perp\!\!\!\perp \tilde{Z}_t \mid Z', \tilde{Z}_s$. Assuming that the modality-specific feature functions $h_{\varphi,s}$ are such that $\{H_s = h_{\varphi,s}(X_s)\}_{s \in \mathcal{S}}$ is exchangeable, the results from Bloem-Reddy and Teh (2020) imply a permutation-equivariant (PE) representation of the private latent variables, conditional on the shared latent variables. This suggests to consider encoders for the private latent variables that satisfy $q'_\phi(\tilde{z}_{\mathcal{S}} | \pi \cdot h_\varphi(x_{\mathcal{S}}), z') = q'_\phi(\pi \cdot \tilde{z}_{\mathcal{S}} | h_\varphi(x_{\mathcal{S}}), z')$ for any permutation $\pi \in \mathbb{S}_{\mathcal{S}}$. Details are given in Appendix F, including PE versions of PoEs, SumPooling and SelfAttention aggregations.

## 4 IDENTIFIABILITY AND MODEL EXTENSIONS

**Identifiability.** Non-linear generative models are generally unidentifiable without imposing some structure (Hyvärinen and Pajunen, 1999; Xi and Bloem-Reddy, 2022). Yet, identifiability up to some ambiguity can be achieved in some conditional models based on observed auxiliary variables and injective decoder functions wherein the prior density is conditional on auxiliary variables. Observations from different modalities can act as auxiliary variables to obtain identifiability of conditional distributions given some modality subset under analogous assumptions, see Appendix H.

**Example 8** (Auxiliary variable as a modality)**.** In the iVAE model (Khemakhem et al., 2020a), the latent variable distribution $p_\theta(z | x_1)$ is independently modulated via an auxiliary variable $X_1 = U$. Instead of interpreting this distribution as a (conditional) prior density, we view it as a posterior density given the first modality $X_1$. Khemakhem et al. (2020a) estimate a model for another modality $X_2$ by lower bounding $\log p_\theta(x_2 | x_1)$ via $\mathcal{L}_{\backslash\{1\}}$ under the assumption that $q_\phi(z | x_1)$ is given by the

prior density $p_\theta(z|x_1)$. Similarly, Mita et al. (2021) optimise $\log p_\theta(x_1, x_2)$ by a double VAE bound that reduces to $\mathcal{L}$ for a masking distribution $\rho(s_1, s_2) = (\delta_1 \otimes \delta_0)(s_1, s_2)$ that always masks the modality $X_2$ and choosing to parameterise separate encoding functions for different conditioning sets. Our bound thus generalises these procedures to multiple modalities in a scalable way.

**Mixture models.** An alternative to the choice of uni-modal prior densities $p_\theta$ has been to use Gaussian mixture priors (Johnson et al., 2016; Jiang et al., 2017; Dilokthanakul et al., 2016) or more flexible mixture models (Falck et al., 2021). Following previous work, we include a latent cluster indicator variable $c \in [K]$ that indicates the mixture component out of $K$ possible mixtures with augmented prior $p_\theta(c, z) = p_\theta(c)p_\theta(z|c)$. The classic example is $p_\theta(c)$ being a categorical distribution and $p_\theta(z|c)$ a Gaussian with mean $\mu_c$ and covariance matrix $\Sigma_c$. Similar to Falck et al. (2021) that use an optimal variational factor in a mean-field model, we use an optimal factor of the cluster indicator in a structured variational density $q_\phi(c, z|x_\mathcal{S}) = q_\phi(z|x_\mathcal{S})q_\phi(c|z, x_\mathcal{S})$ with $q_\phi(c|z, x_\mathcal{S}) = p_\theta(c|z)$. Appendix J details how one can optimize an augmented multi-modal bound.

## 5 EXPERIMENTS

### 5.1 LINEAR MULTI-MODAL VAES

The relationship between uni-modal VAEs and probabilistic PCA (Tipping and Bishop, 1999) has been studied in previous work (Dai et al., 2018; Lucas et al., 2019; Rolinek et al., 2019; Huang et al., 2020; Mathieu et al., 2019). We analyse how different multimodal fusion schemes and multi-modal variational bounds affect (a) the learned generative model in terms of its true marginal log-likelihood (LLH) and (b)

Table 1: Gaussian model: Relative difference of true LLH to the learned LLH. MCC to true latent.

| Aggregation | Our bound | | Mixture bound | |
|---|---|---|---|---|
| | LLH Gap | MCC | LLH Gap | MCC |
| PoE | 0.03 (0.058) | 0.75 (0.20) | 0.04 (0.074) | 0.77 (0.21) |
| MoE | 0.01 (0.005) | 0.82 (0.04) | 0.02 (0.006) | 0.67 (0.03) |
| SumPooling | **0.00 (0.000)** | **0.84 (0.00)** | 0.00 (0.002) | **0.84 (0.02)** |
| SelfAttention | **0.00 (0.003)** | **0.84 (0.00)** | 0.02 (0.007) | 0.83 (0.00) |

the latent representations. In order to evaluate the (weak) identifiability of the method, we follow Khemakhem et al. (2020a;b) to compute the mean correlation co-efficient (MCC) between the true latent variables $Z$ and samples from the variational distribution $q_\phi(\cdot|x_\mathcal{M})$ after an affine transformation using CCA. Our simulation study uses $M = 5$ modalities, see Appendix M for details about the data generation mechanisms[4] with results given in Table 1. Our results suggest that first, more flexible aggregation schemes improve the LLH and the identifiability for both variational objectives. Second, our new bound yields higher LLH for given aggregation scheme.

### 5.2 NON-LINEAR IDENTIFIABLE MODELS

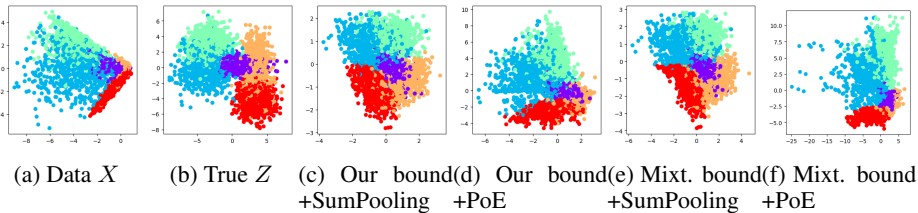

(a) Data $X$    (b) True $Z$    (c) Our bound +SumPooling    (d) Our bound +PoE    (e) Mixt. bound +SumPooling    (f) Mixt. bound +PoE

Figure 1: Continuous modality in (a), true latent variables in (b) and inferred latent variables in (c)-(f) with a linear transformation indeterminancy. Labels are colour coded.

---

[4]We present here results when all latent variables are shared across all modalities. We also consider in Appendix M the generative setting where only parts of the latent variables are shared across all modalities with the remaining latent variables being modality specific. The latter setting can be incorporated by imposing sparsity structures on the decoders and allows us to analyse scenarios with considerable modality-specific variation described through private latent variables with results given in Table 4.

**Auxiliary labels as modalities.** We construct artificial data following Khemakhem et al. (2020a), with the latent variables $Z \in \mathbb{R}^D$ being conditionally Gaussian having means and variances that depend on an observed index value $X_2 \in [K]$. More precisely, $p_\theta(z|x_2) = \mathcal{N}(\mu_{x_2}, \Sigma_{x_2})$, where $\mu_c \sim \otimes\, \mathcal{U}(-5, 5)$ and $\Sigma_c = \text{diag}(\Lambda_c)$, $\Lambda_c \sim \otimes\, \mathcal{U}(0.5, 3)$ iid for $c \in [K]$. The marginal distribution over the labels is uniform $\mathcal{U}([K])$ so that the prior density $p_\theta(z) = \int_{[K]} p_\theta(z|x_2) p_\theta(x_2) \mathrm{d}x_2$ becomes a Gaussian mixture. We choose an injective decoding function $f_1 \colon \mathbb{R}^D \to \mathbb{R}^{D_1}$, $D \leq D_1$, as a composition of MLPs with LeakyReLUs and full rank weight matrices having monotonically increasing row dimensions (Khemakhem et al., 2020b), with iid randomly sampled entries. We assume $X_1|Z \sim \mathcal{N}(f_1(Z), \sigma^2\, \mathrm{I})$ and set $\sigma = 0.1$, $D = D_1 = 2$. $f_1$ has a single hidden layer of size $D_1 = 2$. One realisation of bi-modal data $X$, the true latent variable $Z$, as well as inferred latent variables for a selection of different bounds and aggregation schemes, are shown in Figure 1, with more examples given in Figures 6 and 7. Table 6 indicate that both a tighter variational bound and more flexible aggregation schemes improve the identifiability of the latent variables and the LLH.

**Multiple modalities.** Considering the same generative model for $Z$ with a Gaussian mixture prior, suppose now that instead of observing the auxiliary label, we observe multiple modalities $X_s \in \mathbb{R}^{D_s}$, $X_s|Z \sim \mathcal{N}(f_s(Z), \sigma^2\, \mathrm{I})$, for injective MLPs $f_s$ constructed as above, with $D = 10$, $D_s = 25$, $\sigma = 0.5$ and $K = M = 5$. We consider a semi-supervised setting where modalities are missing completely at random, as in Zhang et al. (2019), with a missing rate $\eta$ as the sample average of $\frac{1}{|\mathcal{M}|} \sum_{s \in \mathcal{M}} (1 - M_s)$. Our

Table 2: Partially observed ($\eta = 0.5$) non-linear identifiable model with 5 modalities: The first four rows use a fixed standard Gaussian prior, while the last four rows use a Gaussian mixture prior.

| | Our bound | | Mixture | |
|---|---|---|---|---|
| Aggregation | LLH | MCC | LLH | MCC |
| PoE | -250.9 (5.19) | 0.94 (0.015) | -288.4 (8.53) | 0.93 (0.018) |
| MoE | -250.1 (4.77) | 0.92 (0.022) | -286.2 (7.63) | 0.90 (0.019) |
| SumPooling | -249.6 (4.85) | 0.95 (0.016) | -275.6 (7.35) | 0.92 (0.031) |
| SelfAttention | -249.7 (4.83) | 0.95 (0.014) | -275.5 (7.45) | 0.93 (0.022) |
| SumPooling | -247.3 (4.23) | 0.95 (0.009) | -269.6 (7.42) | 0.94 (0.018) |
| SelfAttention | -247.5 (4.22) | 0.95 (0.013) | -269.9 (6.06) | 0.93 (0.022) |
| SumPoolingMixture | **-244.8 (4.44)** | 0.95 (0.011) | -271.9 (6.54) | 0.93 (0.021) |
| SelfAttentionMixture | -245.4 (4.55) | **0.96 (0.010)** | -270.3 (5.96) | 0.94 (0.016) |

bound and the suggested PI aggregation schemes can naturally accommodate this partially observed setting, see Appendix I. Table 2 shows that using the new variational bound improves the LLH and the identifiability of the latent representation. Furthermore, using learnable aggregation schemes benefits both variational bounds.

## 5.3 MNIST-SVHN-Text

Following previous work (Sutter et al., 2020; 2021; Javaloy et al., 2022), we consider a tri-modal dataset based on augmenting the MNIST-SVHN dataset (Shi et al., 2019) with a text-based modality. Herein, SVHN consists of relatively noisy images, whilst MNIST and text are clearer modalities. Multi-modal VAEs have been shown to exhibit differing performances relative to their multi-modal coherence, latent classification accuracy or test LLH, see Appendix L for definitions. Previous works often differ in their hyperparameters, from neural network architectures, latent space dimensions, priors and likelihood families, likelihood weightings, decoder variances, etc. We have chosen the same hyperparameters for all models, thereby providing a clearer disentanglement of how either the variational objective or the aggregation scheme affect different multi-modal evaluation measures. In particular, we consider multi-modal generative models with (i) shared latent variables and (ii) private and shared latent variables. We also consider PoE or MoE schemes (denoted PoE+, resp., MoE+) with additional neural network layers in their modality-specific encoding functions so that the number of parameters matches or exceeds those of the introduced PI models, see Appendix P.5 for details. For models without private latent variables, estimates of the test LLHs in Table 3 suggest that our bound improves the LLH across different aggregation schemes for all modalities and different $\beta$s (Table 8), with similar results for PE schemes, except for a Self-Attention model. More flexible fusion schemes yield higher LLHs for both bounds. Qualitative results for the reconstructed modalities are given in Figures 10-12. Realistic cross-generation of the SVHN modality is challenging for the mixture-based bound with all aggregation schemes. In contrast, our bound, particularly when combined with learnable aggregation schemes, improves the cross-generation of SVHN. No

bound or aggregation scheme performs best across all modalities by the generative coherence measures (see Table 9 for uni-modal inputs, Table 10 for bi-modal ones and Tables 11- 14 for models with private latent variables and different $\beta$s), along with reported results from external baselines (MVAE, MMVAE, MoPoE, MMJSD, MVTCAE). Overall, our bound is slightly more coherent for cross-generating SVHN or Text, but less coherent for MNIST. Mixture based bounds tend to improve the unsupervised latent classification accuracy across different fusion approaches and modalities, see Table 15. To provide complementary insights into the trade-offs for the different bounds and fusion schemes, we consider a multi-modal rate-distortion evaluation in Figure 2. Ignoring MoE where reconstructions are similar, our bound improves the full reconstruction, with higher full rates, and across various fusion schemes. Mixture-based bounds yield improved cross-reconstructions for all aggregation models, with increased cross-rates terms. Flexible PI architectures for our bound improve the full reconstruction, even at lower full rates.

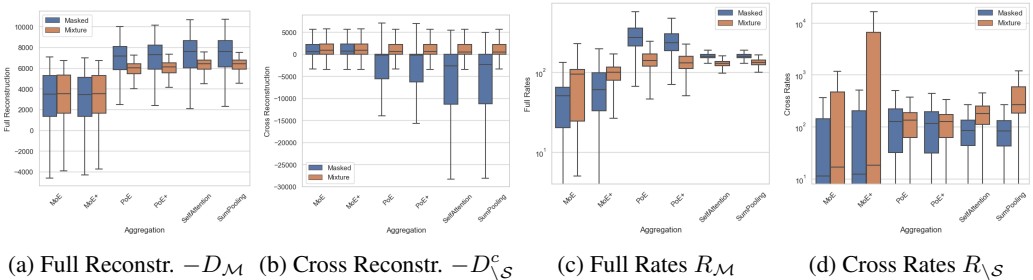

(a) Full Reconstr. $-D_{\mathcal{M}}$  (b) Cross Reconstr. $-D^c_{\backslash\mathcal{S}}$  (c) Full Rates $R_{\mathcal{M}}$  (d) Cross Rates $R_{\backslash\mathcal{S}}$

Figure 2: Rate and distortion terms for MNIST-SVHN-Text with shared latent variables ($\beta = 1$).

Table 3: Test LLH estimates for the joint data (M+S+T) and marginal data (importance sampling with 512 particles). The first part of the table is based on the same generative model with shared latent variable $Z \in \mathbb{R}^{40}$, while the second part of the table is based on a restrictive generative model with a shared latent variable $Z' \in \mathbb{R}^{10}$ and modality-specific latent variables $\tilde{Z}_s \in \mathbb{R}^{10}$.

| | Our bound | | | | Mixture bound | | | |
|---|---|---|---|---|---|---|---|---|
| Aggregation | M+S+T | M | S | T | M+S+T | M | S | T |
| PoE+ | 6872 (9.62) | **2599 (5.6)** | 4317 (1.1) | -9 (0.2) | 5900 (10) | 2449 (10.4) | 3443 (11.7) | -19 (0.4) |
| PoE | 6775 (54.9) | 2585 (18.7) | 4250 (8.1) | -10 (2.2) | 5813 (1.2) | 2432 (11.6) | 3390 (17.5) | -19 (0.1) |
| MoE+ | 5428 (73.5) | 2391 (104) | 3378 (92.9) | -74 (88.7) | 5420 (60.1) | 2364 (33.5) | 3350 (58.1) | -112 (133.4) |
| MoE | 5597 (26.7) | 2449 (7.6) | 3557 (26.4) | -11 (0.1) | 5485 (4.6) | 2343 (1.8) | 3415 (5.0) | -17 (0.4) |
| SumPooling | **7056 (124)** | 2478 (9.3) | **4640 (114)** | **-6 (0.0)** | 6130 (4.4) | 2470 (10.3) | 3660 (1.5) | -16 (1.6) |
| SelfAttention | **7011 (57.9)** | 2508 (18.2) | **4555 (38.1)** | -7 (0.5) | 6127 (26.1) | 2510 (12.7) | 3621 (8.5) | -13 (0.2) |
| PoE+ | 6549 (33.2) | 2509 (7.8) | 4095 (37.2) | -7 (0.2) | 5869 (29.6) | 2465 (4.3) | 3431 (8.3) | -19 (1.7) |
| SumPooling | 6337 (24.0) | 2483 (9.8) | 3965 (16.9) | **-6 (0.2)** | 5930 (23.8) | 2468 (16.8) | 3491 (18.3) | -7 (0.1) |
| SelfAttention | 6662 (20.0) | 2516 (8.8) | 4247 (31.2) | **-6 (0.4)** | 6716 (21.8) | 2430 (26.9) | 4282 (49.7) | -27 (1.1) |

## 6 CONCLUSION

**Limitations.** A drawback of our bound is that computing a gradient step is more expensive as it requires drawing samples from two encoding distributions. Similarly, learning aggregation functions is more computationally expensive compared to fixed schemes. Mixture-based bounds might be preferred if one is interested primarily in cross-modal reconstructions.

**Outlook.** Using modality-specific encoders to learn features and aggregating them with a PI function is clearly not the only choice for building multi-modal encoding distributions. However, it allows us to utilize modality-specific architectures for the encoding functions. Alternatively, our bounds could also be used, e.g., when multi-modal transformer architectures (Xu et al., 2022) encode a distribution on a shared latent space. Our approach applies to general prior densities if we can compute its cross-entropy relative to the multi-modal encoding distributions. An extension would be to apply it with more flexible prior distributions, e.g., as specified via score-based generative models (Vahdat et al., 2021). The ideas in this work might also be of interest for other approaches that require flexible modeling of conditional distributions, such as in meta-learning via Neural processes.

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

CONTENTS

## A    Multi-modal distribution matching

**Proposition 9** (Marginal and conditional distribution matching)**.** *For any $\mathcal{S} \in \mathcal{P}(\mathcal{M})$, we have*

$$\int p_d(x_\mathcal{S}) \mathcal{L}_\mathcal{S}(x_\mathcal{S}, \theta, \phi) \mathrm{d}x_\mathcal{S} + \mathcal{H}(p_d(x_\mathcal{S}))$$

$$= - \mathsf{KL}(q_\phi(z, x_\mathcal{S}) | p_\theta(z, x_\mathcal{S})) \tag{ZX$_\text{marginal}$}$$

$$= - \mathsf{KL}(p_d(x_\mathcal{S}) | p_\theta(x_\mathcal{S})) - \int p_d(x_\mathcal{S}) \mathsf{KL}(q_\phi(z|x_\mathcal{S}) | p_\theta(z|x_\mathcal{S})) \mathrm{d}x_\mathcal{S} \tag{X$_\text{marginal}$}$$

$$= - \mathsf{KL}(q_{\phi,\mathcal{S}}^{agg}(z) | p_\theta(z)) - \int q_{\phi,\mathcal{S}}^{agg}(z) \mathsf{KL}(q^\star(x_\mathcal{S}|z) | p_\theta(x_\mathcal{S}|z)) \mathrm{d}x_\mathcal{S}, \tag{Z$_\text{marginal}$}$$

*where $q_{\phi,\mathcal{S}}^{agg}(z) = \int p_d(x_\mathcal{S}) q_\phi(z|x_\mathcal{S}) \mathrm{d}x_\mathcal{S}$ is the aggregated prior (Makhzani et al., 2016) restricted on modalities from $\mathcal{S}$ and $q^\star(x_\mathcal{S}|z) = q_\phi(x_\mathcal{S}, z)/q_\phi^{agg}(z)$. Moreover, for fixed $x_\mathcal{S}$,*

$$\int p_d(x_{\backslash\mathcal{S}}|x_\mathcal{S}) \mathcal{L}_{\backslash\mathcal{S}}(x, \theta, \phi) \mathrm{d}x_{\backslash\mathcal{S}} + \mathcal{H}(p_d(x_{\backslash\mathcal{S}}|x_\mathcal{S}))$$

$$= - \mathsf{KL}\left(q_\phi(z|x) p_d(x_{\backslash\mathcal{S}}|x_\mathcal{S}) \big| p_\theta(x_{\backslash\mathcal{S}}|z) q_\phi(z|x_\mathcal{S})\right) \tag{ZX$_\text{conditional}$}$$

$$= - \mathsf{KL}(p_d(x_{\backslash\mathcal{S}}|x_\mathcal{S}) | p_\theta(x_{\backslash\mathcal{S}}|x_\mathcal{S})) \tag{X$_\text{conditional}$}$$

$$\quad - \int p_d(x_{\backslash\mathcal{S}}|x_\mathcal{S}) \left( \mathsf{KL}(q_\phi(z|x) | p_\theta(z|x)) + \int q_\phi(z|x) \log \frac{q_\phi(z|x_\mathcal{S})}{p_\theta(z|x_\mathcal{S})} \mathrm{d}z \right) \mathrm{d}x_{\backslash\mathcal{S}}$$

$$= - \mathsf{KL}(q_{\phi,\backslash\mathcal{S}}^{agg}(z|x_\mathcal{S}) | q_\phi(z|x_\mathcal{S})) - \int q_{\phi,\backslash\mathcal{S}}^{agg}(z|x_\mathcal{S}) \left( \mathsf{KL}(q^\star(x_{\backslash\mathcal{S}}|z, x_\mathcal{S}) | p_\theta(x_{\backslash\mathcal{S}}|z)) \right) \mathrm{d}z, \tag{Z$_\text{conditional}$}$$

*where $q_{\phi,\backslash\mathcal{S}}^{agg}(z|x_\mathcal{S}) = \int p_d(x_{\backslash\mathcal{S}}|x_\mathcal{S}) q_\phi(z|x) \mathrm{d}x_{\backslash\mathcal{S}}$ can be seen as an aggregated encoder conditioned on $x_\mathcal{S}$ and $q^\star(x_{\backslash\mathcal{S}}|z, x_\mathcal{S}) = q_\phi(z, x_{\backslash\mathcal{S}}|x_\mathcal{S})/q_{\phi,\backslash\mathcal{S}}^{agg}(z|x_\mathcal{S}) = p_d(x_{\backslash\mathcal{S}}|x_\mathcal{S}) q_\phi(z|x)/q_{\phi,\backslash\mathcal{S}}^{agg}(z|x_\mathcal{S})$.*

*Proof of Proposition 9.* The equations for $\mathcal{L}_\mathcal{S}(x_\mathcal{S})$ are well known for uni-modal VAEs, see for example Zhao et al. (2019). To derive similar representations for the conditional bound, note that the first equation (ZX$_\text{conditional}$) for matching the joint distribution of the latent and the missing modalities conditional on a modality subset follows from the definition of $\mathcal{L}_{\backslash\mathcal{S}}$,

$$\int p_d(x_{\backslash\mathcal{S}}|x_\mathcal{S}) \mathcal{L}_{\backslash\mathcal{S}}(x, \theta, \phi) \mathrm{d}x_{\backslash\mathcal{S}}$$

$$= \int p_d(x_{\backslash\mathcal{S}}|x_\mathcal{S}) \int q_\phi(z|x) \left[ \log p_\theta(x_{\backslash\mathcal{S}}|z) - \log q_\phi(z|x) + \log q_\phi(z|x_\mathcal{S})) \right] \mathrm{d}z \mathrm{d}x_{\backslash\mathcal{S}}$$

$$= \int p_d(x_{\backslash\mathcal{S}}|x_\mathcal{S}) \log p_d(x_{\backslash\mathcal{S}}|x_\mathcal{S}) \mathrm{d}x_{\backslash\mathcal{S}} + \int p_d(x_{\backslash\mathcal{S}}|x_\mathcal{S}) \int q_\phi(z|x) \left[ \log \frac{p_\theta(x_{\backslash\mathcal{S}}|z) q_\phi(z|x_\mathcal{S}))}{q_\phi(z|x) p_d(x_{\backslash\mathcal{S}}|x_\mathcal{S})} \right] \mathrm{d}z \mathrm{d}x_{\backslash\mathcal{S}}$$

$$= - \mathcal{H}(p_d(x_{\backslash\mathcal{S}}|x_\mathcal{S})) - \mathsf{KL}\left(q_\phi(z|x) p_d(x_{\backslash\mathcal{S}}|x_\mathcal{S}) \big| p_\theta(x_{\backslash\mathcal{S}}|z) q_\phi(z|x_\mathcal{S})\right).$$

To obtain the second representation (X$_\text{conditional}$) for matching the conditional distributions in the data space, observe that $p_\theta(x_{\backslash\mathcal{S}}|x_\mathcal{S}, z) = p_\theta(x_{\backslash\mathcal{S}}|z)$ and consequently,

$$- \int p_d(x_{\setminus \mathcal{S}}|x_\mathcal{S})\mathcal{L}_{\setminus \mathcal{S}}(x,\theta,\phi)\mathrm{d}x_{\setminus \mathcal{S}} - \mathcal{H}(p_d(x_{\setminus \mathcal{S}}|x_\mathcal{S}))$$

$$= \int p_d(x_{\setminus \mathcal{S}}|x_\mathcal{S})q_\phi(z|x) \log \frac{p_d(x_{\setminus \mathcal{S}}|x_\mathcal{S})q_\phi(z|x)}{p_\theta(x_{\setminus \mathcal{S}}|z)q_\phi(z|x_\mathcal{S})}\mathrm{d}z\mathrm{d}x_{\setminus \mathcal{S}}$$

$$= \int p_d(x_{\setminus \mathcal{S}}|x_\mathcal{S})q_\phi(z|x) \log \frac{p_d(x_{\setminus \mathcal{S}}|x_\mathcal{S})q_\phi(z|x)p_\theta(z|x_\mathcal{S})}{p_\theta(x_{\setminus \mathcal{S}}|z)p_\theta(z|x_\mathcal{S})q_\phi(z|x_\mathcal{S})}\mathrm{d}z\mathrm{d}x_{\setminus \mathcal{S}}$$

$$= \int p_d(x_{\setminus \mathcal{S}}|x_\mathcal{S})q_\phi(z|x) \log \frac{p_d(x_{\setminus \mathcal{S}}|x_\mathcal{S})q_\phi(z|x)p_\theta(z|x_\mathcal{S})}{p_\theta(x_{\setminus \mathcal{S}}|z,x_\mathcal{S})p_\theta(z|x_\mathcal{S})q_\phi(z|x_\mathcal{S})}\mathrm{d}z\mathrm{d}x_{\setminus \mathcal{S}}$$

$$= \int p_d(x_{\setminus \mathcal{S}}|x_\mathcal{S})q_\phi(z|x) \log \frac{p_d(x_{\setminus \mathcal{S}}|x_\mathcal{S})q_\phi(z|x)p_\theta(z|x_\mathcal{S})}{p_\theta(x_{\setminus \mathcal{S}}|x_\mathcal{S})p_\theta(z|x_\mathcal{S},x_{\setminus \mathcal{S}})q_\phi(z|x_\mathcal{S})}\mathrm{d}z\mathrm{d}x_{\setminus \mathcal{S}}$$

$$= \mathsf{KL}(p_d(x_{\setminus \mathcal{S}}|x_\mathcal{S})|p_\theta(x_{\setminus \mathcal{S}}|x_\mathcal{S})) + \int p_d(x_{\setminus \mathcal{S}}|x_\mathcal{S}) \int q_\phi(z|x) \left[ \log \frac{q_\phi(z|x)}{p_\theta(z|x)} + \log \frac{p_\theta(z|x_\mathcal{S})}{q_\phi(z|x_\mathcal{S})} \right] \mathrm{d}z\mathrm{d}x_{\setminus \mathcal{S}}.$$

Lastly, the representation ($\mathsf{Z_{conditional}}$) for matching the distributions in the latent space given a modality subset follows by recalling that

$$p_d(x_{\setminus \mathcal{S}}|x_\mathcal{S})q_\phi(z|x) = q_{\phi,\setminus \mathcal{S}}^{\mathrm{agg}}(z|x_\mathcal{S})q^\star(x_{\setminus \mathcal{S}}|z,x_\mathcal{S})$$

and consequently,

$$- \int p_d(x_{\setminus \mathcal{S}}|x_\mathcal{S})\mathcal{L}_{\setminus \mathcal{S}}(x,\theta,\phi)\mathrm{d}x_{\setminus \mathcal{S}} - \mathcal{H}(p_d(x_{\setminus \mathcal{S}}|x_\mathcal{S}))$$

$$= \int p_d(x_{\setminus \mathcal{S}}|x_\mathcal{S})q_\phi(z|x) \log \frac{p_d(x_{\setminus \mathcal{S}}|x_\mathcal{S})q_\phi(z|x)}{p_\theta(x_{\setminus \mathcal{S}}|z)q_\phi(z|x_\mathcal{S})}\mathrm{d}z\mathrm{d}x_{\setminus \mathcal{S}}$$

$$= \int q_{\phi,\setminus \mathcal{S}}^{\mathrm{agg}}(z|x_\mathcal{S})q^\star(x_{\setminus \mathcal{S}}|z,x_\mathcal{S}) \log \frac{q_{\phi,\setminus \mathcal{S}}^{\mathrm{agg}}(z|x_\mathcal{S})q^\star(x_{\setminus \mathcal{S}}|z,x_\mathcal{S})}{p_\theta(x_{\setminus \mathcal{S}}|z)q_\phi(z|x_\mathcal{S})}\mathrm{d}z\mathrm{d}x_{\setminus \mathcal{S}}$$

$$= \mathsf{KL}(q_{\phi,\setminus \mathcal{S}}^{\mathrm{agg}}(z|x_\mathcal{S})|q_\phi(z|x_\mathcal{S})) - \int q_{\phi,\setminus \mathcal{S}}^{\mathrm{agg}}(z|x_\mathcal{S}) \left( \mathsf{KL}(q^\star(x_{\setminus \mathcal{S}}|z,x_\mathcal{S})|p_\theta(x_{\setminus \mathcal{S}}|z)) \right) \mathrm{d}z.$$

$\square$

**Remark 10** (Prior-hole problem and Bayes or conditional consistency)**.** In the uni-modal setting, the mismatch between the prior and the aggregated prior can be large and can lead to poor unconditional generative performance, because this would lead to high-probability regions under the prior that have not been trained due to their small mass under the aggregated prior (Hoffman and Johnson, 2016; Rosca et al., 2018). Equation ($\mathsf{Z_{marginal}}$) extents this to the multi-modal case and we expect that unconditional generation can be poor if this mismatch is large. Moreover, ($\mathsf{Z_{conditional}}$) extends this conditioned on some modality subset and we expect that cross-generation for $x_{\setminus \mathcal{S}}$ conditional on $x_\mathcal{S}$ can be poor if the mismatch between $q_{\phi,\setminus \mathcal{S}}^{\mathrm{agg}}(z|x_\mathcal{S})$ and $q_\phi(z|x_\mathcal{S})$ is large for $x_\mathcal{S} \sim p_d$, because high-probability regions under $q_\phi(z|x_\mathcal{S})$ will not have been trained - via optimizing $\mathcal{L}_{\setminus \mathcal{S}}(x)$ - to model $x_{\setminus \mathcal{S}}$ conditional on $x_\mathcal{S}$, due to their small mass under $q_{\phi,\setminus \mathcal{S}}^{\mathrm{agg}}(z|x_\mathcal{S})$. The mismatch will vanish when the encoders are consistent and correspond to a single Bayesian model where they approximate the true posterior distributions.

**Corollary 11** (Multi-modal log-likelihood approximation)**.** *For any modality mask $\mathcal{S}$, we have*

$$\int p_d(x) \left[ \mathcal{L}_\mathcal{S}(x_\mathcal{S},\theta,\phi,1) + \mathcal{L}_{\setminus \mathcal{S}}(x,\theta,\phi,1) \right] \mathrm{d}x - \int p_d(x) \left[ \log p_\theta(x) \right] \mathrm{d}x$$

$$= - \int p_d(x_\mathcal{S}) \left[ \mathsf{KL}(q_\phi(z|x_\mathcal{S})|p_\theta(z|x_\mathcal{S})) \right] \mathrm{d}x - \int p_d(x) \left[ \mathsf{KL}(q_\phi(z|x)|p_\theta(z|x)) \right] \mathrm{d}x$$

$$+ \int p_d(x)q_\phi(z|x) \left[ \log \frac{q_\phi(z|x_\mathcal{S})}{p_\theta(z|x_\mathcal{S})} \right] \mathrm{d}z\mathrm{d}x.$$

*Proof.* This follows from ($\mathsf{X}_{\text{marginal}}$) and ($\mathsf{X}_{\text{conditional}}$). $\square$

**Remark 12.** Corollary 11 shows that the variational bound can become tight in the limiting case where the encoding distributions approximates the true posterior distributions. A similar result does not hold for the mixture-based multi-modal bound. Indeed, as shown in Daunhawer et al. (2022), there is a gap between the variational bound and the log-likelihood given by the conditional entropies that cannot be reduced even for flexible encoding distributions. More precisely, it holds that

$$\int p_d(x) \log p_\theta(x) \mathrm{d}x \geq \int p_d(x) \mathcal{L}^{\text{Mix}}(x, \theta, \phi, 1) \mathrm{d}x + \mathcal{H}(p_d(X_{\backslash\mathcal{S}}|X_\mathcal{S})).$$

Moreover, our bound can be tight for an arbitrary number of modalities in the limiting case of infinite-capacity encoders. In contrast, Daunhawer et al. (2022) show that for mixture-based bounds, this variational gap increases with each additional modality, if the new modality is 'sufficiently diverse', even for infinite-capacity encoders. Note that in practice, we optimize over the objective where the mask $\mathcal{S}$ is not fixed but random, which induces a Jensen gap as in other any-order methods (Hoogeboom et al., 2021; Shih et al., 2022).

**Remark 13.** The term $\int p_d(x) q_\phi(z|x) \left[\log \frac{q_\phi(z|x_\mathcal{S})}{p_\theta(z|x_\mathcal{S})}\right] \mathrm{d}z\mathrm{d}x$ arising in Corollary 11 and in ($\mathsf{X}_{\text{conditional}}$) is not necessarily negative. Analogous to other variational approach for learning conditional distributions such as latent Neural processes, our bound becomes an approximately lower bound. Note that $\mathcal{L}_\mathcal{S}$ is maximized when $q_\phi(z|x_\mathcal{S}) = p_\theta(z|x_\mathcal{S})$, see ($\mathsf{X}_{\text{marginal}}$), while $\mathcal{L}_{\backslash\mathcal{S}}$ is maximized when $q_\phi(z|x_\mathcal{S}) = \int p_d(x_{\backslash\mathcal{S}}|x_\mathcal{S}) q_\phi(z|x) \mathrm{d}x_{\backslash\mathcal{S}} = q^{\text{agg}}_{\phi,\backslash\mathcal{S}}(z|x_\mathcal{S})$, see ($\mathsf{Z}_{\text{conditional}}$). The latter condition implies a lower bound in Corollary 11 of

$$\int p_d(x) \left[\mathcal{L}_\mathcal{S}(x_\mathcal{S}, \theta, \phi, 1) + \mathcal{L}_{\backslash\mathcal{S}}(x, \theta, \phi, 1)\right] \mathrm{d}x = \int p_d(x) \left[\log p_\theta(x) - \mathsf{KL}(q_\phi(z|x)|p_\theta(z|x))\right] \mathrm{d}x.$$

**Remark 14** (Optimization, multi-task learning and the choice of $\rho$)**.** For simplicity, we have chosen to sample $\mathcal{S} \sim \rho$ in our experiments via the hierarchical construction $\gamma \sim \mathcal{U}(0,1)$, $m_j \sim \text{Bern}(\gamma)$ iid for all $j \in [M]$ and setting $\mathcal{S} = \{s \in [M]: m_j = 1\}$. The distribution $\rho$ for masking the modalities can be adjusted to accommodate various weights for different modality subsets. Indeed, (2) can be seen as a linear scalarisation of a multi-task learning problem (Fliege and Svaiter, 2000; Sener and Koltun, 2018). We aim to optimise a loss vector $(\mathcal{L}_\mathcal{S} + \mathcal{L}_{\backslash\mathcal{S}})_{\mathcal{S}\subset\mathcal{M}}$, where the gradients for each $\mathcal{S} \subset \mathcal{M}$ can point in different directions, making it challenging to minimise the loss for all modalities simultaneously. Consequently, Javaloy et al. (2022) used multi-task learning techniques (e.g., as suggested in Chen et al. (2018); Yu et al. (2020)) for adjusting the gradients in mixture based VAEs. Such improved optimisation routines are orthogonal to our approach. Similarly, we do not analyse optimisation issues such as initialisations and training dynamics that have been found challenging for multi-modal learning (Wang et al., 2020; Huang et al., 2022).

## B  META-LEARNING AND NEURAL PROCESSES

**Meta-learning.** We consider a standard meta-learning setup but use slightly non-standard notations to remain consistent with notations used in other parts of this work. We consider a compact input or covariate space $\mathcal{A}$ and output space $\mathcal{X}$. Let $\mathcal{D} = \cup_{M=1}^\infty (\mathcal{A} \times \mathcal{X})^M$ be the collection of all input-output pairs. In meta-learning, we are given a meta-dataset, i.e., a collection of elements from $\mathcal{D}$. Each individual data set $D = (a, x) = D_c \cup D_t \in \mathcal{D}$ is called a task and split into a context set $D_c = (a_c, x_c)$, and target set $D_t = (a_t, x_t)$. We aim to predict the target set from the context set. Consider, therefore, the prediction map

$$\pi: D_c = (a_c, x_c) \mapsto p(x_t|a_t, D_c) = p(x_t, x_c|a_t, a_c)/p(x_c|a_c),$$

mapping each context data set to the predictive stochastic process conditioned on $D_c$.

**Variational lower bounds for Neural processes.** Latent Neural processes (Garnelo et al., 2018b; Foong et al., 2020) approximate this prediction map by using a latent variable model with parameters $\theta$ in the form of

$$z \sim p_\theta, \ p_\theta(x_t|a_t, z) = \prod_{(a,x)\in D_t} p_\epsilon(x - f_\theta(a, z))$$

for a prior $p_\theta$, decoder $f_\theta$ and a parameter free density $p_\epsilon$. The model is then trained by (approximately) maximizing a lower bound on $\log p_\theta(x_t|a_t, a_c, x_c)$. Note that for an encoding density $q_\phi$, we have that

$$\log p_\theta(x_t|a_t, a_c, x_x) = \int q_\phi(z|x, a) \log p_\theta(x_t|a_t, z)\mathrm{d}z - \mathsf{KL}(q_\phi(z|a, x)|p_\theta(z|a_c, x_c)).$$

Since the posterior distribution $p_\theta(z|a_c, x_c)$ is generally intractable, one instead replaces it with a variational approximation or learned conditional prior $q_\phi(z|a_c, x_c)$, and optimizes the following objective

$$\mathcal{L}^{\mathrm{LNP}}_{\backslash\mathcal{C}}(x, a) = \int q_\phi(z|x, a) \log p_\theta(x_t|a_t, z)\mathrm{d}z - \mathsf{KL}(q_\phi(z|a, x)|q_\phi(z|a_c, x_c)).$$

Note that this objective coincides with $\mathcal{L}_{\backslash\mathcal{C}}$ conditioned on the covariate values $a$ and where $\mathcal{C}$ comprises the indices of the data points that are part of the context set.

Using this variational lower bound can yield subpar performance compared to other biased log-likelihood objectives (Kim et al., 2018; Foong et al., 2020), possibly because the variational approximation $q_\phi(z|a_c, x_c)$ needs not to be close the posterior distribution $p_\theta(z|a_c, x_c)$. It would therefore be interesting to analyze in future work if one can alleviate such issues if one optimizes additionally the variational objective corresponding to $\mathcal{L}_\mathcal{C}$, i.e.,

$$\mathcal{L}^{\mathrm{LNP}}_{\mathcal{C}}(x_c, a_c) = \int q_\phi(z|x_c, a_c) \log p_\theta(x_c|a_c, z)\mathrm{d}z - \mathsf{KL}(q_\phi(z|a_c, x_c)|p_\theta(z)),$$

as we do in this work for multi-modal generative models. Note that the objective $\mathcal{L}^{\mathrm{LNP}}_{\mathcal{C}}$ alone can be seen as a form of a neural statistician model (Edwards and Storkey, 2016) where $\mathcal{C}$ coincides with the indices of the target set, while a form of the mixture-based bound corresponds to a neural process bound similar to variational homoencoders (Hewitt et al., 2018), see also the discussion in Le et al. (2018).

## C  Information-theoretic perspective

We recall first that the mutual information on the inference path[5] is given by

$$\mathrm{I}_{q_\phi}(X_\mathcal{S}, Z_\mathcal{S}) = \int q_\phi(x_\mathcal{S}, z) \log \frac{q_\phi(x_\mathcal{S}, z)}{p_d(x_\mathcal{S})q^{\mathrm{agg}}_{\phi,\mathcal{S}}(z)}\mathrm{d}z\mathrm{d}x_\mathcal{S},$$

where $q^{\mathrm{agg}}_{\phi,\mathcal{S}}(z) = \int p_d(x_\mathcal{S})q_\phi(z|x_\mathcal{S})\mathrm{d}x_\mathcal{S}$ is the aggregated prior (Makhzani et al., 2016). It can be bounded by standard (Barber and Agakov, 2004; Alemi et al., 2016; 2018) lower and upper bounds using the rate and distortion:

$$\mathcal{H}_\mathcal{S} - D_\mathcal{S} \leq \mathcal{H}_\mathcal{S} - D_\mathcal{S} + \Delta_1 = \mathrm{I}_{q_\phi}(X_\mathcal{S}, Z_\mathcal{S}) = R_\mathcal{S} - \Delta_2 \leq R_\mathcal{S},$$

with $\Delta_1 = \int q^{\mathrm{agg}}_\phi(z)\mathsf{KL}(q^\star(x_\mathcal{S}|z)|p_\theta(x_\mathcal{S}|z))\mathrm{d}z > 0$, $\Delta_2 = \mathsf{KL}(q^{\mathrm{agg}}_{\phi,\mathcal{S}}(z)|p_\theta(z)) > 0$ and $q^\star(x_\mathcal{S}|z) = q_\phi(x_\mathcal{S}, z)/q^{\mathrm{agg}}_\phi(z)$.

Moreover, if the bounds in (5) become tight with $\Delta_1 = \Delta_2 = 0$ in the hypothetical scenario of infinite-capacity decoders and encoders, one obtains $\int p_d\mathcal{L}_\mathcal{S} = (1 - \beta)\mathrm{I}_{q_\phi}(X_\mathcal{S}, Z_\mathcal{S}) + \mathcal{H}_\mathcal{S}$. For $\beta > 1$, maximizing $\mathcal{L}_\mathcal{S}$ yields an auto-decoding limit that minimizes $\mathrm{I}_{q_\phi}(x_\mathcal{S}, z)$ for which the latent representations do not encode any information about the data, whilst $\beta < 1$ yields an auto-encoding limit that maximizes $\mathrm{I}_{q_\phi}(X_\mathcal{S}, Z)$ and for which the data is perfectly encoded and decoded.

To arrive at a similar interpretation for the conditional bound $\mathcal{L}_{\backslash\mathcal{S}}$, recall that we have defined $R_{\backslash\mathcal{S}} = \int p_d(x)\mathsf{KL}(q_\phi(z|x)|q_\phi(z|x_\mathcal{S}))\mathrm{d}x$ for a conditional or cross rate term and $D_{\backslash\mathcal{S}} = -\int p_d(x)q_\phi(z|x) \log p_\theta(x_{\backslash\mathcal{S}}|z)\mathrm{d}z\mathrm{d}x$ for the distortion term. Bounds on the conditional mutual information

$$\mathrm{I}_{q_\phi}(X_{\backslash\mathcal{S}}, Z_\mathcal{M}|X_\mathcal{S}) = \int p_d(x_\mathcal{S})\mathsf{KL}(p_d(x_{\backslash\mathcal{S}}, z|x_\mathcal{S}))|p_d(x_{\backslash\mathcal{S}}|x_\mathcal{S})q^{\mathrm{agg}}_{\phi,\backslash\mathcal{S}}(z|x_\mathcal{S}))\mathrm{d}x_\mathcal{S}$$

with $q^{\mathrm{agg}}_{\phi,\backslash\mathcal{S}}(z|x_\mathcal{S}) = \int p_d(x_{\backslash\mathcal{S}}|x_\mathcal{S})q_\phi(z|x)\mathrm{d}x_{\backslash\mathcal{S}}$ can be established as follows.

---

[5] We include the conditioning modalities as an index for the latent variable $Z$ when the condtitioning set is unclear.

*Proof of Lemma 2.* The proof follows by adapting the arguments in Alemi et al. (2018). The law of $X_{\backslash \mathcal{S}}$ and $Z$ conditional on $X_{\mathcal{S}}$ on the encoder path can be written as

$$q_\phi(z, x_{\backslash \mathcal{S}}|x_{\mathcal{S}}) = p_d(x_{\backslash \mathcal{S}}|x_{\mathcal{S}})q_\phi(z|x) = q_{\phi,\backslash \mathcal{S}}^{\mathrm{agg}}(z|x_{\mathcal{S}})q^\star(x_{\backslash \mathcal{S}}|z, x_{\mathcal{S}})$$

with $q^\star(x_{\backslash \mathcal{S}}|z, x_{\mathcal{S}}) = q_\phi(z, x_{\backslash \mathcal{S}}|x_{\mathcal{S}})/q_{\phi,\backslash \mathcal{S}}^{\mathrm{agg}}(z|x_{\mathcal{S}})$. To prove a lower bound on the conditional mutual information, note that

$$\mathrm{I}_{q_\phi}(X_{\backslash \mathcal{S}}, Z_{\mathcal{M}}|X_{\mathcal{S}})$$

$$= \int p_d(x_{\mathcal{S}}) \int q_{\phi,\backslash \mathcal{S}}^{\mathrm{agg}}(z|x_{\mathcal{S}}) \int q^\star(x_{\backslash \mathcal{S}}|z, x_{\mathcal{S}}) \log \frac{q_{\phi,\backslash \mathcal{S}}^{\mathrm{agg}}(z|x_{\mathcal{S}})q^\star(x_{\backslash \mathcal{S}}|z, x_{\mathcal{S}})}{q_{\phi,\backslash \mathcal{S}}^{\mathrm{agg}}(z|x_{\mathcal{S}})p_d(x_{\backslash \mathcal{S}}|x_{\backslash \mathcal{S}})} \mathrm{d}z \mathrm{d}x_{\backslash \mathcal{S}} \mathrm{d}x_{\mathcal{S}}$$

$$= \int p_d(x_{\mathcal{S}}) \int q_{\phi,\backslash \mathcal{S}}^{\mathrm{agg}}(z|x_{\mathcal{S}}) \left[ q^\star(x_{\backslash \mathcal{S}}|z, x_{\mathcal{S}}) \log p_\theta(x_{\backslash \mathcal{S}}|z)) + \mathsf{KL}(q^\star(x_{\backslash \mathcal{S}}|z, x_{\mathcal{S}})|p_\theta(x_{\backslash \mathcal{S}}|z)) \right] \mathrm{d}z \mathrm{d}x_{\mathcal{S}}$$

$$- \int p_d(x_{\mathcal{S}}) \int p_d(x_{\backslash \mathcal{S}}|x_{\mathcal{S}}) \log p_d(x_{\backslash \mathcal{S}}|x_{\mathcal{S}}) \mathrm{d}x$$

$$= \int p_d(x) \int q_\phi(z|x) \log p_\theta(x_{\backslash \mathcal{S}}|z) \mathrm{d}z \mathrm{d}x - \underbrace{\int p_d(x_{\mathcal{S}}) \int p_d(x_{\backslash \mathcal{S}}|x_{\mathcal{S}}) \log p_d(x_{\backslash \mathcal{S}}|x_{\mathcal{S}}) \mathrm{d}x}_{=-\mathcal{H}_{\backslash \mathcal{S}}=-\mathcal{H}(X_{\backslash \mathcal{S}}|X_{\mathcal{S}})}$$

$$+ \underbrace{\int p_d(x_{\mathcal{S}}) \int q_{\phi,\backslash \mathcal{S}}^{\mathrm{agg}}(z|x_{\mathcal{S}})\mathsf{KL}(q^\star(x_{\backslash \mathcal{S}}|z, x_{\mathcal{S}})|p_\theta(x_{\backslash \mathcal{S}}|z)) \mathrm{d}x_{\mathcal{S}}}_{=\Delta_{\backslash \mathcal{S},1}\geq 0}$$

$$=\Delta_{\backslash \mathcal{S},1} + D_{\backslash \mathcal{S}} + \mathcal{H}_{\backslash \mathcal{S}}.$$

The upper bound follows by observing that

$$\mathrm{I}_{q_\phi}(X_{\backslash \mathcal{S}}, Z_{\mathcal{M}}|X_{\mathcal{S}})$$

$$= \int p_d(x_{\mathcal{S}}) \int p_d(x_{\backslash \mathcal{S}}|x_{\backslash}) \log \frac{q_\phi(z|x)p_d(x_{\backslash \mathcal{S}}|x_{\mathcal{S}})}{q_{\phi,\backslash \mathcal{S}}^{\mathrm{agg}}(z|x_{\mathcal{S}})p_d(x_{\backslash \mathcal{S}}|x_{\mathcal{S}})} \mathrm{d}z \mathrm{d}x$$

$$= \int p_d(x)\mathsf{KL}(q_\phi(z|x)|q_\phi(z|x_{\mathcal{S}})) \mathrm{d}x - \underbrace{\int p_d(x_{\mathcal{S}})\mathsf{KL}(q_{\phi,\backslash \mathcal{S}}^{\mathrm{agg}}(z|x_{\mathcal{S}})|q_\phi(z|x_{\mathcal{S}})) \mathrm{d}x_{\mathcal{S}}}_{=\Delta_{\backslash \mathcal{S},2}\geq 0}$$

$$=R_{\backslash \mathcal{S}} - \Delta_{\backslash \mathcal{S},2}.$$

$\square$

**Remark 15** (Total correlation based objectives)**.** The objective suggested in Hwang et al. (2021) is motivated by a conditional variational bottleneck perspective that aims to maximize the reduction of total correlation of $X$ when conditioned on $Z$, as measured by the conditional total correlation, see Watanabe (1960); Ver Steeg and Galstyan (2015); Gao et al. (2019), i.e.,

$$\text{minimizing } \left\{ \mathrm{TC}(X|Z) = \mathrm{TC}(X) - \mathrm{TC}(X, Z) = \mathrm{TC}(X) + \mathrm{I}_{q_\phi}(X, Z) - \sum_{s=1}^{M} \mathrm{I}_{q_\phi}(X_s, Z) \right\}, \quad (7)$$

where $\mathrm{TC}(X) = \mathsf{KL}(p(x)|\prod_{i=1}^{d} p(x_i))$ for $d$-dimensional $X$. Resorting to variational lower bounds and using a constant $\beta > 0$ that weights the contributions of the mutual information terms, approximations of (7) can be optimized by maximizing

$$\mathcal{L}^{\mathrm{TC}}(\theta, \phi, \beta) = \int \rho(\mathcal{S}) \int \left\{ q_\phi(z|x) \left[ \log p_\theta(x|z) \right] \mathrm{d}z - \beta \mathsf{KL}(q_\phi(z|x)|q_\phi(z|x_{\mathcal{S}})) \right\} \mathrm{d}\mathcal{S},$$

where $\rho$ is concentrated on the uni-modal subsets of $\mathcal{M}$.

# D  OPTIMAL VARIATIONAL DISTRIBUTIONS

The optimal variational density for the mixture-based (1) multi-modal objective,

$$\int p_d(\mathrm{d}x)\mathcal{L}_{\mathcal{S}}^{\mathrm{Mix}}(x) = \int p_d(x_{\mathcal{S}}) \int q_\phi(z|x_{\mathcal{S}}) \int p_d(x_{\backslash \mathcal{S}}|x_{\mathcal{S}})$$

$$\left[ \log p_\theta(x_{\mathcal{S}}|z) + \log p_\theta(x_{\backslash \mathcal{S}}|z) - \beta \log p_\theta(z) - \beta \log q_\phi(z|x_{\mathcal{S}}) \right] \mathrm{d}x_{\backslash \mathcal{S}} \mathrm{d}z \mathrm{d}x_{\mathcal{S}}$$

is attained at

$$q^\star(z|x_\mathcal{S}) \propto \exp\left(\frac{1}{\beta}\int p_d(x_{\backslash\mathcal{S}}|x_\mathcal{S})\left[\log p_\theta(x_\mathcal{S}|z) + \log p_\theta(x_{\backslash\mathcal{S}}|z) - \beta\log p_\theta(z)\right]\mathrm{d}x_{\backslash\mathcal{S}}\right)$$

$$\propto \tilde{p}_{\beta,\theta}(z|x_\mathcal{S})\exp\left(\int p_d(x_{\backslash\mathcal{S}}|x_\mathcal{S})\log\tilde{p}_{\beta,\theta}(x_{\backslash\mathcal{S}}|z)\mathrm{d}x_{\backslash\mathcal{S}}\right).$$

## E  PERMUTATION-INVARIANT ARCHITECTURES

**Multi-head attention and masking.**  We introduce here a standard multi-head attention (Bahdanau et al., 2014; Vaswani et al., 2017) mapping $\mathrm{MHA}_\vartheta\colon \mathbb{R}^{I\times D_X}\times\mathbb{R}^{S\times D_Y}\to\mathbb{R}^{I\times D_Y}$ given by

$$\mathrm{MHA}_\vartheta(X,Y) = W^O\left[\mathrm{Head}^1(X,Y,Y),\ldots,\mathrm{Head}^H(X,Y,Y)\right],\quad \vartheta = (W_Q,W_K,W_V,W_O),$$

with output matrix $W_O\in\mathbb{R}^{D_A\times D_Y}$, projection matrices $W_Q,W_K,W_V\in\mathbb{R}^{D_Y\times D_A}$ and

$$\mathrm{Head}^h(Q,K,V) = \mathrm{Att}(QW_Q^h, KW_K^h, VW_V^h)\in\mathbb{R}^{I\times D}\tag{8}$$

where we assume that $D = D_A/H\in\mathbb{N}$ is the head size. Here, the dot-product attention function is

$$\mathrm{Att}(Q,K,V) = \sigma(QK^\top)V,$$

where $\sigma$ is the softmax function applied to each column of $Q$ and $K^\top$, respectively.

**Masked multi-head attention.**  In practice, it is convenient to consider masked multi-head attention models $\mathrm{MMHA}_{\vartheta,M}\colon \mathbb{R}^{I\times D_X}\times\mathbb{R}^{T\times D_Y}\to\mathbb{R}^{I\times D_Y}$ for mask matrix $M\in\{0,1\}^{I\times T}$ that operate on key or value sequences of fixed length $T$ where the $h$-th head (8) is given by

$$\mathrm{Head}^h(Q,K,V) = \left[M\odot\sigma(QW_Q^h(KW_K^h)^\top)\right]V_{t'}W_V^h\in\mathbb{R}^{T\times D}.$$

Using the softmax kernel function $\mathrm{SM}_D(q,k) = \exp(q^\top k/\sqrt{D})$, we set

$$\mathrm{MMHA}_{\vartheta,M}(X,Y)_i = \sum_{t=1}^T\sum_{h=1}^H\frac{M_{it}\mathrm{SM}_D(W_h^QX_i, W_h^KY_t)}{\sum_{t'=1}^T M_{it'}\mathrm{SM}_D(X_iW_h^Q, Y_{t'}W_h^K)}Y_tW_h^VW_h^O\tag{9}$$

which does not depend on $Y_t$ if $M_{\cdot t} = 0$.

**Masked self-attention.**  For mask matrix $M = mm^\top$ with $m = (1_{\{s\in\mathcal{S}\}})_{s\in\mathcal{M}}$, we write

$$\mathrm{MHA}_\vartheta(Y_\mathcal{S}, Y_\mathcal{S}) = \mathrm{MMHA}_{\vartheta,M}(\mathfrak{i}(Y_\mathcal{S}), \mathfrak{i}(Y_\mathcal{S}))_\mathcal{S}.$$

where $\mathrm{MMHA}_{\vartheta,M}$ operates on sequences with fixed length and $\mathfrak{i}(Y_\mathcal{S}))_t = Y_t$ if $t\in\mathcal{S}$ and 0 otherwise.

**LayerNorm and SetNorm.**  Let $h\in\mathbb{R}^{T\times D}$ and consider the normalisation

$$\mathrm{N}(h) = \frac{h-\mu(h)}{\sigma(h)}\odot\gamma + \beta$$

where $\mu$ and $\sigma$ standardise the input $h$ by computing the mean, and the variance, respectively, over some axis of $h$, whilst $\gamma$ and $\beta$ define a transformation. LayerNorm (Ba et al., 2016) standardises inputs over the last axis, e.g., $\mu(h) = \frac{1}{D}\sum_{d=1}^D\mu_{\cdot,d}$, i.e., separately for each element. In contrast, Set-Norm (Zhang et al., 2022b) standardises inputs over both axes, e.g., $\mu(h) = \frac{1}{TD}\sum_{t=1}^T\sum_{d=1}^D\mu_{t,d}$, thereby losing the global mean and variance only. In both cases, $\gamma$ and $\beta$ share their values across the first axis. Both normalisations are permutation-equivariant.

**Transformer.**  We consider a masked pre-layer-norm (Wang et al., 2019; Xiong et al., 2020) multi-head transformer block

$$(\mathrm{MMTB}_{\vartheta,M}(\mathfrak{i}_\mathcal{S}(Y_\mathcal{S})))_\mathcal{S} = (Z + \sigma_{\mathrm{ReLU}}(\mathrm{LN}(Z)))_\mathcal{S}$$

with $\sigma_{\mathrm{ReLU}}$ being a ReLU non-linearity and

$$Z = \mathfrak{i}_\mathcal{S}(Y_\mathcal{S}) + \mathrm{MMHA}_{\vartheta,M}(\mathrm{LN}(\mathfrak{i}_\mathcal{S}(Y_\mathcal{S})), \mathrm{LN}(\mathfrak{i}_\mathcal{S}(Y_\mathcal{S})))$$

where $M = mm^\top$ for $m = (1_{\{s\in\mathcal{S}\}})_{s\in\mathcal{M}}$.

**Set-Attention Encoders.** Set $g^0 = \mathfrak{i}_{\mathcal{S}}(\chi_\vartheta(h_{\mathcal{S}}))$ and for $k \in \{1, \dots, L\}$, let $g^k = \mathrm{MMTB}_{\vartheta, M}(g_{\mathcal{S}}^{k-1})$. Then, we can express the self-attention multi-modal aggregation mapping via $f_\vartheta(h_{\mathcal{S}}) = \rho_\vartheta\left(\sum_{s \in \mathcal{S}} g_s^L\right)$.

**Remark 16** (Mixture-of-Product-of-Experts or MoPoEs)**.** Sutter et al. (2021) introduced a MoPoE aggregation scheme that extends MoE or PoE schemes by considering a mixture distribution of all $2^M$ modality subsets, where each mixture component consists of a PoE model, i.e.,

$$q_\phi^{\mathrm{MoPoE}}(z|x_{\mathcal{M}}) = \frac{1}{2^M} \sum_{x_{\mathcal{S}} \in \mathcal{P}(x_{\mathcal{M}})} q_\phi^{\mathrm{PoE}}(z|x_{\mathcal{S}}).$$

This can also be seen as another PI model. While it does not require learning separate encoding models for all modality subsets, it however becomes computationally expensive to evaluation for large $M$. Our mixture models using components with a SumPooling or SelfAttention aggregation can be seen as an alternative that allows one to choose the number of mixture components $K$ to be smaller than $2^M$, with non-uniform weights, while the individual mixture components are not constrained to have a PoE form.

**Remark 17** (Multi-modal time series models)**.** We have introduced our generative model in a general form that also applies to the time-series setup, such as when a latent Markov process drives multiple time series. For example, consider a latent Markov process $Z = (Z_t)_{t \in \mathbb{N}}$ with prior dynamics $p_\theta(z_1, \dots, z_T) = p_\theta(z_1) \prod_{t=2}^T p_\theta(z_t|z_{t-1})$ for an initial density $p_\theta(z_1)$ and homogeneous Markov kernels $p_\theta(z_t|z_{t-1})$. Conditional on $Z$, suppose that the time-series $(X_{s,t})_{t \in \mathbb{N}}$ follows the dynamics $p_\theta(x_{s,1}, \dots, x_{s,T}|z_1, \dots, z_T) = \prod_{t=2}^T p_\theta(x_{s,t}|z_t)$ for decoding densities $p_\theta(x_{s,t}|z_t)$. A common choice (Chung et al., 2015) for modeling the encoding distribution for such sequential (uni-modal) VAEs is to assume the factorisation $q_\phi(z_1, \dots z_T|x_1, \dots x_T) = q_\phi(z_1|x_1) \prod_{t=2}^T q_\phi(z_t|z_{t-1}, x_t)$ for $x_t = (x_{s,t})_{s \in \mathcal{M}}$, with initial encoding densities $q_\phi(z_1|x_1)$ and encoding Markov kernels $q_\phi(z_t|z_{t-1}, x_t)$. One can again consider modality-specific encodings $h_s = (h_{s,1}, \dots, h_{s,T})$, $h_{s,t} = h_{s,\varphi}(x_{s,t})$, now applied separately at each time step that are then used to construct Markov kernels that are permutation-invariant in the form of $q'_\phi(z_t|z_{t-1}, \pi h_\varphi(x_{t,\mathcal{S}})) = q'_\phi(z_t|z_{t-1}, h_\varphi(x_{t,\mathcal{S}}))$ for permutations $\pi \in \mathbb{S}_{\mathcal{S}}$. Alternatively, in absence of the auto-regressive encoding structure with Markov kernels, one could also use transformer models that use absolute or relative positional embeddings across the last temporal axis, but no positional embeddings across the first modality axis, followed by a sum-pooling operation across the modality axis. Note that previous works using multi-modal time series such as Kramer et al. (2022) use a non-amortized encoding distribution for the full multi-modal posterior only. A numerical evaluation of permutation-invariant schemes for time series models is however outside the scope of this work.

## F  PERMUTATION-EQUIVARIANCE AND PRIVATE LATENT VARIABLES

In principle, the general permutation invariant aggregation schemes that have been introduced could also be used for learning multi-modal models with private latent variables. For example, suppose that the generative model factorises as

$$p_\theta(z, x) = p(z) \prod_{s \in \mathcal{M}} p_\theta(x_s|z', \tilde{z}_s) \tag{10}$$

for $z = (z', \tilde{z}_1, \dots, \tilde{z}_M) \in \mathsf{Z}$, for shared latent variables $Z'$ and private latent variable $\tilde{Z}^s$ for each $s \in \mathcal{M}$. Note that for $s \neq t \in [M]$,

$$X_s \perp\!\!\!\perp \tilde{Z}_t \mid Z', \tilde{Z}_s. \tag{11}$$

Consequently,

$$p_\theta(z', \tilde{z}_{\mathcal{S}}, \tilde{z}_{\backslash \mathcal{S}}|x_{\mathcal{S}}) = p_\theta(z', \tilde{z}_{\mathcal{S}}, |x_{\mathcal{S}}) p_\theta(\tilde{z}_{\backslash \mathcal{S}}|z', \tilde{z}_{\mathcal{S}}, x_{\mathcal{S}}) = p_\theta(z', \tilde{z}_{\mathcal{S}}, |x_{\mathcal{S}}) p_\theta(\tilde{z}_{\backslash \mathcal{S}}|z', \tilde{z}_{\mathcal{S}}). \tag{12}$$

An encoding distribution $q_\phi(z|x_{\mathcal{S}})$ that approximates $p_\theta(z|x_{\mathcal{S}})$ should thus be unaffected by the inputs $x_{\mathcal{S}}$ when encoding $\tilde{z}_s$ for $s \notin \mathcal{S}$, provided that, a priori, all private and shared latent variables are independent. Observe that for $f_\vartheta$ with the representation

$$f_\vartheta(h_{\mathcal{S}}) = \rho_\vartheta\left(\sum_{s \in \mathcal{S}} g_\vartheta(h_{\mathcal{S}})_s\right),$$

where $\rho_\vartheta$ has aggregated inputs $y$, and that parameterises the encoding distribution of $z = (z', \tilde{z}_\mathcal{S}, \tilde{z}_{\backslash \mathcal{S}})$, the gradients of its $i$-th dimension with respect to the modality values $x_s$ is

$$\frac{\partial}{\partial x_s}[f_\vartheta(h_\mathcal{S}(x_\mathcal{S}))_i] = \frac{\partial \rho_{\vartheta,i}}{\partial y}\left(\sum_{t \in \mathcal{S}} g_\vartheta(h_\mathcal{S}(x_\mathcal{S})_t)\right)\frac{\partial}{\partial x_s}\left(\sum_{t \in \mathcal{S}} g_\vartheta(h_\mathcal{S}(x_\mathcal{S}))_t\right).$$

In the case of a SumPooling aggregation, the gradient simplifies to

$$\frac{\partial \rho_{\vartheta,i}}{\partial y}\left(\sum_{t \in \mathcal{S}} \chi_\vartheta(h_t(x_t))\right)\frac{\partial \chi_\vartheta}{\partial h}(h_s(x_s))\frac{\partial h_s(x_s)}{\partial x_s}.$$

Suppose that the $i$-th component of $\rho_\vartheta$ maps to the the mean or log-standard deviation of some component of $\tilde{Z}_s$ for some $s \in \mathcal{M} \setminus \mathcal{S}$. Notice that only the first factor depends on $i$ so that for this gradient to be zero, $\rho_{\vartheta,i}$ has to be locally constant around $y = \sum_{s \in \mathcal{S}} \chi_\vartheta(h_s(x_s))$ if some other components have a non-zero gradient with respect to $X_s$. It it thus very likely that inputs $X_s$ for $s \in \mathcal{S}$ can impact the distribution of the private latent variables $\tilde{z}_{\backslash \mathcal{S}}$.

However, the specific generative model also lends itself to an alternative parameterisation which guarantees that cross-modal reconstruction likelihoods from $X_{\backslash \mathcal{S}}$ do not affect the encoding distribution of $\tilde{Z}_\mathcal{S}$ under our new variational bound. The assumption of private latent variables suggests an additional permutation-equivariance into the encoding distribution that approximates the posterior in (12), in the sense that for any permutation $\pi \in \mathbb{S}_\mathcal{S}$, it holds that

$$q'_\phi(\tilde{z}_\mathcal{S}|\pi \cdot h_\varphi(x_\mathcal{S}), z') = q'_\phi(\pi \cdot \tilde{z}_\mathcal{S}|h_\varphi(x_\mathcal{S}), z'),$$

assuming that all private latent variables are of the same dimension $D$.[6] Indeed, suppose we have modality-specific feature functions $h_{\varphi,s}$ such that $\{H_s = h_{\varphi,s}(X_s)\}_{s \in \mathcal{S}}$ is exchangeable. Clearly, (11) implies for any $s \neq t$ that

$$h_{\varphi,s}(X_s) \perp\!\!\!\perp \tilde{Z}_t \mid Z', \tilde{Z}_s.$$

The results from Bloem-Reddy and Teh (2020) then imply, for fixed $|\mathcal{S}|$, the existence of a function $f^\star$ such that for all $s \in \mathcal{S}$, almost surely,

$$(H_\mathcal{S}, \tilde{Z}_s) = (H_\mathcal{S}, f^\star(\Xi_s, Z', H_s, \mathbb{M}_{H_\mathcal{S}})), \text{ where } \Xi_s \sim \mathcal{U}[0,1] \text{ iid and } \Xi_s \perp\!\!\!\perp H_\mathcal{S}. \tag{13}$$

This fact suggests an alternative route to approximate the posterior distribution in (12): First, $p_\theta(\tilde{z}_{\backslash \mathcal{S}}|z', \tilde{z}_\mathcal{S})$ can often be computed analytically based on the learned or fixed prior distribution. Second, a permutation-invariant scheme can be used to approximate $p_\theta(z'|x_\mathcal{S})$. Finally, a permutation-equivariant scheme can be employed to approximate $p_\theta(\tilde{z}_\mathcal{S}|x_\mathcal{S}, z')$ with a reparameterisation in the form of (13). Three examples of such permutation-equivariant schemes are given below with pseudocode for optimising the variational bound given in Algorithm 2.

**Example 18** (Permutation-equivariant PoE). Similar to previous work Wang et al. (2016); Lee and Pavlovic (2021); Sutter et al. (2020), we consider an encoding density of the form

$$q_\phi(z', \tilde{z}_\mathcal{M}|x_\mathcal{S}) = q_\varphi^{\text{PoE}}(z'|x_\mathcal{S})\prod_{s \in \mathcal{S}} q_\mathcal{N}(\tilde{z}_s|\tilde{\mu}_{s,\varphi}(x_s), \tilde{\Sigma}_{s,\varphi}(x_s))\prod_{s \in \mathcal{M} \setminus \mathcal{S}} p_\theta(\tilde{z}_s),$$

where

$$q_\varphi^{\text{PoE}}(z'|x_\mathcal{S}) = \frac{1}{\mathcal{Z}}p_\theta(z')\prod_{s \in \mathcal{S}} q_\mathcal{N}(z'|\mu'_{s,\varphi}(x_s), \Sigma'_{s,\varphi}(x_s))$$

is a (permutation-invariant) PoE aggregation, and we assumed that the prior density factorises over the shared and different private variables. For each modality $s$, we encode different features $h'_{s,\varphi} = (\mu'_{s,\varphi}, \Sigma'_{s,\varphi})$ and $\tilde{h}_{s,\varphi} = (\tilde{\mu}_{s,\varphi}, \tilde{\Sigma}_{s,\varphi})$ for the shared, respectively, private, latent variables.

**Example 19** (Permutation-equivariant Sum-Pooling). We consider an encoding density that writes as

$$q_\phi(z', \tilde{z}_\mathcal{M}|x_\mathcal{S}) = q_\phi^{\text{SumP}}(z'|x_\mathcal{S})q_\phi^{\text{Equiv-SumP}}(\tilde{z}_\mathcal{S}|z', x_\mathcal{S})\prod_{s \in \mathcal{M} \setminus \mathcal{S}} p_\theta(\tilde{z}_s|z').$$

---

[6]The effective dimension can vary across modalities in practice if the decoders are set to mask redundant latent dimensions.

Here, we use a (permutation-invariant) Sum-Pooling aggregation scheme for constructing the shared latent variable $Z' = \mu'(h_{\mathcal{S}}) + \sigma'(h_{\mathcal{S}}) \odot \Xi' \sim q_\phi^{\text{SumP}}(z'|x_{\mathcal{S}})$, where $\Xi' \sim p$ and $f_\vartheta \colon \mathbb{R}^{|\mathcal{S}| \times D_E} \to \mathbb{R}^D$ given as in Example (5) with $[\mu'(h), \log \sigma'(h)] = f_\vartheta(h)$. To sample $\tilde{Z}_{\mathcal{S}} \sim q_\phi^{\text{Equiv-SumP}}(\tilde{z}_{\mathcal{S}}|z', x_{\mathcal{S}})$, consider functions $\chi_{j,\vartheta} \colon \mathbb{R}^{D_E} \to \mathbb{R}^{D_P}$, $j \in [3]$, and $\rho_\vartheta \colon \mathbb{R}^{D_P} \to \mathbb{R}^{D_O}$, e.g., fully-connected neural networks. We define $f_\vartheta^{\text{Equiv-SumP}} \colon \mathsf{Z} \times \mathbb{R}^{|\mathcal{S}| \times D_E} \to \mathbb{R}^{|\mathcal{S}| \times D_O}$ via

$$f_\vartheta^{\text{Equiv-SumP}}(z', h_{\mathcal{S}})_s = \rho_\vartheta \left( \left[ \sum_{t \in \mathcal{S}} \chi_{0,\vartheta}(h_t) \right] + \chi_{1,\vartheta}(z') + \chi_{2,\vartheta}(h_s) \right).$$

With $\left[ \tilde{\mu}(h_{\mathcal{S}})^\top, \log \tilde{\sigma}(h_{\mathcal{S}})^\top \right]^\top = f_\vartheta^{\text{Equiv-SumP}}(z', h_{\mathcal{S}})$, we then set $\tilde{Z}_s = \tilde{\mu}(h_{\mathcal{S}})_s + \tilde{\sigma}(h_{\mathcal{S}})_s \odot \tilde{\Xi}_s$ for $\tilde{\Xi}_s \sim p$ iid, $h_s = h_{\varphi,s}(x_s)$ for modality-specific feature functions $h_{\varphi,s} \colon \mathsf{X}_s \to \mathbb{R}^{D_E}$.

**Example 20** (Permutation-equivariant Self-Attention)**.** Similar to a Sum-Pooling approach, we consider an encoding density that writes as

$$q_\phi(z', \tilde{z}_{\mathcal{M}}|x_{\mathcal{S}}) = q_\phi^{\text{SA}}(z'|x_{\mathcal{S}}) q_\phi^{\text{Equiv-SA}}(\tilde{z}_{\mathcal{S}}|z', x_{\mathcal{S}}) \prod_{s \in \mathcal{M} \setminus \mathcal{S}} p_\theta(\tilde{z}_s|z').$$

Here, the shared latent variable $Z'$ is sampled via the permutation-invariant aggregation above by summing the elements of a permutation-equivariant transformer model of depth $L'$. For encoding the private latent variables, we follow the example above but set

$$\left[ \tilde{\mu}(h_{\mathcal{S}})^\top, \log \tilde{\sigma}(h_{\mathcal{S}})^\top \right]^\top = f_\vartheta^{\text{Equiv-SA}}(z', h_{\mathcal{S}})_s = g_{\mathcal{S}}^L,$$

with $g_{\mathcal{S}}^k = \text{MTB}_\vartheta(g_{\mathcal{S}}^{k-1})$ an $g^0 = (\chi_{1,\vartheta}(h_s) + \chi_{2,\vartheta}(z'))_{s \in \mathcal{S}}$.

**Remark 21** (Cross-modal context variables)**.** In contrast to the PoE model, where the private encodings are independent, the private encodings are dependent in the Sum-Pooling model by conditioning on a sample from the shared latent space. The shared latent variable $Z'$ can be seen as a shared cross-modal context variable, and similar probabilistic constructions to encode such context variables via permutation-invariant models have been suggested in few-shot learning algorithms (Edwards and Storkey, 2016; Giannone and Winther, 2022) or, particularly, for neural process models (Garnelo et al., 2018b;a; Kim et al., 2018).

**Remark 22** (Variational bounds with private latent variables)**.** To compute the multi-modal variational bounds, notice that the required KL-divergences can be written as follows:

$$\mathsf{KL}(q_\phi(z', \tilde{z}|x_{\mathcal{S}})|p_\theta(z', \tilde{z})) = \mathsf{KL}(q_\phi(z'|x_{\mathcal{S}})|p_\theta(z')) + \int q_\phi(z'|x_{\mathcal{S}}) \mathsf{KL}(q_\phi(\tilde{z}\mathcal{S}|z', x_{\mathcal{S}})|p_\theta(\tilde{z}_{\mathcal{S}}|z')) \mathrm{d}z'$$

and

$$\mathsf{KL}(q_\phi(z', \tilde{z}|x_{\mathcal{M}})|q_\phi(z', \tilde{z}|x_{\mathcal{S}}))$$
$$= \mathsf{KL}(q_\phi(z'|x_{\mathcal{M}})|(q_\phi(z'|x_{\mathcal{S}})) + \int q_\phi(z'|x_{\mathcal{M}}) \mathsf{KL}(q_\phi(\mathsf{P}_{\mathcal{S}}\tilde{z}|z', x_{\mathcal{M}})|q_\phi(\mathsf{P}_{\mathcal{S}}\tilde{z}|z', x_{\mathcal{S}})) \mathrm{d}z'$$
$$+ \int q_\phi(z'|x_{\mathcal{M}}) \mathsf{KL}(q_\phi(\mathsf{P}_{\setminus \mathcal{S}}\tilde{z}|z', x_{\mathcal{S}})|p_\theta(\mathsf{P}_{\setminus \mathcal{S}}\tilde{z}|z')) \mathrm{d}z'$$

where $\mathsf{P}_{\mathcal{S}} \colon (\tilde{z}_1, \ldots \tilde{z}_M) \mapsto (\tilde{z}_s)_{s \in \mathcal{S}}$ projects all private latent variables to those contained in $\mathcal{S}$.

These expressions can be used to compute our overall variational bound $\mathcal{L}_{\mathcal{S}} + \mathcal{L}_{\setminus \mathcal{S}}$ via

$$\int q_\phi(z'|x_{\mathcal{S}}) q_\phi(\tilde{z}_{\mathcal{S}}|z', x_{\mathcal{S}})] \log p_\theta(x_{\mathcal{S}}|z', \tilde{z}_{\mathcal{S}}) \mathrm{d}z' \mathrm{d}\tilde{z}_{\mathcal{S}}$$
$$- \mathsf{KL}\Big( q_\phi(z'|x_{\mathcal{S}}) q_\phi(\tilde{z}_{\mathcal{S}}|z', x_{\mathcal{S}}) \Big| p_\theta(z') p_\theta(\tilde{z}_{\mathcal{S}}|z') \Big)$$
$$+ \int q_\phi(z'|x_{\mathcal{M}}) q_\phi(\tilde{z}_{\setminus \mathcal{S}}|z', x_{\mathcal{M}})] \log p_\theta(x_{\mathcal{S}}|z', \tilde{z}_{\setminus \mathcal{S}}) \mathrm{d}z' \mathrm{d}\tilde{z}_{\mathcal{S}}$$
$$- \mathsf{KL}\Big( q_\phi(z', \tilde{z}_{\mathcal{S}}, \tilde{z}_{\setminus \mathcal{S}}|x_{\mathcal{M}}) \Big| q_\phi(z', \tilde{z}_{\mathcal{S}}, \tilde{z}_{\setminus \mathcal{S}}|x_{\mathcal{S}}) \Big).$$

**Remark 23** (Comparison with MMVAE+ variational bound). It is instructive to compare our bound with the MMVAE+ approach suggested in Palumbo et al. (2023). Assuming a uniform masking distribution restricted to uni-modal sets so that $\mathcal{S} = \{s\}$ for some $s \in \mathcal{M}$, we can write the bound from Palumbo et al. (2023) as $\frac{1}{M} \sum_{s=1}^{M} \mathcal{L}_{\{s\}}^{\text{MMVAE+}}(x)$ with

$$
\begin{aligned}
\mathcal{L}_{\{s\}}^{\text{MMVAE+}}(x) &= \int q_\phi(z'|x_{\{s\}}) q_\phi(\tilde{z}_{\{s\}}|x_{\{s\}}) \Big[ \log p_\theta(x_{\{s\}}|z', \tilde{z}_{\{s\}}) \Big] \mathrm{d}z' \mathrm{d}\tilde{z}_{\{s\}} \\
&+ \int q_\phi(z'|x_{\{s\}}) r_\phi(\tilde{z}_{\backslash\{s\}}) \Big[ \log p_\theta(x_{\backslash\{s\}}|z', \tilde{z}_{\backslash\{s\}}) \Big] \mathrm{d}z' \mathrm{d}\tilde{z}_{\backslash\{s\}} \\
&- \mathsf{KL}\Big( q_\phi^{\text{MoE}}(z', \tilde{z}_\mathcal{M}|x_\mathcal{M}) \Big| p_\theta(z') p_\theta(\tilde{z}_\mathcal{M}) \Big).
\end{aligned}
$$

Here, it is assumed that the multi-modal encoding distribution for computing the KL-divergence is of the form

$$
q_\phi^{\text{MoE}}(z', \tilde{z}_\mathcal{M}|x_\mathcal{M}) = \frac{1}{M} \sum_{s \in \mathcal{M}} \left( q_\phi(z'|x_s) q_\phi(\tilde{z}_s|x_s) \right)
$$

and $r_\phi(\tilde{z}_\mathcal{A}) = \prod_{s \in \mathcal{A}} r_\phi(\tilde{z}_s)$ are additional trainable *prior* distributions.

## G  Multi-modal posterior in exponential family models

Consider the setting where the decoding and encoding distributions are of the exponential family form, that is

$$
p_\theta(x_s|z) = \mu_s(x_s) \exp\left[ \langle T_s(x_s), f_{s,\theta}(z) \rangle - \log Z_s(f_{s,\theta}(z)) \right]
$$

for all $s \in \mathcal{M}$, while for all $\mathcal{S} \subset \mathcal{M}$,

$$
q_\phi(z|x_\mathcal{S}) = \mu(z) \exp\left[ \langle V(z), \lambda_{\phi,\mathcal{S}}(x_\mathcal{S}) \rangle - \log \Gamma_\mathcal{S}(\lambda_{\phi,\mathcal{S}}(x_\mathcal{S})) \right]
$$

where $\mu_s$ and $\mu$ are base measures, $T_s(x_s)$ and $V(z)$ are sufficient statistics, while the natural parameters $\lambda_{\phi,\mathcal{S}}(x_\mathcal{S})$ and $f_{s,\theta}(z)$ are parameterised by the decoder or encoder networks, respectively, with $Z_s$ and $\Gamma_\mathcal{S}$ being normalising functions. Note that we made a standard assumption that the multi-modal encoding distribution has a fixed base measure and sufficient statistics for any modality subset. For fixed generative parameters $\theta$, we want to learn a multi-modal encoding distribution that minimises, see Remark 4, over $x_\mathcal{S} \sim p_d$,

$$
\begin{aligned}
&\mathsf{KL}(q_\phi(z|x_\mathcal{S})|p_\theta(z|x_\mathcal{S})) \\
&= \int q_\phi(z|x_\mathcal{S}) \Big[ \log q_\phi(z|x_\mathcal{S}) - \log p_\theta(z) - \sum_{s \in \mathcal{S}} \log p_\theta(x_s|z) \Big] \mathrm{d}z - \log p_\theta(x_\mathcal{S}) \\
&= \int q_\phi(z|x_\mathcal{S}) \Big[ \langle V(z), \lambda_{\phi,\mathcal{S}}(x_\mathcal{S}) \rangle - \log \Gamma_\mathcal{S}(\lambda_{\phi,\mathcal{S}}(x_\mathcal{S})) - \sum_{s \in \mathcal{S}} \log \mu_s(x_s) \\
&\quad - \Big\{ \sum_{s \in \mathcal{S}} \langle T_{s,\theta}(x_s), f_{s,\theta}(z) \rangle + \log p_\theta(z) - \sum_{s \in \mathcal{S}} Z_s(f_{s,\theta}(z)) \Big\} \Big] \mathrm{d}z - \log p_\theta(x_\mathcal{S}) \\
&= \int q_{\phi,\vartheta}(z|x_\mathcal{S}) \Big[ \Big\langle \begin{bmatrix} V(z) \\ 1 \end{bmatrix}, \begin{bmatrix} \lambda_{\phi,\vartheta,\mathcal{S}}(x_\mathcal{S}) \\ -\log \Gamma_\mathcal{S}(\lambda_{\phi,\vartheta,\mathcal{S}}(x_\mathcal{S})) \end{bmatrix} \Big\rangle - \sum_{s \in \mathcal{S}} \Big\langle \begin{bmatrix} T_s(x_s) \\ 1 \end{bmatrix}, \begin{bmatrix} f_{\theta,s}(z) \\ b_{\theta,s}(z) \end{bmatrix} \Big\rangle \Big] \mathrm{d}z,
\end{aligned}
$$

with $b_{\theta,s}(z) = \frac{1}{|\mathcal{S}|} p_\theta(z) - \log Z_s(f_{s,\theta}(z))$.

## H  Identifiability

Identifiability of parameters and latent variables in latent structure models is a classic problem (Koopmans and Reiersol, 1950; Kruskal, 1976; Allman et al., 2009), that has been studied increasingly for non-linear latent variable models, e.g., for ICA (Hyvarinen and Morioka, 2016; Hälvä and Hyvarinen, 2020; Hälvä et al., 2021), VAEs (Khemakhem et al., 2020a; Zhou and Wei, 2020; Wang et al., 2021; Moran et al., 2021; Lu et al., 2022), EBMs (Khemakhem et al., 2020b), flow-based (Sorrenson et al., 2020) or mixture models (Kivva et al., 2022).

We are interested in identifiability, conditional on having observed some non-empty modality subset $\mathcal{S} \subset \mathcal{M}$. For illustration, we translate an identifiability result from the uni-modal iVAE setting in Lu et al. (2022), which does not require the conditional independence assumption from Khemakhem et al. (2020a). We assume that the encoding distribution $q_\phi(z|x_\mathcal{S})$ approximates the true posterior $p_\theta(z|x_\mathcal{S})$ and belongs to a strongly exponential family, i.e.,

$$p_\theta(z|x_\mathcal{S}) = q_\phi(z|x_\mathcal{S}) = p_{V_{\phi,\mathcal{S}}, \lambda_{\phi,\mathcal{S}}}^{\mathrm{EF}}(z|x_\mathcal{S}), \tag{14}$$

with

$$p_{V_\mathcal{S}, \lambda_\mathcal{S}}^{\mathrm{EF}}(z|x_\mathcal{S}) = \mu(z) \exp\left[\langle V_\mathcal{S}(z), \lambda(x_\mathcal{S})\rangle - \log\Gamma_\mathcal{S}(\lambda_\mathcal{S}(x_\mathcal{S}))\right],$$

where $\mu$ is a base measure, $V_\mathcal{S}\colon \mathsf{Z} \to \mathbb{R}^k$ is the sufficient statistics, $\lambda_\mathcal{S}(x_\mathcal{S}) \in \mathbb{R}^k$ the natural parameters and $\Gamma_\mathcal{S}$ a normalising term. Furthermore, one can only reduce the exponential component to the base measure on sets having measure zero. In this section, we assume that

$$p_\theta(x_s|z) = p_{s,\epsilon}(x_s - f_{\theta,s}(z)) \tag{15}$$

for some fixed noise distribution $p_{s,\epsilon}$ with a Lebesgue density, which excludes observation models for discrete modalities. Let $\Theta_\mathcal{S}$ be the domain of the parameters $\theta_\mathcal{S} = (f_{\backslash\mathcal{S}}, V_\mathcal{S}, \lambda_\mathcal{S})$ with $f_{\backslash\mathcal{S}}\colon \mathsf{Z} \ni z \mapsto (f_s(z))_{s \in \mathcal{M}\backslash\mathcal{S}} \in \times_{s \in \mathcal{M}\backslash\mathcal{S}}\mathsf{X}_s = \mathsf{X}_{\backslash\mathcal{S}}$. Assuming (14), note that

$$p_{\theta_\mathcal{S}}(x_{\backslash\mathcal{S}}|x_\mathcal{S}) = \int p_{V_\mathcal{S}, \lambda_\mathcal{S}}(z|x_\mathcal{S}) p_{\backslash\mathcal{S},\epsilon}(x_{\backslash\mathcal{S}} - f_{\backslash\mathcal{S}}(z)) \mathrm{d}z,$$

with $p_{\backslash\mathcal{S},\epsilon} = \otimes_{s \in \mathcal{M}\backslash\mathcal{S}} p_{s,\epsilon}$. We define an equivalence relation on $\Theta_\mathcal{S}$ by $(f_{\backslash\mathcal{S}}, V_\mathcal{S}, \lambda_\mathcal{S}) \sim_{A_\mathcal{S}} (\tilde{f}_{\backslash\mathcal{S}}, \tilde{V}_\mathcal{S}, \tilde{\lambda}_\mathcal{S})$ iff there exist invertible $A_\mathcal{S} \in \mathbb{R}^{k \times k}$ and $c_\mathcal{S} \in \mathbb{R}^k$ such that

$$V_\mathcal{S}(f_{\backslash\mathcal{S}}^{-1}(x_{\backslash\mathcal{S}})) = A_\mathcal{S}\tilde{V}_\mathcal{S}(\tilde{f}_{\backslash\mathcal{S}}^{-1}(x_{\backslash\mathcal{S}})) + c_\mathcal{S}$$

for all $x_{\backslash\mathcal{S}} \in \mathsf{X}_{\backslash\mathcal{S}}$.

**Proposition 24** (Weak identifiability). *Consider the data generation mechanism $p_\theta(z,x) = p_\theta(z)\prod_{s \in \mathcal{M}} p_\theta(x_s|z)$ where the observation model satisfies (15) for an injective $f_{\backslash\mathcal{S}}$. Suppose further that $p_\theta(z|x_\mathcal{S})$ is strongly exponential and (14) holds. Assume that the set $\{x_{\backslash\mathcal{S}} \in \mathsf{X}_{\backslash\mathcal{S}}|\varphi_{\backslash\mathcal{S},\epsilon}(x_{\backslash\mathcal{S}}) = 0\}$ has measure zero, where $\varphi_{\backslash\mathcal{S},\epsilon}$ is the characteristic function of the density $p_{\backslash\mathcal{S},\epsilon}$. Furthermore, suppose that there exist $k+1$ points $x_\mathcal{S}^0, \ldots, x_\mathcal{S}^k \in \mathsf{X}_\mathcal{S}$ such that*

$$L = \left[\lambda_\mathcal{S}(x_\mathcal{S}^1) - \lambda_\mathcal{S}(x_\mathcal{S}^0), \ldots, \lambda_\mathcal{S}(x_\mathcal{S}^k) - \lambda_\mathcal{S}(x_\mathcal{S}^0)\right] \in \mathbb{R}^{k \times k}$$

*is invertible. Then $p_{\theta_\mathcal{S}}(x_{\backslash\mathcal{S}}|x_\mathcal{S}) = p_{\tilde{\theta}_\mathcal{S}}(x_{\backslash\mathcal{S}}|x_\mathcal{S})$ for all $x \in \mathsf{X}$ implies $\theta \sim_{A_\mathcal{S}} \tilde{\theta}$.*

This result follows from Theorem 4 in Lu et al. (2022). Note that $p_{\theta_\mathcal{S}}(x_{\backslash\mathcal{S}}|x_\mathcal{S}) = p_{\tilde{\theta}_\mathcal{S}}(x_{\backslash\mathcal{S}}|x_\mathcal{S})$ for all $x \in \mathsf{X}$ implies with the regularity assumption on $\varphi_{\backslash\mathcal{S},\epsilon}$ that the transformed variables $Z = f_{\backslash\mathcal{S}}^{-1}(X_{\backslash\mathcal{S}})$ and $\tilde{Z} = \tilde{f}_{\backslash\mathcal{S}}^{-1}(X_{\backslash\mathcal{S}})$ have the same density function conditional on $X_\mathcal{S}$.

**Remark 25.** The joint decoder function $f_{\backslash\mathcal{S}}$ can be injective, even if the individual modality-specific decoder functions are not, suggesting that the identifiability of latent variables can be improved when training a multi-modal model compared to separate uni-modal models.

**Remark 26.** The identifiability result above is about conditional models and does not contradict the un-identifiability of VAEs: When $\mathcal{S} = \emptyset$ and we view $x = x_\mathcal{M}$ as one modality, then the parameters of $p_{\theta_\emptyset}(x)$ characterised by the parameters $V_\emptyset$ and $\lambda_\emptyset$ of the prior $p_{\theta_\emptyset}(z|x_\emptyset)$ and the encoders $f_\mathcal{M}$ will not be identifiable as the invertibility condition will not be satisfied.

**Remark 27.** Note that the identifiablity concerns parameters of the multi-modal posterior distribution. We believe that our inference approach is beneficial for this type of identifiability because (a) unlike some other variational bounds, the posterior is the optimal variational distribution with $\mathcal{L}_{\backslash\mathcal{S}}(x)$ being a lower bound on $\log p_\theta(x_{\backslash\mathcal{S}}|x_\mathcal{S})$ for flexible encoders, and (b) the trainable aggregation schemes can be more flexible for approximating the optimal encoding distribution.

**Remark 28.** For models with private latent variables, we might not expect that conditioning on $X_\mathcal{S}$ helps to identify $\tilde{Z}_{\backslash\mathcal{S}}$ as $p_\theta(z', \tilde{z}_\mathcal{S}, \tilde{z}_{\backslash\mathcal{S}}|x_\mathcal{S}) = p_\theta(z', \tilde{z}_\mathcal{S}|x_\mathcal{S})p_\theta(\tilde{z}_{\backslash\mathcal{S}}|z', \tilde{z}_{\backslash\mathcal{S}})$. Indeed, Proposition 24 will not apply in such models as $f_{\backslash\mathcal{S}}$ will not be injective.

## I  MISSING MODALITIES

In practical applications, modalities can be missing for different data points. We describe this missingness pattern by missingness mask variables $m_s \in \{0, 1\}$ where $m_s = 1$ indicates that observe modality $s$, while $m_s = 0$ means it is missing. The joint generative model that extends (16) will be of the form $p_\theta(z, x, m) = p_\theta(z) \prod_{s \in \mathcal{M}} p_\theta(x_s|z) p_\theta(m|x)$ for some distribution $p_\theta(m|x)$ over the mask variables $m = (m_s)_{s \in \mathcal{M}}$. For $\mathcal{S} \subset \mathcal{M}$, we denote by $x_\mathcal{S}^o = \{x_s : m_s = 1, s \in \mathcal{S}\}$ and $x_\mathcal{S}^m = \{x_s : m_s = 0, s \in \mathcal{S}\}$ the set of observed, respectively missing, modalities. The full likelihood of the observed and missingness masks becomes then $p_\theta(x_\mathcal{S}^o, m) = \int p_\theta(z) \prod_{s \in \mathcal{S}} p_\theta(x_s|z) p_\theta(m|x) \mathrm{d}x_s^m \mathrm{d}z$. If $p_\theta(m|x)$ does not depend on the observations, that is, observations are missing completely at random (Rubin, 1976), then the missingness mechanisms $p_\theta(m|x)$ for inference approaches maximizing $p_\theta(x^o, m)$ can be ignored. Consequently, one can instead concentrate on maximizing $\log p_\theta(x^o)$ only, based on the joint generative model $p_\theta(z, x^o) = p_\theta(z) \prod_{\{s \in \mathcal{M} : m_s = 1\}} p_\theta(x_s|z)$. In particular, one can employ the variational bounds above by considering only the observed modalities. Since masking operations are readily supported for the considered permutation-invariant models, appropriate imputation strategies (Nazabal et al., 2020; Ma et al., 2019) for the encoded features of the missing modalities are not necessarily required. Settings allowing for not (completely) at random missingness have been considered in the uni-modal case, for instance, in Ipsen et al. (2021); Ghalebikesabi et al. (2021); Gong et al. (2021), and we leave multi-modal extensions thereof for future work.

## J  MIXTURE MODEL EXTENSIONS FOR DIFFERENT VARIATIONAL BOUNDS

We consider the optimization of an augmented variational bound

$$\mathcal{L}(x, \theta, \phi) = \int \rho(\mathcal{S}) \Big[ \int q_\phi(c, z|x_\mathcal{S}) \left[\log p_\theta(c, x_\mathcal{S}|z)\right] \mathrm{d}z \mathrm{d}c - \mathsf{KL}(q_\phi(c, z|x_\mathcal{S})|p_\theta(c, z))$$
$$+ \int q_\phi(c, z|x_\mathcal{S}) \left[\log p_\theta(x_{\setminus \mathcal{S}}|z)\right] \mathrm{d}z \mathrm{d}c - \mathsf{KL}(q_\phi(c, z|x)|q_\phi(c, z|x_\mathcal{S})) \Big] \mathrm{d}\mathcal{S}.$$

We will pursue here an encoding approach that does not require modelling the encoding distribution over the discrete latent variables explicitly, thus avoiding large variances in score-based Monte Carlo estimators or resorting to advanced variance reduction techniques or alternatives such as continuous relaxation approaches.

Assuming a structured variational density of the form

$$q_\phi(c, z|x_\mathcal{S}) = q_\phi(z|x_\mathcal{S}) q_\phi(c|z, x_\mathcal{S}),$$

we can express the augmented version of (3) via

$$\mathcal{L}_\mathcal{S}(x_\mathcal{S}, \theta, \phi) = \int q_\phi(c, z|x_\mathcal{S}) \left[\log p_\theta(c, x_\mathcal{S}|z)\right] \mathrm{d}z - \beta \mathsf{KL}(q_\phi(c, z|x_\mathcal{S})|p_\theta(c, z))$$
$$= \int q_\phi(z|x_\mathcal{S}) \left[f_x(z, x_\mathcal{S}) + f_c(z, x_\mathcal{S})\right] \mathrm{d}z,$$

where $f_x(z, x_\mathcal{S}) = \log p_\theta(x_\mathcal{S}|z) - \beta \log q_\phi(z|x_\mathcal{S}))$ and

$$f_c(z, x_\mathcal{S}) = \int q_\phi(c|z, x_\mathcal{S}) \left[-\beta \log q_\phi(c|z, x_\mathcal{S}) + \beta \log p_\theta(c, z)\right] \mathrm{d}c. \tag{16}$$

We can also write the augmented version of (4) in the form of

$$\mathcal{L}_{\setminus \mathcal{S}}(x, \theta, \phi) = \int q_\phi(c, z|x_\mathcal{S}) \left[\log p_\theta(x_{\setminus \mathcal{S}}|z)\right] \mathrm{d}z - \beta \mathsf{KL}(q_\phi(c, z|x)|q_\phi(c, z|x_\mathcal{S}))$$
$$= \int q_\phi(z|x) g_x(z, x) \mathrm{d}z$$

where

$$g_x(z, x) = \log p_\theta(x_{\setminus \mathcal{S}}|z) - \beta \log q_\phi(z|x) + \beta \log q_\phi(z|x_\mathcal{S})$$

which does not depend on the encoding density of the cluster variable. To optimize the variational bound with respect to the cluster density, we can thus optimize (16), which attains its maximum value of

$$f_c^\star(z, x_S) = \beta \log \int p_\theta(c) p_\theta(z|c) \mathrm{d}c = \beta \log p_\theta(z)$$

at $q_\phi(c|z, x_S) = p_\theta(c|z)$ due to Remark 29 below with $g(c) = \beta \log p_\theta(c, z)$.

**Remark 29** (Entropy regularised optimization). Let $q$ be a density over $\mathsf{C}$, $\exp(g)$ be integrable with respect to $q$ and $\tau > 0$. The maximum of

$$f(q) = \int_\mathsf{C} q(c) \left[ g(c) - \tau \log q(c) \right] \mathrm{d}c$$

that is attained at $q^\star(c) = \frac{1}{\mathcal{Z}} \mathrm{e}^{g(c)/\tau}$ with normalising constant $\mathcal{Z} = \int_\mathsf{C} \mathrm{e}^{g(c)/\tau} \mathrm{d}c$ is

$$f^\star = f(q^\star) = \tau \log \int_\mathsf{C} \mathrm{e}^{g(c)/\tau} \mathrm{d}c$$

We can derive an analogous optimal structured variational density for the mixture-based and total-correlation-based variational bounds. First, we can write the mixture-based bound (1) as

$$\mathcal{L}_S^{\mathrm{Mix}}(x, \theta, \phi) = \int q_\phi(z|x_S) \left[ \log p_\theta(c, x|z) \right] \mathrm{d}z - \beta \mathsf{KL}(q_\phi(c, z|x_S)|p_\theta(c, z))$$

$$= \int q_\phi(z|x_S) \left[ f_x^{\mathrm{Mix}}(z, x) + f_c(z, x) \right] \mathrm{d}z,$$

where $f_x^{\mathrm{Mix}}(z, x) = \log p_\theta(x|z) - \beta \log q_\phi(z|x_S)$ and $f_c(z, x)$ has a maximum value of $f_c^\star(z, x) = \beta \log p_\theta(z)$. Second, we can express the corresponding terms from the total-correlation-based bound as

$$\mathcal{L}_S^{\mathrm{TC}}(\theta, \phi) = \int q_\phi(z|x) \left[ \log p_\theta(x|z) \right] \mathrm{d}z - \beta \mathsf{KL}(q_\phi(c, z|x)|q_\phi(c, z|x_S))$$

$$= \int q_\phi(z|x) \left[ f_x^{\mathrm{TC}}(z, x) \right] \mathrm{d}z,$$

where $f_x^{\mathrm{TC}}(z, x) = \log p_\theta(x|z) - \beta \log q_\phi(z|x) + \beta \log q_\phi(z|x_S)$.

## K  ALGORITHM AND STL-GRADIENT ESTIMATORS

We consider a multi-modal extension of the sticking-the-landing (STL) gradient estimator (Roeder et al., 2017) that has also been used in previous multi-modal bounds (Shi et al., 2019). The gradient estimator ignores the score function terms when sampling $q_\phi(z|x_S)$ for variance reduction purposes due to the fact that it has a zero expectation. For the bounds (2) that involves sampling from $q_\phi(z|x_S)$ and $q_\phi(z|x_\mathcal{M})$, we thus ignore the score terms for both integrals. Consider the reparameterisation with noise variables $\epsilon_S, \epsilon_\mathcal{M} \sim p$ and transformations $z_S = t_S(\phi, \epsilon_S, x_S) = f_{\text{invariant-agg}}(\vartheta, \epsilon_S, S, h_S)$, for $h_S = h_{\varphi,s}(x_s)_{s \in S}$ and $z_\mathcal{M} = t_\mathcal{M}(\phi, \epsilon_\mathcal{M}, x_\mathcal{M}) = f_{\text{invariant-agg}}(\vartheta, \epsilon_\mathcal{M}, \mathcal{M}, h_\mathcal{M})$, for $h_\mathcal{M} = h_{\varphi,s}(x_s)_{s \in \mathcal{M}}$. We need to learn only a single aggregation function that applies that masks the modalities appropriately. Pseudo-code for computing the gradients are given in Algorithm 1. If the encoding distribution is a mixture distribution, we apply the stop-gradient operation also to the mixture weights. Notice that in the case of a mixture prior and an encoding distribution that includes the mixture component, the optimal encoding density over the mixture variable has no variational parameters and is given as the posterior density of the mixture component under the generative parameters of the prior.

In the case of private latent variables, we proceed analogously and rely on reparameterisations $z_S' = t_S'(\phi, \epsilon_S', x_S)$ for the shared latent variable $z_S' \sim q_\phi(z'|x_S)$ as above and $\tilde{z}_S = \tilde{t}_S(\phi, z', \epsilon_S, x_S) = f_{\text{equivariant-agg}}(\vartheta, \tilde{\epsilon}_S, z', S, h_S)$ for the private latent variables $\tilde{z}_S \sim q_\phi(\tilde{z}_S|z', x_S)$. Moreover, we write $\mathsf{P}_S$ for a projection on the $S$-coordinates. Pseudo-code for computing unbiased gradient estimates for our bound is given in Algorithm 2.

---

**Algorithm 1** Single training step for computing unbiased gradients of $\mathcal{L}(x)$.

---

**Input:** Multi-modal data point $x$, generative parameter $\theta$, variational parameters $\phi = (\varphi, \vartheta)$.
Sample $\mathcal{S} \sim \rho$.
Sample $\epsilon_\mathcal{S}, \epsilon_\mathcal{M} \sim p$.
Set $z_\mathcal{S} = t_\mathcal{S}(\phi, \epsilon_\mathcal{S}, x_\mathcal{M})$ and $z_\mathcal{M} = t_\mathcal{M}(\phi, \epsilon_\mathcal{M}, x_\mathcal{M})$.
Stop gradients of variational parameters $\phi' = \mathtt{stop\_grad}(\phi)$.
Set $\widehat{\mathcal{L}}_\mathcal{S}(\theta, \phi) = \log p_\theta(x_\mathcal{S}|z_\mathcal{S}) + \beta \log p_\theta(z_\mathcal{S}) - \beta \log q_{\phi'}(z_\mathcal{S}|x_\mathcal{S})$.
Set $\widehat{\mathcal{L}}_{\backslash\mathcal{S}}(\theta, \phi) = \log p_\theta(x_{\backslash\mathcal{S}}|z_\mathcal{M}) + \beta \log q_\phi(z_\mathcal{M}|x_\mathcal{S}) - \beta \log q_{\phi'}(z_\mathcal{M}|x_\mathcal{M})$.
**Output:** $\nabla_{\theta,\phi}\left[\widehat{\mathcal{L}}_\mathcal{S}(\theta, \phi) + \widehat{\mathcal{L}}_{\backslash\mathcal{S}}(\theta, \phi)\right]$

---

**Algorithm 2** Single training step for computing unbiased gradients of $\mathcal{L}(x)$ with private latent variables.

---

**Input:** Multi-modal data point $x$, generative parameter $\theta$, variational parameters $\phi = (\varphi, \vartheta)$.
Sample $\mathcal{S} \sim \rho$.
Sample $\epsilon'_\mathcal{S}, \epsilon_\mathcal{S}, \epsilon_{\backslash\mathcal{S}}, \epsilon'_\mathcal{M}, \epsilon_\mathcal{M}, \epsilon_{\backslash\mathcal{M}} \sim p$.
Set $z'_\mathcal{S} = t'_\mathcal{S}(\phi, \epsilon'_\mathcal{S}, x_\mathcal{S}), \tilde{z}_\mathcal{S} = \tilde{t}_\mathcal{S}(\phi, z'_\mathcal{S}, \epsilon_\mathcal{S}, x_\mathcal{S})$.
Set $z'_\mathcal{M} = t'_\mathcal{M}(\phi, \epsilon'_\mathcal{M}, x_\mathcal{M}), \tilde{z}_\mathcal{M} = \tilde{t}_\mathcal{M}(\phi, z'_\mathcal{M}, \epsilon_\mathcal{M}, x_\mathcal{M})$.
Stop gradients of variational parameters $\phi' = \mathtt{stop\_grad}(\phi)$.
Set $\widehat{\mathcal{L}}_\mathcal{S}(\theta, \phi) = \log p_\theta(x_\mathcal{S}|z'_\mathcal{S}, \tilde{z}_\mathcal{S}) + \beta \log p_\theta(z'_\mathcal{S}) - \beta \log q_{\phi'}(z'_\mathcal{S}|x_\mathcal{S}) + \beta \log p_\theta(\tilde{z}_\mathcal{S}|z'_\mathcal{S}) - \beta \log q_{\phi'}(\tilde{z}_\mathcal{S}|z'_\mathcal{S}, x_\mathcal{S})$.
Set $\widehat{\mathcal{L}}_{\backslash\mathcal{S}}(\theta, \phi) = \log p_\theta(x_{\backslash\mathcal{S}}|z'_\mathcal{M}) + \beta \log q_\phi(z'_\mathcal{M}|x_\mathcal{S}) - \beta \log q_{\phi'}(\tilde{z}_\mathcal{M}|z'_\mathcal{M}, x_\mathcal{M}) + \beta \log q_\phi(\mathsf{P}_\mathcal{S}(\tilde{z}_\mathcal{M})|z'_\mathcal{M}, x_\mathcal{S}) + \beta \log p_\theta(\mathsf{P}_{\backslash\mathcal{S}}(\tilde{z}_\mathcal{M})|z'_\mathcal{M}, \tilde{z}_\mathcal{M}) - \beta \log q_{\phi'}(\tilde{z}_\mathcal{M}|z'_\mathcal{M}, x_\mathcal{M})$.
**Output:** $\nabla_{\theta,\phi}\left[\widehat{\mathcal{L}}_\mathcal{S}(\theta, \phi) + \widehat{\mathcal{L}}_{\backslash\mathcal{S}}(\theta, \phi)\right]$

---

## L  EVALUATION OF MULTI-MODAL GENERATIVE MODELS

We evaluate models using different metrics suggested previously for multi-modal learning, see for example Shi et al. (2019); Wu and Goodman (2019); Sutter et al. (2021).

**Marginal, conditional and joint log-likelihoods.** We can estimate the marginal log-likelihood using classic importance sampling

$$\log p_\theta(x_\mathcal{S}) \approx \log \frac{1}{K} \sum_{k=1}^{K} \frac{p_\theta(z^k, x_\mathcal{S})}{q_\phi(z^k|x_\mathcal{S})}$$

for $z^k \sim q_\phi(\cdot|x_\mathcal{S})$. This also allows to approximate the joint log-likelihood $\log p_\theta(x)$, and consequently also the conditional $\log p_\theta(x_{\backslash\mathcal{S}}|x_\mathcal{S}) = \log p_\theta(x) - \log p_\theta(x_\mathcal{S})$.

**Generative coherence with joint auxiliary labels.** Following previous work (Shi et al., 2019; Sutter et al., 2021; Daunhawer et al., 2022; Javaloy et al., 2022), we assess whether the generated data share the same information in the form of the class labels across different modalities. To do so, we use pre-trained classifiers $\mathrm{clf}_s: \mathsf{X}_s \to [K]$ that classify values from modality $s$ to $K$ possible classes. More precisely, for $\mathcal{S} \subset \mathcal{M}$ and $m \in \mathcal{M}$, we compute the self- ($m \in \mathcal{S}$) or cross- ($m \notin \mathcal{S}$) coherence $\mathrm{C}_{\mathcal{S} \to m}$ as the empirical average of

$$1_{\{\mathrm{clf}_m(\hat{x}_m)=y\}},$$

over test samples $x$ with label $y$ where $\hat{z}_\mathcal{S} \sim q_\phi(z|x_\mathcal{S})$ and $\hat{x}_m \sim p_\theta(x_m|\hat{z}_\mathcal{S})$. The case $\mathcal{S} = \mathcal{M} \setminus \{m\}$ corresponds to a leave-one-out conditional coherence.

**Linear classification accuracy of latent representations.** To evaluate how the latent representation can be used to predict the shared information contained in the modality subset $\mathcal{S}$ based on a linear model, we consider the accuracy $\mathrm{Acc}_\mathcal{S}$ of a linear classifier $\mathrm{clf}_z: \mathsf{Z} \to [K]$ that is trained to predict the label based on latent samples $z_\mathcal{S} \sim q_\phi(z_\mathcal{S}|x_\mathcal{S}^{\mathrm{train}})$ from the training values $x_\mathcal{S}^{\mathrm{train}}$ and evaluated on latent samples $z_\mathcal{S} \sim q_\phi(z|x_\mathcal{S}^{\mathrm{test}})$ from the test values $x_\mathcal{S}^{\mathrm{test}}$.

# M LINEAR MODELS

**Generative model.** Suppose that a latent variable $Z$ taking values in $\mathbb{R}^D$ is sampled from a standard Gaussian prior $p_\theta(z) = \mathcal{N}(0, \mathrm{I})$ generates $M$ data modalities $X_s \in \mathbb{R}^{D_s}$, $D \le D_s$, based on a linear decoding model $p_\theta(x_s|z) = \mathcal{N}(W_s z + b_s, \sigma^2 \mathrm{I})$ for a factor loading matrix $W_s \in \mathbb{R}^{D_s \times D}$, bias $b_s \in \mathbb{R}^{D_s}$ and observation scale $\sigma > 0$. Note that the annealed likelihood function $\tilde{p}_{\beta,\theta}(x_s|z) = \mathcal{N}(W_s z + b_s, \beta \sigma^2 \mathrm{I})$ corresponds to a scaling of the observation noise, so that we consider only the choice $\sigma = 1$, set $\sigma_\beta = \sigma \beta^{1/2}$ and vary $\beta > 0$. It is obvious that for any $\mathcal{S} \subset \mathcal{M}$, it holds that $\tilde{p}_{\beta,\theta}(x_\mathcal{S}|z) = \mathcal{N}(W_\mathcal{S} z + b_\mathcal{S}, \sigma_\beta^2 \mathrm{I}_\mathcal{S})$, where $W_\mathcal{S}$ and $b_\mathcal{S}$ are given by concatenating row-wise the emission or bias matrices for modalities in $\mathcal{S}$, while $\sigma_\beta^2 \mathrm{I}_\mathcal{S}$ is the diagonal matrix of the variances of the corresponding observations. By standard properties of Gaussian distributions, it follows that $\tilde{p}_{\beta,\theta}(x_\mathcal{S}) = \mathcal{N}(b_\mathcal{S}, C_\mathcal{S})$ where $C_\mathcal{S} = W_\mathcal{S} W_\mathcal{S}^\top + \sigma_\beta^2 \mathrm{I}_\mathcal{S}$ is the data covariance matrix. Furthermore, with $K_\mathcal{S} = W_\mathcal{S}^\top W_\mathcal{S} + \sigma_\beta^2 \mathrm{I}_d$, the adjusted posterior is $\tilde{p}_{\beta,\theta}(z|x_\mathcal{S}) = \mathcal{N}(K_\mathcal{S}^{-1} W_\mathcal{S}^\top (x_\mathcal{S} - b_\mathcal{S}), \sigma_\beta^2 \mathrm{I}_d K_\mathcal{S}^{-1})$. We sample orthogonal rows of $W$ so that the posterior covariance becomes diagonal so that it can – in principle – be well approximated by an encoding distribution with a diagonal covariance matrix. Indeed, the inverse of the posterior covariance matrix is only a function of the generative parameters of the modalities within $\mathcal{S}$ and can be written as the sum $\sigma_\beta^2 \mathrm{I} + W_\mathcal{S}^\top W_\mathcal{S} = \sigma_\beta^2 \mathrm{I} + \sum_{s \in \mathcal{S}} W_s^\top W_s$, while the posterior mean function is $x_\mathcal{S} \mapsto (\sigma_\beta^2 \mathrm{I} + \sum_{s \in \mathcal{S}} W_s^\top W_s)^{-1} \sum_{s \in \mathcal{S}} W_s(x_s - b_s)$.

**Data generation.** We generate 5 data sets of $N = 5000$ samples, each with $M = 5$ modalities. We set the latent dimension to $D = 30$, while the dimension $D_s$ of modality $s$ is drawn from $\mathcal{U}(30, 60)$. We set the observation noise to $\sigma = 1$, shared across all modalities, as is standard for a PCA model. We sample the components of $b_s$ independently from $\mathcal{N}(0, 1)$. For the setting without modality-specific latent variables, $W_s$ is the orthonormal matrix from a QR algorithm applied to a matrix with elements sampled iid from $\mathcal{U}(-1, 1)$. The bias coefficients $W_b$ are sampled independently from $\mathcal{N}(0, 1/d)$. Conversely, the setting with private latent variables in the ground truth model allows us to describe modality-specific variation by considering the sparse loading matrix

$$W_\mathcal{M} = \begin{bmatrix} W'_1 & \tilde{W}_1 & 0 & \dots & 0 \\ W'_2 & 0 & \tilde{W}_2 & \dots & 0 \\ \vdots & \vdots & \ddots & \ddots & \vdots \\ W'_M & 0 & \dots & 0 & \tilde{W}_M \end{bmatrix}.$$

Here, $W'_s, \tilde{W}_s \in \mathbb{R}^{D_s \times D'}$ with $D' = D/(M+1) = 5$, Furthermore, the latent variable $Z$ can be written as $Z = (Z', \tilde{Z}_1, \dots, \tilde{Z}_M)$ for private and shared latent variables $\tilde{Z}_s$, resp. $Z'$. We similarly generate orthonormal $[W'_s, \tilde{W}_s]$ from a QR decomposition. Observe that the general generative model with latent variable $Z$ corresponds to the generative model (10) with shared $Z'$ and private latent variables $\tilde{Z}$ with straightforward adjustments for the decoding functions. Similar models have been considered previously, particularly from a Bayesian standpoint with different sparsity assumptions on the generative parameters (Archambeau and Bach, 2008; Virtanen et al., 2012; Zhao et al., 2016).

**Maximum likelihood estimation.** Assume now that we observe $N$ data points $\{x_n\}_{n \in [N]}$, consisting of stacking the views $x_n = (x_{s,n})_{s \in \mathcal{S}}$ for each modality in $\mathcal{S}$ and let $S = \frac{1}{N} \sum_{n=1}^N (x_n - b)(x_n - b)^\top \in \mathbb{R}^{D_x \times D_x}$, $D_x = \sum_{s=1}^M D_s$, be the sample covariance matrix across all modalities. Let $U_d \in \mathbb{R}^{D_x \times D}$ be the matrix of the first $D$ eigenvectors of $S$ with corresponding eigenvalues $\lambda_1, \dots \lambda_D$ stored in the diagonal matrix $\Lambda_D \in \mathbb{R}^{D \times D}$. The maximum likelihood estimates are then given by $b_{\mathrm{ML}} = \frac{1}{N} \sum_{n=1}^N x_n$, $\sigma_{\mathrm{ML}}^2 = \frac{1}{N-D} \sum_{j=D+1}^N \lambda_j$ and $W_{\mathrm{ML}} = U_D(\Lambda_D - \sigma_{\mathrm{ML}}^2 \mathrm{I})^{1/2}$ with the loading matrix identifiable up to rotations.

**Model architectures.** We estimate the observation noise scale $\sigma$ based on the maximum likelihood estimate $\sigma_{\mathrm{ML}}$. We assume linear decoder functions $p_\theta(x_s|z) = \mathcal{N}(W_s^\theta z + b^\theta, \sigma_{\mathrm{ML}}^2)$, fixed standard Gaussian prior $p(z) = \mathcal{N}(0, \mathrm{I})$ and generative parameters $\theta = (W_1^\theta, b_1^\theta, \dots, W_M^\theta, b_M^\theta)$. Details about the various encoding architectures are given in Table 17. The modality-specific encoding

functions for the PoE and MoE schemes have a hidden size of $512$, whilst they are of size $256$ for the learnable aggregation schemes having additional aggregation parameters $\varphi$.

**Simulation results.** We show different rate-distortion terms for the learned models where the true data generation mechanism does not contain private latent variables (see Figure 3) or does contain private latent variables (see Figure 4 ). In both settings, we use the general multi-modal model without private latent variables in order to compare different aggregation schemes and bounds. We find that our bound yields encoding distributions that are closer to the true posterior distribution across various aggregation schemes. Note that in the case of the mixture-based bound, the posterior distribution is only optimal as an encoding distribution that uses all modalities (Sub-figures (c)). The trade-offs between full reconstruction quality and full rates vary across ground truth models, bounds and aggregation. Cross-reconstruction terms are usually better for the mixture-based bound. Moreover, the mixture-based bound has lower cross-modal rates, i.e., the encoding distribution does not change as much if additional modalities are included. Table 4 shows the log-likelihood of the generative model and the value of the lower bound when the true data has private latent variables. Compared to the results in Table 1 with full decoder matrices, there appear to be smaller differences across different bounds and fusion schemes.

Finally, we consider permutation-equivariant schemes for learning models with private latent variables as detailed in Appendix F, applied to the setting with sparse variables in the data generation mechanism. Figure 5 shows different rate-distortion terms for $\beta \in \{0.1, 1, 4.\}$ for PoE and SumPooling and SelfAttention aggregation models. We find that our variational bound tends to obtain higher full reconstruction terms, while the full rates vary for different configurations. Conversely, the mixture-based bound obtains better cross-model reconstruction, with less clear patterns in the cross-rate terms. Table 5 shows the log-likelihood values for the learned generative model that is similar across different configurations, apart from a PoE scheme that achieves lower log-likelihood for a mixture-based bound.

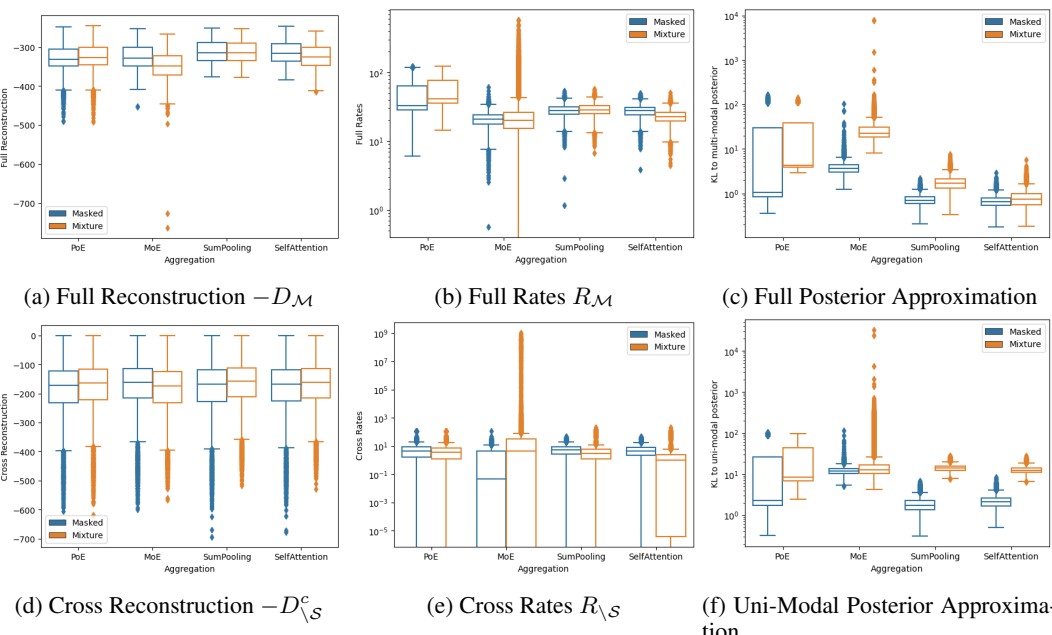

(a) Full Reconstruction $-D_{\mathcal{M}}$

(b) Full Rates $R_{\mathcal{M}}$

(c) Full Posterior Approximation

(d) Cross Reconstruction $-D_{\backslash S}^{c}$

(e) Cross Rates $R_{\backslash S}$

(f) Uni-Modal Posterior Approximation

Figure 3: Linear Gaussian models with dense decoder matrix: Rate and distortion terms and KL-divergence of encoding distributions to posterior distribution from learned generative model.

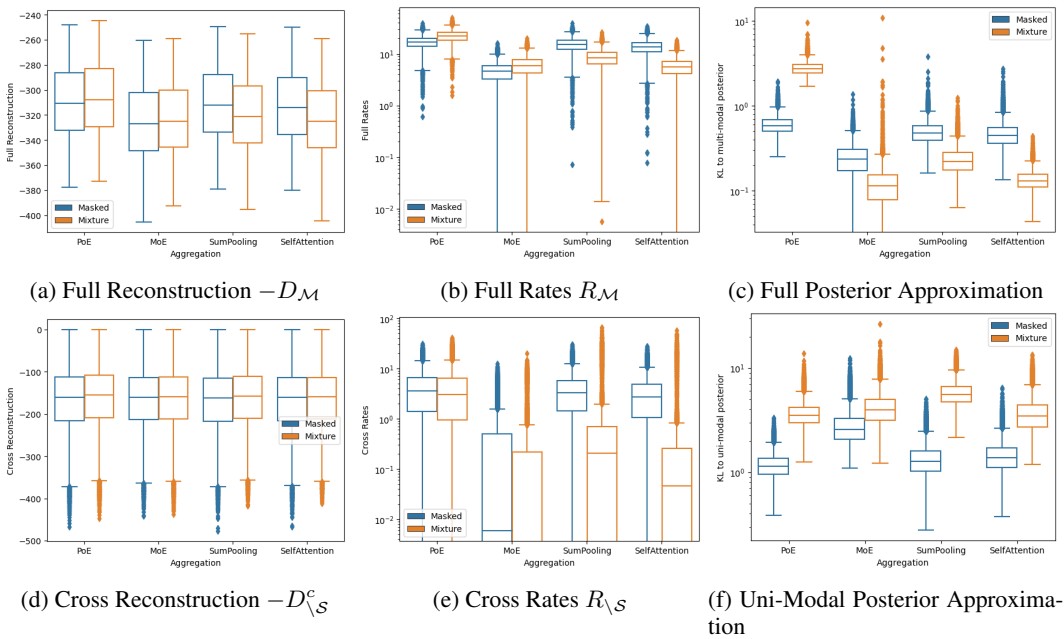

(a) Full Reconstruction $-D_{\mathcal{M}}$

(b) Full Rates $R_{\mathcal{M}}$

(c) Full Posterior Approximation

(d) Cross Reconstruction $-D_{\setminus \mathcal{S}}^{c}$

(e) Cross Rates $R_{\setminus \mathcal{S}}$

(f) Uni-Modal Posterior Approximation

Figure 4: Linear Gaussian models with sparse decoder matrix: Rate and distortion terms and KL-divergence of encoding distributions to posterior distribution from learned generative model.

Table 4: Multi-modal Gaussian model with sparse decoders in the ground truth model: LLH Gap is the relative difference of the log-likelihood of the learned model relative to the log-likelihood based on the exact MLE.

| | Our bound | | Mixture bound | |
|---|---|---|---|---|
| Aggregation | LLH Gap | MCC | LLH Gap | MCC |
| PoE | **0.00 (0.000)** | 0.84 (0.004) | 0.00 (0.007) | **0.87 (0.004)** |
| MoE | 0.01 (0.001) | 0.81 (0.001) | 0.01 (0.002) | 0.83 (0.003) |
| SumPooling | **0.00 (0.000)** | 0.84 (0.015) | 0.01 (0.001) | 0.84 (0.013) |
| SelfAttention | **0.00 (0.001)** | 0.84 (0.005) | 0.01 (0.002) | 0.83 (0.004) |

Table 5: Multi-modal Gaussian model with sparse decoders in the ground truth model and permutation-equivariant encoders: LLH Gap is the relative difference of the log-likelihood of the learned model relative to the log-likelihood based on the exact MLE.

| | Our bound | | Mixture bound | |
|---|---|---|---|---|
| Aggregation | LLH Gap | MCC | LLH Gap | MCC |
| PoE (equivariant) | **0.00 (0.000)** | **0.91 (0.016)** | 0.01 (0.001) | 0.88 (0.011) |
| SumPooling (equivariant) | **0.00 (0.000)** | 0.85 (0.004) | **0.00 (0.000)** | 0.82 (0.003) |
| SelfAttention (equivariant) | **0.00 (0.000)** | 0.83 (0.006) | **0.00 (0.000)** | 0.83 (0.003) |

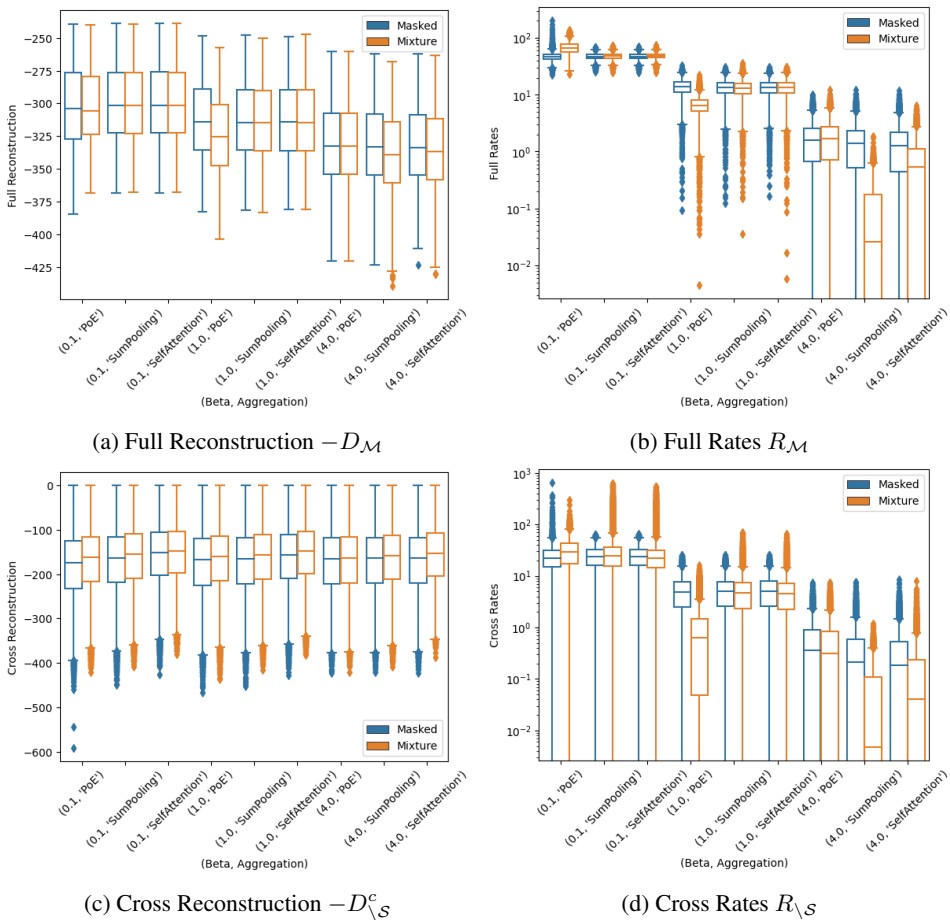

(a) Full Reconstruction $-D_{\mathcal{M}}$

(b) Full Rates $R_{\mathcal{M}}$

(c) Cross Reconstruction $-D^c_{\backslash \mathcal{S}}$

(d) Cross Rates $R_{\backslash \mathcal{S}}$

Figure 5: Linear Gaussian models with sparse decoder matrix and permutation-equivariant aggregation: Rate and distortion terms for varying $\beta$.

# N  NON-LINEAR IDENTIFIABLE MODELS

## N.1  AUXILIARY LABELS

Table 19 illustrates first the benefits of our bound that obtain better log-likelihood estimates for different fusion schemes. Second, it demonstrates the advantages of our new fusion schemes that achieve better log-likelihoods for both bounds. Third, it shows the benefit of using aggregation schemes that have the capacity to accommodate prior distributions different from a single Gaussian. Observe also that MoE schemes lead to low MCC values, while PoE schemes had high MCC values. We also show in Figure 6 the reconstructed modality values and inferred latent variables for one realisation with our bound, with the corresponding results for a mixture-based bound in Figure 7.

Table 6: Non-linear identifiable model with one real-valued modality and an auxiliary label acting as a second modality: The first four rows use a fixed standard Gaussian prior, while the last four rows use a Gaussian mixture prior with 5 components. Mean and standard deviation over 4 repetitions. Log-likelihoods are estimated using importance sampling with 64 particles.

| Aggregation | Our bound | | | Mixture bound | | |
| --- | --- | --- | --- | --- | --- | --- |
| | LLH ($\beta = 1$) | MCC ($\beta = 1$) | MCC ($\beta = 0.1$) | LLH ($\beta = 1$) | MCC ($\beta = 1$) | MCC ($\beta = 0.1$) |
| PoE | -43.4 (10.74) | 0.98 (0.006) | 0.99 (0.003) | -318 (361.2) | 0.97 (0.012) | 0.98 (0.007) |
| MoE | -20.5 (6.18) | 0.94 (0.013) | 0.93 (0.022) | -57.9 (6.23) | 0.93 (0.017) | 0.93 (0.025) |
| SumPooling | -17.9 (3.92) | 0.99 (0.004) | 0.99 (0.002) | -18.9 (4.09) | 0.99 (0.005) | 0.99 (0.008) |
| SelfAttention | -18.2 (4.17) | 0.99 (0.004) | 0.99 (0.003) | -18.6 (3.73) | 0.99 (0.004) | 0.99 (0.007) |
| SumPooling | **-15.4 (2.12)** | **1.00 (0.001)** | 0.99 (0.004) | -18.6 (2.36) | 0.98 (0.008) | 0.99 (0.006) |
| SelfAttention | **-15.2 (2.05)** | **1.00 (0.001)** | **1.00 (0.004)** | -18.6 (2.27) | 0.98 (0.014) | 0.98 (0.006) |
| SumPoolingMixture | **-15.1 (2.15)** | **1.00 (0.001)** | 0.99 (0.012) | -18.2 (2.80) | 0.98 (0.010) | 0.99 (0.005) |
| SelfAttentionMixture | **-15.3 (2.35)** | 0.99 (0.005) | 0.99 (0.004) | -18.4 (2.63) | 0.99 (0.007) | 0.99 (0.007) |

## N.2  FIVE CONTINUOUS MODALITIES

Table 7 demonstrates that our bound can yield to higher log-likelihoods and tigher bounds compared to a mixture-based bound, as do more flexible fusion schemes. Similar results for the partially observed case ($\eta = 0.5$) have been illustrated in the main text in Table 2.

Table 7: Fully observed ($\eta = 0$) non-linear identifiable model with 5 modalities: The first four rows use a fixed standard Gaussian prior, while the last four rows use a Gaussian mixture prior with 5 components. Mean and standard deviation over 4 repetitions.

| Aggregation | Our bound | | Mixture bound | |
| --- | --- | --- | --- | --- |
| | LLH | MCC | LLH | MCC |
| PoE | -473.6 (9.04) | 0.98 (0.005) | -497.7 (11.26) | 0.97 (0.008) |
| MoE | -477.9 (8.50) | 0.91 (0.014) | -494.6 (9.20) | 0.92 (0.004) |
| SumPooling | -471.4 (8.29) | **0.99 (0.004)** | -480.5 (8.84) | 0.98 (0.005) |
| SelfAttention | -471.4 (8.97) | **0.99 (0.002)** | -482.8 (10.51) | 0.98 (0.004) |
| SumPooling | **-465.4 (8.16)** | 0.98 (0.002) | -475.1 (7.54) | 0.98 (0.003) |
| SelfAttention | -469.3 (4.76) | 0.98 (0.003) | -474.7 (8.20) | 0.98 (0.002) |
| SumPoolingMixture | **-464.5 (8.16)** | **0.99 (0.003)** | -474.2 (7.61) | 0.98 (0.004) |
| SelfAttentionMixture | **-464.4 (8.50)** | **0.99 (0.003)** | -473.6 (8.24) | 0.98 (0.002) |

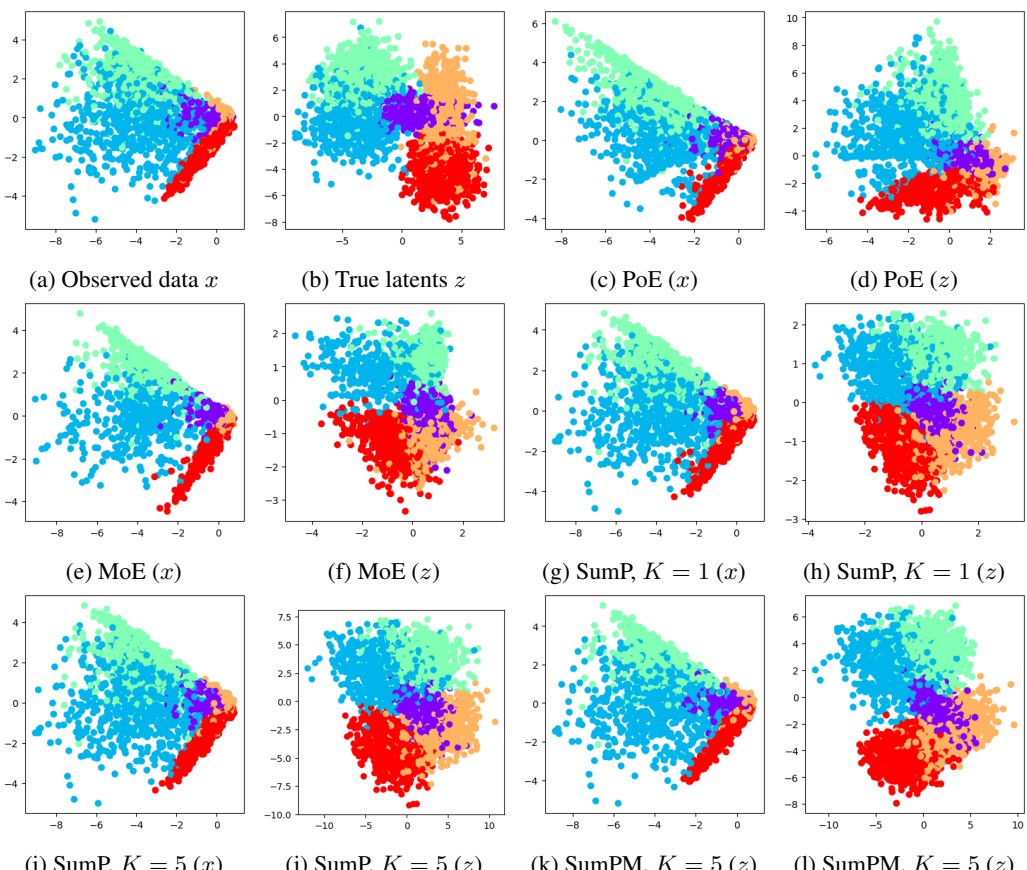

Figure 6: Bi-modal non-linear model with label and continuous modality based on our bound.

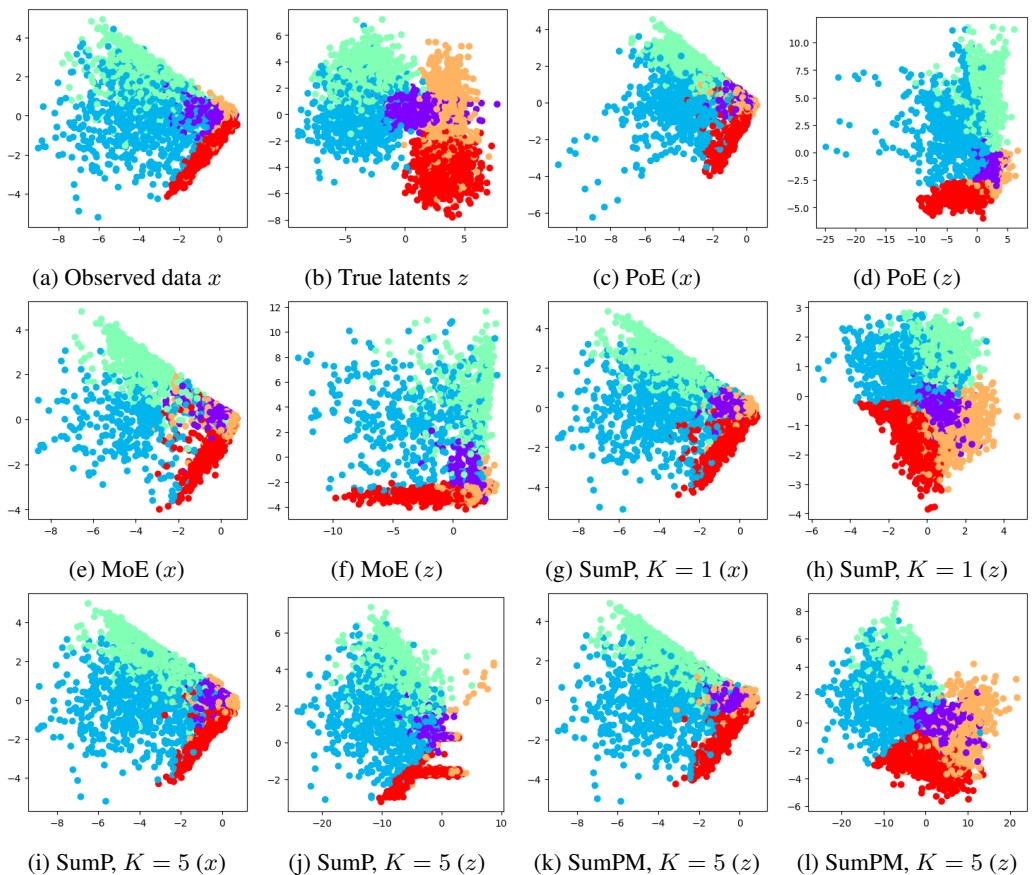

Figure 7: Bi-modal non-linear model with label and continuous modality based on mixture bound.

# O  MNIST-SVHN-TEXT

## O.1  TRAINING HYPERPARAMTERS

The MNIST-SVHN-Text data set is taken from the code accompanying Sutter et al. (2021) with around 1.1 million train and 200k test samples. All models are trained for 100 epochs with a batch size of 250 using Adam (Kingma and Ba, 2014) and a cosine decay schedule from 0.0005 to 0.0001.

## O.2  MULTI-MODAL RATES AND DISTORTIONS

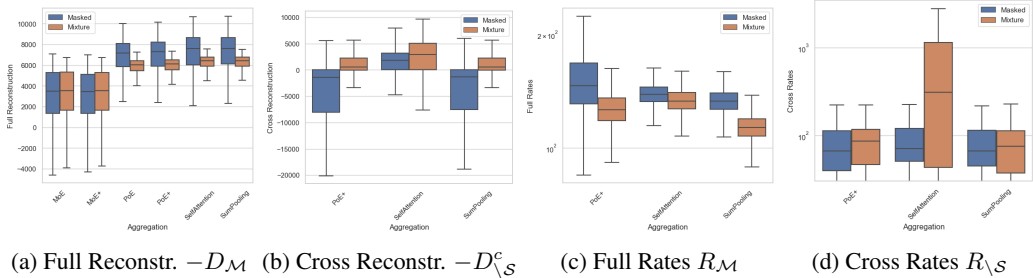

(a) Full Reconstr. $-D_{\mathcal{M}}$  (b) Cross Reconstr. $-D_{\backslash \mathcal{S}}^{c}$  (c) Full Rates $R_{\mathcal{M}}$  (d) Cross Rates $R_{\backslash \mathcal{S}}$

Figure 8: Rate and distortion terms for MNIST-SVHN-Text with shared and private latent variables.

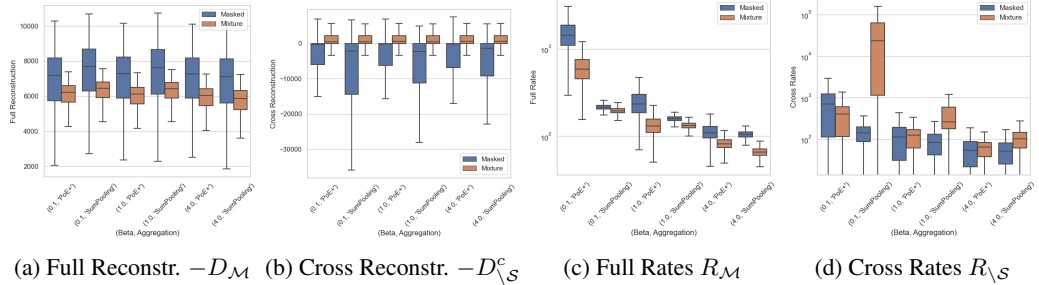

(a) Full Reconstr. $-D_{\mathcal{M}}$  (b) Cross Reconstr. $-D_{\backslash \mathcal{S}}^{c}$  (c) Full Rates $R_{\mathcal{M}}$  (d) Cross Rates $R_{\backslash \mathcal{S}}$

Figure 9: Rate and distortion terms for MNIST-SVHN-Text with shared latent variables and different $\beta$.

## O.3  LOG-LIKELIHOOD ESTIMATES

## O.4  GENERATED MODALITIES

## O.5  CONDITIONAL COHERENCE

**Latent classification accuracy.**

Table 8: Test log-likelihood estimates for varying $\beta$ choices for the joint data (M+S+T) as well as for the marginal data of each modality based on importance sampling (512 particles). Multimodal generative model with a 40-dimensional shared latent variable. The second part of the Table contains reported log-likelihood values from baseline methods that however impose more restrictive assumptions on the decoder variances which likely contributes to much lower log-likelihood values reported in previous works, irrespective of variational objectives and aggregation schemes.

| | Our bound | | | | Mixture bound | | | |
|---|---|---|---|---|---|---|---|---|
| ($\beta$, Aggregation) | M+S+T | M | S | T | M+S+T | M | S | T |
| (0.1, PoE+) | 5433 (24.5) | 1786 (41.6) | 3578 (63.5) | -29 (2.4) | 5481 (18.4) | 2207 (19.8) | 3180 (33.7) | -39 (1.0) |
| (0.1, SumPooling) | **7067 (78.0)** | 2455 (3.3) | **4701 (83.5)** | -9 (0.4) | 6061 (15.7) | 2398 (9.3) | 3552 (7.4) | -50 (1.9) |
| (1.0, PoE+) | 6872 (9.6) | 2599 (5.6) | 4317 (1.1) | -9 (0.2) | 5900 (10.0) | 2449 (10.4) | 3443 (11.7) | -19 (0.4) |
| (1.0, SumPooling) | 7056 (124.4) | 2478 (9.3) | 4640 (113.9) | -6 (0.0) | 6130 (4.4) | 2470 (10.3) | 3660 (1.5) | -16 (1.6) |
| (4.0, PoE+) | 7021 (13.3) | **2673 (13.2)** | 4413 (30.5) | **-5 (0.1)** | 5895 (6.2) | 2484 (5.5) | 3434 (2.2) | -13 (0.4) |
| (4.0, SumPooling) | 6690 (113.4) | 2483 (9.9) | 4259 (117.2) | **-5 (0.0)** | 5659 (48.3) | 2448 (10.5) | 3233 (27.7) | -10 (0.2) |
| Results from Sutter et al. (2021) and Sutter et al. (2020) | | | | | | | | |
| MVAE | -1790 (3.3) | NA | NA | NA | | | | |
| MMVAE | -1941 (5.7) | NA | NA | NA | | | | |
| MoPoE | -1819 (5.7) | NA | NA | NA | | | | |
| MMJSD | -1961 (NA) | NA | NA | NA | | | | |

(a) Our bound      (b) Mixture-based bound

Figure 10: Conditional generation for different aggregation schemes and bounds and shared latent variables. The first column is the conditioned modality. The next three columns are the generated modalities using a SumPooling aggregation, followed by the three columns for a SelfAttention aggregation, followed by PoE+ and lastly MoE+.

Table 9: Conditional coherence with shared latent variables and uni-modal inputs. The letters on the second line represent the generated modality based on the input modalities on the line below it.

| | Our bound | | | | | | | | | Mixture bound | | | | | | | | |
|---|---|---|---|---|---|---|---|---|---|---|---|---|---|---|---|---|---|---|
| | M | | | S | | | T | | | M | | | S | | | T | | |
| Aggregation | M | S | T | M | S | T | M | S | T | M | S | T | M | S | T | M | S | T |
| PoE | **0.97** | 0.22 | 0.56 | **0.29** | 0.60 | 0.36 | 0.78 | 0.43 | **1.00** | 0.96 | 0.83 | **0.99** | 0.11 | 0.57 | 0.10 | 0.44 | 0.39 | **1.00** |
| PoE+ | **0.97** | 0.15 | 0.63 | 0.24 | 0.63 | **0.42** | **0.79** | 0.35 | **1.00** | 0.96 | 0.83 | **0.99** | 0.11 | 0.59 | 0.11 | 0.45 | 0.39 | **1.00** |
| MoE | 0.96 | 0.80 | **0.99** | 0.11 | 0.59 | 0.11 | 0.44 | 0.37 | **1.00** | 0.94 | 0.81 | 0.97 | 0.10 | 0.54 | 0.10 | 0.45 | 0.39 | **1.00** |
| MoE+ | 0.93 | 0.77 | 0.95 | 0.11 | 0.54 | 0.10 | 0.44 | 0.37 | 0.98 | 0.94 | 0.80 | 0.98 | 0.10 | 0.53 | 0.10 | 0.45 | 0.39 | **1.00** |
| SumPooling | **0.97** | 0.48 | 0.87 | 0.25 | **0.72** | 0.36 | 0.73 | **0.48** | **1.00** | 0.97 | 0.83 | **0.99** | 0.10 | 0.63 | 0.10 | 0.45 | 0.40 | **1.00** |
| SelfAttention | **0.97** | 0.44 | 0.79 | 0.20 | 0.71 | 0.36 | 0.61 | 0.43 | **1.00** | **0.97** | **0.86** | **0.99** | 0.10 | 0.63 | 0.11 | 0.45 | 0.40 | **1.00** |
| Results from Sutter et al. (2021), Sutter et al. (2020) and Hwang et al. (2021) | | | | | | | | | | | | | | | | | | |
| MVAE | NA | 0.24 | 0.20 | 0.43 | NA | 0.30 | 0.28 | 0.17 | NA | | | | | | | | | |
| MMVAE | NA | 0.75 | 0.99 | 0.31 | NA | 0.30 | 0.96 | 0.76 | NA | | | | | | | | | |
| MoPoE | NA | 0.74 | 0.99 | 0.36 | NA | 0.34 | 0.96 | 0.76 | NA | | | | | | | | | |
| MMJSD | NA | 0.82 | 0.99 | 0.37 | NA | 0.36 | 0.97 | 0.83 | NA | | | | | | | | | |
| MVTCAE (w/o T) | NA | 0.60 | NA | 0.82 | NA | NA | NA | NA | NA | | | | | | | | | |

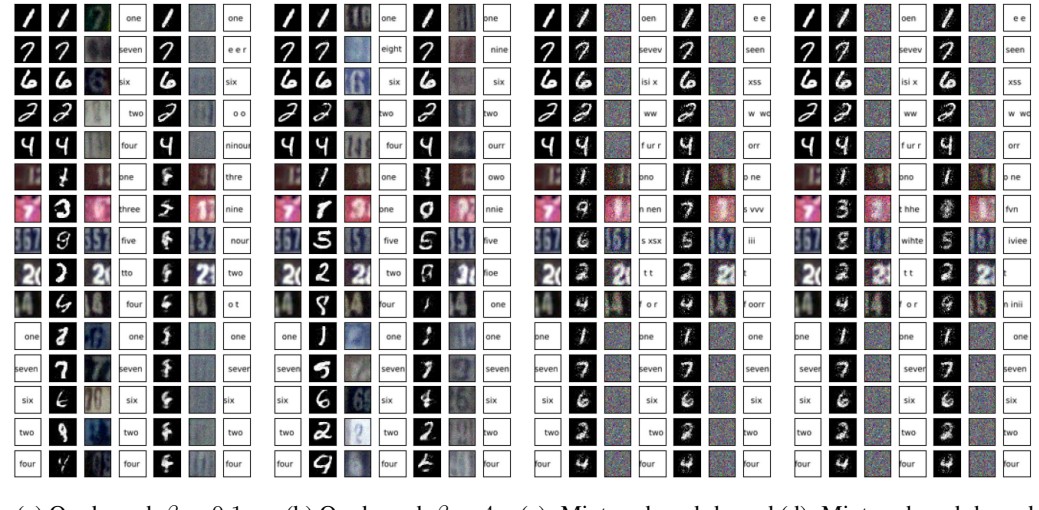

(a) Our bound, $\beta = 0.1$      (b) Our bound, $\beta = 4$     (c) Mixture-based bound, (d) Mixture-based bound, $\beta = 0.1$            $\beta = 4$

Figure 11: Conditional generation for different $\beta$ parameters. The first column is the conditioned modality. The next three columns are the generated modalities using a SumPooling aggregation, followed by the three columns for a PoE+ scheme.

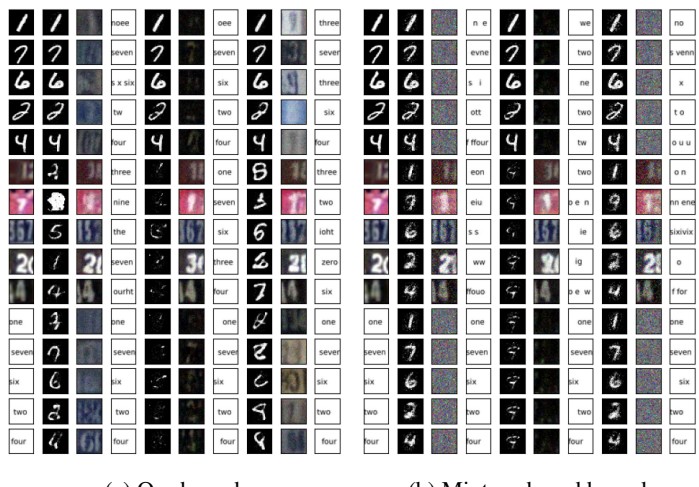

(a) Our bound               (b) Mixture-based bound

Figure 12: Conditional generation for permutation-equivariant schemes and private latent variable constraints. The first column is the conditioned modality. The next three columns are the generated modalities using a SumPooling aggregation, followed by the three columns for a SelfAttention scheme and a PoE model.

Table 10: Conditional coherence for models with shared latent variables and bi-modal conditionals. The letters on the second line represent the modality which is generated based on the sets of modalities on the line below it.

| | Our bound | | | | | | | | | Mixture bound | | | | | | | | |
| | M | | | S | | | T | | | M | | | S | | | T | | |
| Aggregation | M+S | M+T | S+T | M+S | M+T | S+T | M+S | M+T | S+T | M+S | M+T | S+T | M+S | M+T | S+T | M+S | M+T | S+T |
|---|---|---|---|---|---|---|---|---|---|---|---|---|---|---|---|---|---|---|
| PoE | **0.98** | **0.98** | 0.60 | 0.75 | **0.58** | 0.77 | 0.82 | **1.00** | **1.00** | 0.96 | 0.97 | 0.95 | 0.61 | 0.11 | 0.61 | 0.45 | 0.99 | 0.98 |
| PoE+ | 0.97 | **0.98** | 0.55 | 0.73 | 0.52 | 0.75 | **0.83** | **1.00** | 0.99 | 0.97 | 0.97 | **0.96** | 0.64 | 0.11 | 0.63 | 0.45 | 0.99 | 0.97 |
| MoE | 0.88 | 0.97 | 0.90 | 0.35 | 0.11 | 0.35 | 0.41 | 0.72 | 0.69 | 0.88 | 0.96 | 0.89 | 0.32 | 0.10 | 0.33 | 0.42 | 0.72 | 0.69 |
| MoE+ | 0.85 | 0.94 | 0.86 | 0.32 | 0.10 | 0.32 | 0.40 | 0.71 | 0.67 | 0.87 | 0.96 | 0.89 | 0.32 | 0.10 | 0.32 | 0.42 | 0.72 | 0.69 |
| SumPooling | 0.97 | 0.97 | 0.86 | 0.78 | 0.30 | **0.80** | 0.76 | 0.99 | **1.00** | 0.97 | 0.97 | 0.95 | 0.65 | 0.10 | 0.65 | 0.45 | 0.99 | 0.97 |
| SelfAttention | 0.97 | 0.97 | 0.82 | **0.76** | 0.30 | 0.78 | 0.69 | **1.00** | **1.00** | 0.97 | 0.97 | 0.99 | 0.66 | 0.10 | 0.65 | 0.45 | 0.99 | **1.00** |
| Results from Sutter et al. (2021), Sutter et al. (2020) and Hwang et al. (2021) | | | | | | | | | | | | | | | | | | |
| MVAE | NA | NA | 0.32 | NA | 0.43 | NA | 0.29 | NA | NA | | | | | | | | | |
| MMVAE | NA | NA | 0.87 | NA | 0.31 | NA | 0.84 | NA | NA | | | | | | | | | |
| MoPoE | NA | NA | 0.94 | NA | 0.36 | NA | 0.93 | NA | NA | | | | | | | | | |
| MMJSD | NA | NA | 0.95 | NA | 0.48 | NA | 0.92 | NA | NA | | | | | | | | | |
| MVTCAE (w/o T) | NA | NA | NA | NA | NA | NA | NA | NA | NA | | | | | | | | | |

Table 11: Conditional coherence for models with private latent variables and uni-modal conditionals. The letters on the second line represent the modality which is generated based on the sets of modalities on the line below it.

| | Our bound | | | | | | | | | Mixture bound | | | | | | | | |
| | M | | | S | | | T | | | M | | | S | | | T | | |
| Aggregation | M | S | T | M | S | T | M | S | T | M | S | T | M | S | T | M | S | T |
|---|---|---|---|---|---|---|---|---|---|---|---|---|---|---|---|---|---|---|
| PoE+ | 0.97 | 0.12 | 0.13 | 0.20 | 0.62 | 0.24 | 0.16 | 0.15 | 1.00 | 0.96 | 0.83 | **0.99** | 0.11 | 0.58 | 0.11 | 0.44 | 0.39 | 1.00 |
| SumPooling | 0.97 | 0.42 | 0.59 | **0.44** | 0.67 | **0.40** | **0.65** | **0.45** | 1.00 | 0.97 | **0.86** | **0.99** | 0.11 | 0.62 | 0.11 | 0.45 | 0.40 | 1.00 |
| SelfAttention | 0.97 | 0.12 | 0.12 | 0.27 | **0.71** | 0.28 | 0.46 | 0.40 | 1.00 | 0.96 | 0.09 | 0.08 | 0.12 | 0.67 | 0.12 | 0.15 | 0.17 | 1.00 |

Table 12: Conditional coherence for models with private latent variables and bi-modal conditionals. The letters on the second line represent the modality which is generated based on the sets of modalities on the line below it.

| | Our bound | | | | | | | | | Mixture bound | | | | | | | | |
| | M | | | S | | | T | | | M | | | S | | | T | | |
| Aggregation | M+S | M+T | S+T | M+S | M+T | S+T | M+S | M+T | S+T | M+S | M+T | S+T | M+S | M+T | S+T | M+S | M+T | S+T |
|---|---|---|---|---|---|---|---|---|---|---|---|---|---|---|---|---|---|---|
| PoE+ | **0.97** | **0.97** | 0.14 | 0.66 | 0.33 | 0.67 | 0.18 | **1.00** | **1.00** | **0.97** | **0.97** | **0.94** | 0.63 | 0.11 | 0.63 | 0.45 | 0.99 | 0.96 |
| SumPooling | **0.97** | **0.97** | 0.54 | 0.79 | **0.43** | 0.80 | **0.57** | **1.00** | **1.00** | **0.97** | **0.97** | 0.93 | 0.64 | 0.11 | 0.63 | 0.45 | 0.99 | 0.97 |
| SelfAttention | **0.97** | **0.97** | 0.12 | **0.80** | 0.29 | **0.81** | 0.49 | **1.00** | **1.00** | 0.96 | 0.96 | 0.08 | 0.70 | 0.12 | 0.70 | 0.15 | **1.00** | **1.00** |

Table 13: Conditional coherence for models with shared latent variables for different βs and uni-modal conditionals. The letters on the second line represent the modality which is generated based on the sets of modalities on the line below it.

| | Our bound | | | | | | | | | Mixture bound | | | | | | | | |
| | M | | | S | | | T | | | M | | | S | | | T | | |
| (β, Aggregation) | M | S | T | M | S | T | M | S | T | M | S | T | M | S | T | M | S | T |
|---|---|---|---|---|---|---|---|---|---|---|---|---|---|---|---|---|---|---|
| (0.1, PoE+) | **0.98** | 0.11 | 0.12 | 0.12 | 0.62 | 0.14 | 0.61 | 0.25 | **1.00** | 0.96 | 0.83 | **0.99** | 0.11 | 0.58 | 0.11 | 0.45 | 0.39 | **1.00** |
| (0.1, SumPooling) | 0.97 | 0.48 | 0.81 | 0.30 | **0.72** | 0.33 | **0.86** | 0.55 | **1.00** | 0.97 | 0.86 | **0.99** | 0.11 | 0.64 | 0.11 | 0.45 | 0.40 | **1.00** |
| (1.0, PoE+) | 0.97 | 0.15 | 0.63 | 0.24 | 0.63 | 0.42 | 0.79 | 0.35 | **1.00** | 0.96 | 0.83 | **0.99** | 0.11 | 0.59 | 0.11 | 0.45 | 0.39 | **1.00** |
| (1.0, SumPooling) | 0.97 | 0.48 | 0.87 | 0.25 | **0.72** | 0.36 | 0.73 | 0.48 | **1.00** | 0.97 | 0.86 | **0.99** | 0.10 | 0.63 | 0.10 | 0.45 | 0.40 | **1.00** |
| (4.0, PoE+) | 0.97 | 0.29 | 0.83 | **0.41** | 0.60 | **0.58** | 0.76 | 0.38 | **1.00** | 0.96 | 0.82 | **0.99** | 0.10 | 0.57 | 0.10 | 0.44 | 0.38 | **1.00** |
| (4.0, SumPooling) | 0.97 | 0.48 | 0.88 | 0.35 | 0.66 | 0.44 | 0.83 | 0.53 | **1.00** | 0.96 | 0.85 | **0.99** | 0.11 | 0.57 | 0.10 | 0.45 | 0.39 | **1.00** |

Table 14: Conditional coherence for models with shared latent variables for different βs and bi-modal conditionals. The letters on the second line represent the modality which is generated based on the sets of modalities on the line below it.

| | Our bound | | | | | | | | | Mixture bound | | | | | | | | |
| | M | | | S | | | T | | | M | | | S | | | T | | |
| (β, Aggregation) | M+S | M+T | S+T | M+S | M+T | S+T | M+S | M+T | S+T | M+S | M+T | S+T | M+S | M+T | S+T | M+S | M+T | S+T |
|---|---|---|---|---|---|---|---|---|---|---|---|---|---|---|---|---|---|---|
| (0.1, PoE+) | **0.98** | **0.98** | 0.15 | 0.70 | 0.14 | 0.72 | 0.66 | **1.00** | **1.00** | 0.96 | 0.96 | 0.93 | 0.62 | 0.11 | 0.62 | 0.45 | 0.99 | 0.95 |
| (0.1, SumPooling) | 0.97 | 0.97 | 0.86 | **0.83** | 0.31 | **0.84** | 0.85 | 0.99 | **1.00** | 0.97 | 0.97 | 0.94 | 0.66 | 0.11 | 0.65 | 0.45 | 0.99 | 0.96 |
| (1.0, PoE+) | 0.97 | **0.98** | 0.55 | 0.73 | 0.52 | 0.75 | 0.83 | **1.00** | 0.99 | 0.97 | 0.97 | **0.96** | 0.64 | 0.11 | 0.63 | 0.45 | 0.99 | 0.97 |
| (1.0, SumPooling) | 0.97 | 0.97 | 0.86 | 0.78 | 0.30 | 0.80 | 0.76 | 0.99 | **1.00** | 0.97 | 0.97 | 0.95 | 0.65 | 0.10 | 0.65 | 0.45 | 0.99 | 0.97 |
| (4.0, PoE+) | 0.97 | **0.98** | 0.84 | 0.76 | **0.66** | 0.78 | 0.82 | **1.00** | **1.00** | 0.97 | 0.97 | **0.96** | 0.62 | 0.10 | 0.62 | 0.45 | 0.99 | 0.98 |
| (4.0, SumPooling) | 0.97 | 0.97 | 0.89 | 0.77 | 0.40 | 0.78 | **0.86** | 0.99 | **1.00** | 0.97 | 0.97 | **0.96** | 0.61 | 0.10 | 0.60 | 0.45 | 0.99 | 0.97 |

Table 15: Unsupervised latent classification for $\beta = 1$ and models with shared latent variables only (top half) and shared plus private latent variables (bottom half). Accuracy is computed with a linear classifier (logistic regression) trained on multi-modal inputs (M+S+T) or uni-modal inputs (M, S or T).

| | Our bound | | | | Mixture bound | | | |
|---|---|---|---|---|---|---|---|---|
| Aggregation | M+S+T | M | S | T | M+S+T | M | S | T |
| PoE | 0.988 (0.000) | 0.940 (0.009) | 0.649 (0.039) | 0.998 (0.001) | 0.991 (0.004) | 0.977 (0.002) | 0.845 (0.000) | **1.000 (0.000)** |
| PoE+ | 0.978 (0.002) | 0.934 (0.001) | 0.624 (0.040) | 0.999 (0.001) | **0.998 (0.000)** | 0.981 (0.000) | 0.851 (0.000) | **1.000 (0.000)** |
| MoE | 0.841 (0.008) | 0.974 (0.000) | 0.609 (0.032) | **1.000 (0.000)** | 0.940 (0.001) | 0.980 (0.001) | 0.843 (0.001) | **1.000 (0.000)** |
| MoE+ | 0.850 (0.039) | 0.967 (0.014) | 0.708 (0.167) | 0.983 (0.023) | 0.928 (0.017) | 0.983 (0.002) | 0.846 (0.001) | **1.000 (0.000)** |
| SelfAttention | 0.985 (0.001) | 0.954 (0.002) | 0.693 (0.037) | 0.986 (0.006) | 0.991 (0.000) | 0.981 (0.001) | 0.864 (0.003) | **1.000 (0.000)** |
| SumPooling | 0.981 (0.000) | 0.962 (0.000) | 0.704 (0.014) | 0.992 (0.008) | 0.994 (0.000) | **0.983 (0.000)** | 0.866 (0.002) | **1.000 (0.000)** |
| PoE+ | 0.979 (0.009) | 0.944 (0.000) | 0.538 (0.032) | 0.887 (0.07) | 0.995 (0.002) | 0.980 (0.002) | 0.848 (0.006) | **1.000 (0.000)** |
| SumPooling | 0.987 (0.004) | 0.966 (0.004) | 0.370 (0.348) | 0.992 (0.002) | 0.994 (0.001) | 0.982 (0.000) | **0.870 (0.001)** | **1.000 (0.000)** |
| SelfAttention | 0.990 (0.003) | 0.968 (0.002) | 0.744 (0.004) | 0.985 (0.004) | 0.997 (0.001) | 0.974 (0.000) | 0.681 (0.031) | **1.000 (0.000)** |
| Results from Sutter et al. (2021), Sutter et al. (2020) and Hwang et al. (2021) | | | | | | | | |
| MVAE | 0.96 (0.02) | 0.90 (0.01) | 0.44 (0.01) | 0.85 (0.10) | | | | |
| MMVAE | 0.86 (0.03) | 0.95 (0.01) | 0.79 (0.05) | 0.99 (0.01) | | | | |
| MoPoE | 0.98 (0.01) | 0.95 (0.01) | 0.80 (0.03) | 0.99 (0.01) | | | | |
| MMJSD | 0.98 (NA) | 0.97 (NA) | 0.82 (NA) | 0.99 (NA) | | | | |
| MVTCAE (w/o T) | NA | 0.93 (NA) | 0.78 (NA) | NA | | | | |

Table 16: Unsupervised latent classification for different $\beta$s and models with shared latent variables only. Accuracy is computed with a linear classifier (logistic regression) trained on multi-modal inputs (M+S+T) or uni-modal inputs (M, S or T).

| | Our bound | | | | Mixture bound | | | |
|---|---|---|---|---|---|---|---|---|
| ($\beta$, Aggregation) | M+S+T | M | S | T | M+S+T | M | S | T |
| (0.1, PoE+) | 0.983 (0.006) | 0.919 (0.001) | 0.561 (0.048) | 0.988 (0.014) | 0.992 (0.002) | 0.979 (0.002) | 0.846 (0.004) | **1.000 (0.000)** |
| (0.1, SumPooling) | 0.982 (0.004) | 0.965 (0.002) | 0.692 (0.047) | 0.999 (0.001) | 0.994 (0.000) | 0.981 (0.002) | 0.863 (0.005) | **1.000 (0.000)** |
| (1.0, PoE+) | 0.978 (0.002) | 0.934 (0.001) | 0.624 (0.040) | 0.999 (0.001) | **0.998 (0.000)** | 0.981 (0.000) | 0.851 (0.000) | **1.000 (0.000)** |
| (1.0, SumPooling) | 0.981 (0.000) | 0.962 (0.000) | 0.704 (0.014) | 0.992 (0.008) | 0.994 (0.000) | **0.983 (0.000)** | **0.866 (0.002)** | **1.000 (0.000)** |
| (4.0, PoE+) | 0.981 (0.006) | 0.943 (0.007) | 0.630 (0.008) | 0.993 (0.001) | **0.998 (0.000)** | 0.981 (0.000) | 0.846 (0.001) | **1.000 (0.000)** |
| (4.0, SumPooling) | 0.984 (0.004) | 0.963 (0.001) | 0.681 (0.009) | 0.995 (0.000) | 0.992 (0.002) | 0.980 (0.001) | 0.856 (0.001) | **1.000 (0.000)** |

# P ENCODER MODEL ARCHITECTURES

## P.1 LINEAR MODELS

Table 17: Encoder architectures for Gaussian models.

(a) Modality-specific encoding functions $h_s(x_s)$. Latent dimension $D = 30$, modality dimension $D_s \sim \mathcal{U}(30, 60)$.

| MoE/PoE | SumPooling/SelfAttention |
|---|---|
| Input: $D_s$ | Input: $D_s$ |
| Dense $D_s \times 512$, ReLU | Dense $D_s \times 256$, ReLU |
| Dense $512 \times 512$, ReLU | Dense $256 \times 256$, ReLU |
| Dense $512 \times 60$ | Dense $256 \times 60$ |

(b) Model for outer aggregation function $\rho_\vartheta$ for SumPooling and SelfAttention schemes.

| Outer Aggregation |
|---|
| Input: 256 |
| Dense $256 \times 256$, ReLU |
| Dense $256 \times 256$, ReLU |
| Dense $256 \times 60$ |

(c) Inner aggregation function $\chi_\vartheta$.

| SumPooling | SelfAttention |
|---|---|
| Input: 256 | Input: 256 |
| Dense $256 \times 256$, ReLU | Dense $256 \times 256$, ReLU |
| Dense $256 \times 256$, ReLU | Dense $256 \times 256$ |
| Dense $256 \times 256$ | |

(d) Transformer parameters.

| SelfAttention (1 Layer) |
|---|
| Input: 256 |
| Heads: 4 |
| Attention size: 256 |
| Hidden size FFN: 256 |

## P.2 LINEAR MODELS WITH PRIVATE LATENT VARIABLES

Table 18: Encoder architectures for Gaussian models with private latent variables.

(a) Modality-specific encoding functions $h_s(x_s)$. All private and shared latent variables are of dimension 10. Modality dimension $D_s \sim \mathcal{U}(30, 60)$.

| PoE ($h_s^{\text{shared}}$ and $h_s^{\text{private}}$) | SumPooling/SelfAttention (one $h_s$) |
|---|---|
| Input: $D_s$ | Input: $D_s$ |
| Dense $D_s \times 512$, ReLU | Dense $D_s \times 128$, ReLU |
| Dense $512 \times 512$, ReLU | Dense $128 \times 128$, ReLU |
| Dense $512 \times 10$ | Dense $128 \times 10$ |

(b) Model for outer aggregation function $\rho_\vartheta$ for SumPooling scheme.

| Outer Aggregation ($\rho_\vartheta$) |
|---|
| Input: 128 |
| Dense $128 \times 128$, ReLU |
| Dense $128 \times 128$, ReLU |
| Dense $128 \times 10$ |

(c) Inner aggregation functions.

| SumPooling ($\chi_{0,\vartheta}, \chi_{1,\vartheta}, \chi_{2,\vartheta}$) | SelfAttention ($\chi_{1,\vartheta}, \chi_{2,\vartheta}$) |
|---|---|
| Input: 128 | Input: 128 |
| Dense $128 \times 128$, ReLU | Dense $128 \times 128$, ReLU |
| Dense $128 \times 128$, ReLU | Dense $128 \times 128$ |
| Dense $128 \times 128$ | |

(d) Transformer parameters.

| SelfAttention (1 Layer) |
|---|
| Input: 128 |
| Heads: 4 |
| Attention size: 128 |
| Hidden size FFN: 128 |

## P.3 NONLINEAR MODEL WITH AUXILIARY LABEL

## P.4 NONLINEAR MODEL WITH FIVE MODALITIES

## P.5 MNIST-SVHN-TEXT

For SVHN and and Text, we use 2d- or 1d-convolutional layers, respectively, denoted as Conv$(f, k, s)$ for feature dimension $f$, kernel-size $k$ and stride $s$. We denote transposed convolutions as tConv. We use the neural network architectures as implemented in Flax Heek et al. (2023).

Table 19: Encoder architectures for nonlinear model with auxiliary label.

(a) Modality-specific encoding functions $h_s(x_s)$. Modality dimension $D_1 = 2$ (continuous modality) and $D_2 = 5$ (label). Embedding dimension $D_E = 4$ for PoE and MoE and $D_E = 128$ otherwise.

| Modality-specific encoders |
| --- |
| Input: $D_s$ |
| Dense $D_s \times 128$, ReLU |
| Dense $128 \times 128$, ReLU |
| Dense $128 \times D_E$ |

(b) Model for outer aggregation function $\rho_\vartheta$ for SumPooling and SelfAttention schemes and mixtures thereof. Output dimension is $D_0 = 25$ for mixture densities and $D_O = 4$ otherwise.

| Outer Aggregation |
| --- |
| Input: 128 |
| Dense $128 \times 128$, ReLU |
| Dense $128 \times 128$, ReLU |
| Dense $128 \times D_O$ |

(c) Inner aggregation function $\chi_\vartheta$.

| SumPooling | SelfAttention |
| --- | --- |
| Input: 128 | Input: 128 |
| Dense $128 \times 128$, ReLU | Dense $128 \times 128$, ReLU |
| Dense $128 \times 128$, ReLU | Dense $128 \times 128$ |
| Dense $128 \times 128$ | |

(d) Transformer parameters.

| SelfAttention |
| --- |
| Input: 128 |
| Heads: 4 |
| Attention size: 128 |
| Hidden size FFN: 128 |

Table 20: Encoder architectures for nonlinear model with five modalities.

(a) Modality-specific encoding functions $h_s(x_s)$. Modality dimensions $D_s = 25$. Latent dimension $D = 25$

| MoE/PoE | SumPooling/SelfAttention |
| --- | --- |
| Input: $D_s$ | Input: $D_s$ |
| Dense $D_s \times 512$, ReLU | Dense $D_s \times 256$, ReLU |
| Dense $512 \times 512$, ReLU | Dense $256 \times 256$, ReLU |
| Dense $512 \times 50$ | Dense $256 \times 256$ |

(b) Model for outer aggregation function $\rho_\vartheta$ for SumPooling and SelfAttention schemes and mixtures thereof. Output dimension is $D_0 = 50$ for mixture densities and $D_O = 25$ otherwise.

| Outer Aggregation |
| --- |
| Input: 256 |
| Dense $256 \times 256$, ReLU |
| Dense $256 \times 256$, ReLU |
| Dense $256 \times D_O$ |

(c) Inner aggregation function $\chi_\vartheta$.

| SumPooling | SelfAttention |
| --- | --- |
| Input: 256 | Input: 256 |
| Dense $256 \times 256$, ReLU | Dense $256 \times 256$, ReLU |
| Dense $256 \times 256$, ReLU | Dense $\times 256$ |
| Dense $256 \times 256$ | |

(d) Transformer parameters.

| SelfAttention |
| --- |
| Input: 256 |
| Heads: 4 |
| Attention size: 256 |
| Hidden size FFN: 256 |

P.6 MNIST-SVHN-TEXT WITH PRIVATE LATENT VARIABLES

# Q MNIST-SVHN-TEXT DECODER MODEL ARCHITECTURES

For models with private latent variables, we concatenate the shared and private latent variables. We use a Laplace likelihood as the decoding distribution for MNIST and SVHN, where the decoder function learns both its mean as a function of the latent and a constant log-standard-deviation at each pixel. Following previous works (Shi et al., 2019; Sutter et al., 2021), we re-weight the log-likelihoods for different modalities relative to their dimensions.

Table 21: Encoder architectures for MNIST-SVHN-Text.

(a) MNIST-specific encoding functions $h_s(x_s)$. Modality dimensions $D_s = 28 \times 28$. Embedding dimension is $D_E = 2D$ for PoE/MoE and $D_E = 256$ for SumPooling/SelfAttention. For PoE+/MoE+, we add four times a Dense layer of size 256 with ReLU layer before the last linear layer.

| MoE/PoE/SumPooling/SelfAttention |
| --- |
| Input: $D_s$, |
| Dense $D_s \times 400$, ReLU |
| Dense $400 \times 400$, ReLU |
| Dense $400 \times D_E$ |

(b) SVHN-specific encoding functions $h_s(x_s)$. Modality dimensions $D_s = 3 \times 32 \times 32$. Embedding dimension is $D_E = 2D$ for PoE/MoE and $D_E = 256$ for SumPooling/SelfAttention. For PoE+/MoE+, we add four times a Dense layer of size 256 with ReLU layer before the last linear layer.

| MoE/PoE/SumPooling/SelfAttention |
| --- |
| Input: $D_s$ |
| Conv(32, 4, 2), ReLU |
| Conv(64, 4, 2), ReLU |
| Conv(64, 4, 2), ReLU |
| Conv(128, 4, 2), ReLU, Flatten |
| Dense $2048 \times D_E$ |

(c) Text-specific encoding functions $h_s(x_s)$. Modality dimensions $D_s = 8 \times 71$. Embedding dimension is $D_E = 2D$ for PoE/MoE and $D_E = 256$ for permutation-invariant models (SumPooling/SelfAttention) and $D_E = 128$ for permutation-equivariant models (SumPooling/SelfAttention). For PoE+/MoE+, we add four times a Dense layer of size 256 with ReLU layer before the last linear layer.

| MoE/PoE/SumPooling/SelfAttention |
| --- |
| Input: $D_s$ |
| Conv(128, 1, 1), ReLU |
| Conv(128, 4, 2), ReLU |
| Conv(128, 4, 2), ReLU, Flatten |
| Dense $128 \times D_E$ |

(d) Model for outer aggregation function $\rho_\vartheta$ for SumPooling and SelfAttention schemes. Output dimension is $D_O = 2D = 80$ for models with shared latent variables only and $D_O = 10 + 10$ for models with private and shared latent variables. $D_E = 256$ for permutation-invariant and $D_I = 128$ for permutation-invariant models.

| Outer Aggregation |
| --- |
| Input: $D_E$ |
| Dense $D_E \times D_E$, LReLU |
| Dense $D_E \times D_E$, LReLU |
| Dense $D_E \times D_O$ |

(e) Inner aggregation function $\chi_\vartheta$ for permutation-invariant models ($D_E = 256$) and permutaion-equivariant models ($D_E = 128$).

| SumPooling | SelfAttention |
| --- | --- |
| Input: $D_E$ | Input: $D_E$ |
| Dense $D_E \times D_E$, LReLU | Dense $D_E \times D_E$, LReLU |
| Dense $D_E \times D_E$, LReLU | Dense $\times D_E$ |
| Dense $D_E \times D_E$ | |

(f) Transformer parameters for permutation-invariant models. $D_E = 256$ for permutation-invariant and $D_I = 128$ for permutation-invariant models.

| SelfAttention (2 Layers) |
| --- |
| Input: $D_E$ |
| Heads: 4 |
| Attention size: $D_E$ |
| Hidden size FFN: $D_E$ |

# R  COMPUTE RESOURCES AND EXISTING ASSETS

Our computations were performed on shared HPC systems. All experiments except Section 5.3 were run on a CPU server using one or two CPU cores. The experiments in Section 5.3 were run a GPU server using one NVIDIA A100.

Our implementation is based on JAX (Bradbury et al., 2018) and Flax (Heek et al., 2023). We compute the mean correlation co-efficient (MCC) between true and inferred latent variables following Khemakhem et al. (2020b), as in https://github.com/ilkhem/icebeem and follow the data and model generation from Khemakhem et al. (2020a), https://github.com/ilkhem/iVAE in Section 5.2, as well as https://github.com/hanmenghan/CPM_Nets from Zhang et al. (2019) for generating the missingness mechanism. In our MNIST-SVHN-Text experiments, we use code from Sutter et al. (2021), https://github.com/thomassutter/MoPoE.

Table 22: Decoder architectures for MNIST-SVHN-Text.

(a) MNIST decoder. $D_I = 40$ for models with shared latent variables only, and $D_I = 10 + 10$ otherwise.

| MNIST |
| --- |
| Input: $D_I$ |
| Dense $40 \times 400$, ReLU |
| Dense $400 \times 400$, ReLU |
| Dense $400 \times D_s$, Sigmoid |

(b) SVHN decoder. $D_I = 40$ for models with shared latent variables only, and $D_I = 10 + 10$ otherwise.

| SVHN |
| --- |
| Input: $D_I$ |
| Dense $D_I \times 128$, ReLU |
| tConv(64, 4, 3), ReLU |
| tConv(64, 4, 2), ReLU |
| tConv(32, 4, 2), ReLU |
| tConv(3, 4, 2) |

(c) Text decoder. $D_I = 40$ for models with shared latent variables only, and $D_I = 10+10$ otherwise.

| Text |
| --- |
| Input: $D_I$ |
| Dense $D_I \times 128$, ReLU |
| tConv(128, 4, 3), ReLU |
| tConv(128, 4, 2), ReLU |
| tConv(71, 1, 1) |

