# OpenReview forum: "Learning multi-modal generative models with permutation-invariant encoders and tighter variational bounds"
_ICLR.cc/2024/Conference — Submitted to ICLR 2024_

### Official Review · Reviewer_yUvm · 2023-10-23

**Soundness:** 2 fair
**Presentation:** 3 good
**Contribution:** 2 fair
**Rating:** 3
**Confidence:** 4

**Summary:**

This work describes a new variational bound to train multimodal VAE models and proposes to use two new aggregation functions rather than the mixture of experts (MoE) and product of experts (PoE) that have been typically used in the literature. The proposed variational bound is claimed to address the known limitations of previous variational bounds, while the new aggregation functions should help in learning identifiable models, and achieve higher log-likelihood.

After an introduction to the multimodal generative model problem and a discussion the multimodal VAE literature, the authors describe a  new variational lower bound that is used as a proxy to learn model parameters. This bound is computed using two non overlapping subsets, randomly sampled at each training step. The bound consists of the sum of the conditional bound and a marginal bound of these two subsets, respectively.  In other words, this approach is a generalization to the bound derived in [1].
This bound is claimed to become tighter than the ones from the literature, provided that the encoding distribution closely approximates the true posterior distribution. The authors then discuss an information theoretic perspective of the proposed bound.

Then, new aggregation functions are presented, whereby PoE and MoE are considered as fixed, special cases, and two new,  learnable, aggregation functions, which are permutation invariant, are proposed: (i) Sum pooling, (ii) Transformer encoder.

The authors also discuss the unidentifiability of nonlinear models. It's claimed that the proposed bound generalises some identifiability properties that have been studied in the context of unimodal VAEs, to the case of multiple modalities. Then the authors discuss alternatives to the choice of uni-modal prior densities via Gaussian mixture priors.

Experimental validation of proposed approach and the new aggregation functions consists in a comparison to vanilla PoE and MoE methods: first, on a synthetic dataset, by considering both linear and non-linear problems, then, using a real world dataset, MINIST-SVHN-Text, where the performance of the proposed approach is studied.

[1] M. Wu and N. Goodman. Multimodal generative models for scalable weakly-supervised learning. Advances in neural information processing systems, 31, 2018.

**Strengths:**

- This work targets an important problem, which is related to multimodal generative modeling. It proposes new aggregation functions, e.g. attention-based functions, which are an interesting addition to the current methods to aggregate latent modalities into a single latent space.

- The study of identifiability properties of multimodal models is, to the best of the reviewer's knowledge, novel and important.

- The editorial quality of the paper is very good, albeit some related work are missing.

**Weaknesses:**

- The work does not directly demonstrate that the proposed bound is tighter than previously reported bounds in the literature. It's not clear how the new bound overcomes the multimodal VAE limitation detailed in [2], which are related to sub-sampling of modalities, which has the problematic effect of inducing an undesirable upper bound.

- The well known limitations from Multimodal VAE literature consist of the undesired trade-off between generation coherence and quality. Additionally, the modality collapse problem due to gradient conflicts [3] can also affect the training process. These limitations are not discussed, although the paper claims that the new proposed bound solves the limitations of Multimodal VAE.

- Some works are missing from the literature review, such as [4].

- The results on the MNIST-SVHN-Text dataset, using the proposed approach, do not seem to improve the overall conditional generation coherence. LLH improvements do not correlate with the overall conditional coherence / latent classification accuracy. In table 9, the cross generation of SVHN given MNIST is slightly better using the proposed approach but the generation of MNIST given SVHN is a lot worse. The proposed extensions do not perform very well across all modalities in terms of coherence: some recent works obtain far superior results with respect to the multimodal VAE literature [5].

- The experimental campaign misses many competitors and benchmarks that are widely used in the literature: for example, competitors such as  Nexus [4],  MMVAE+ [6], MOPOE [7], MVTCAE [8] are not included as baselines.

- The evaluation protocol adopted in most of the related literature sheds light on two desired properties of multimodal generative models which are generative coherence and quality. Measuring the generation quality is not considered in this work. Joint generation (unconditional generation) by sampling from the prior is one setting in which, in general, Multimodal VAEs fail. Results on joint generation are not included.

- The author considers only one real world dataset containing highly correlated modalities, with the label digit (Text modality). Other benchmarks widely studied in the literature are missing from the experimental section: for example, the Polymnist [7], MHD [4], CUB [2, 9] datasets. This latter showed to be a challenging case for VAE models [2].

[2] I. Daunhawer, T. M. Sutter, K. Chin-Cheong, E. Palumbo, and J. E. Vogt. On the limitations of multimodal VAEs. In International Conference on Learning Representations, 2022.

[3] A. Javaloy, M. Meghdadi, and I. Valera. Mitigating modality collapse in multi-modal VAEs via impartial optimization. In International Conference on Machine Learning, 2022.

[4] M. Vasco, H. Yin, F. S. Melo, and A. Paiva. Leveraging hierarchy in multimodal generative models for effective cross-modality inference. In Neural Networks, 2022.

[5] M. Bounoua, G. Franzese, and P. Michiardi. Multi-modal latent diffusion. arXiv preprint arXiv:2306.04445, 2023.

[6] E. Palumbo, I. Daunhawer, and J. E. Vogt. MMVAE+: Enhancing the generative quality of multimodal VAEs without compromises. In ICLR Workshop on Deep Generative Models for Highly Structured Data, 2022.

[7] T. M. Sutter, I. Daunhawer, and J. E. Vogt. Generalized multimodal ELBO. In International Conference on Learning Representations, 2021.

[8] H. Hwang, G.-H. Kim, S. Hong, and K.-E. Kim. Multi-view representation learning via total correlation objective. In Advances in Neural Information Processing Systems, 2021.

[9] Shi, Yuge, Brooks Paige, and Philip Torr. Variational mixture-of-experts autoencoders for multi-modal deep generative models. In Advances in neural information processing systems, 2019.

**Questions:**

- Why the proposed bound is tighter than the bound studied in the larger family of multimodal VAE models? Is this an empirical claim via LLH estimation?

- How does the new bound overcome the limitations of Multimodal VAEs, such as the undesirable upper bound due to sub-sampling of modalities? In practice, the new proposed bound is also based on a similar sub-sampling approach as in [1]

- How does the new bound and the presented aggregation methods handle other limitations of Multimodal VAEs, such as the coherence-quality trade-off and modality collapse where stronger modalities dominate over weaker ones [3]?

- The evaluation of the generative quality of the compared models using metrics such as the FID score should be considered. Joint generation results (sampling from the prior) should be reported to evaluate if the new approach helps in this challenging scenario.

- Several missing baselines from the multimodal generative models literature should be included in the experimental comparison: VAE based models such as Nexus,  MMVAE+, MOPOE, MVTCAE are typical baselines (please, see references above).

- The experimental section should be enriched with more challenging datasets, such as Polymnist, MHD, and CUB datasets (please, see references above).

---

> ### Author Response · Authors · 2023-11-22
> **Response to review**
>
> Thank you for your comprehensive review and valuable feedback on our paper. Your comments were extremely helpful for our updated submission.
>
> > Well-known limitations from Multimodal VAE (i.e., undesired trade-off between generation coherence and quality; gradient conflicts) are not discussed, although the paper claims that the new proposed bound solves the limitations of Multimodal VAE.
>
> We realized that we formulated our contributions in an unintended strong form as (i) a new variational bound that addresses known limitations of previous variational bounds.
> We have updated it now to the below, which we believe is supported by our analyses:
> (i) a new variational bound as an approximate lower bound on the multi-modal log-likelihood (LLH). We avoid a limitation of mixture-based bounds, which may not provide tight lower bounds on the joint LLH if there is considerable modality-specific variation (Daunhawer et al., 2022), even for flexible encoding distributions. The novel variational bound contains a lower bound of the marginal LLH $\log p_\theta(x_\mathcal{S})$ and a term approximating the conditional $\log p_\theta(x_{\setminus \mathcal{S}}|x_\mathcal{S})$ for any choice of $\mathcal{S} \in \mathcal{P}(\mathcal{M})$, provided that we can learn a flexible multimodal encoding distribution.
>
> In particular, we do not want to claim that our approach solves all well-known limitations of multimodal VAEs, see also our comments further below. For example, we illustrate that other variational bounds can be beneficial for cross-modal reconstructions. Due to space constraints, we have only mentioned the issue of gradient conflicts in Appendix A that also applies to our work. While we did not test this empirically, we still expect that multi-task learning techniques can be similarly effective for our approach, as they are for other variational bounds or aggregation schemes.
>
>
> > It's not clear how the new bound overcomes the multimodal VAE limitation detailed in [2].
>
> We thank the reviewer for pointing out that it was unclear how our bound overcomes the multimodal VAE limitation of inducing an undesirable lower bound as in [2]. We have now adjusted the discussion and proof in the updated manuscript to better address this. To be more specific, for the mixture based variational objective, one obtains the bound
> $$ \int p_d(x) \log p_\theta(x)  d x \geq \int p_d(x) L^{\text{Mix}}(x, \theta, \phi, 1) d x  +\mathcal{H} (p_d(X_{\setminus \mathcal{S}}|X_\mathcal{S})) .$$
> Notice that there exists a gap between the variational bound and the log-likelihood given by the conditional entropies that cannot be reduced even for flexible encoding distributions. Conversely, our variational objective satisfies
> $$ \int p_d(x) \log p_\theta(x)  d x  = \int p_d(x) L^{\text{Ours}}(x, \theta, \phi, 1) d x  +K_1+K_2+C,$$
> where
> $$ K_1= \int p_d(x_\mathcal{S})  \left[KL(q_\phi(z|x_\mathcal{S})|p_{\theta}(z|x_\mathcal{S})) \right]  d x, $$
> $$K_2= \int p_d(x)  \left[KL(q_\phi(z|x)|p_{\theta}(z|x)) \right]  d x $$
> and
> $$C = - \int p_d(x)  q_\phi(z|x )\left[\log \frac{q_\phi(z|x_\mathcal{S})}{p_{\theta}(z|x_\mathcal{S})} \right]  dz dx .$$
> C is not necessarily positive. See Remarks  12 and 13 for details.
> We thus overcome the undesirable upper bound by contending ourselves to maximize an approximate lower bound. However, in the limiting case of infinite-capacity encoding, the variational gap can be zero, whilst it is a  lower bound if the encoding distribution satisfies some consistency condition that will likely not hold in practice.
>
> > Multimodal VAEs fail in joint generation. Recent diffusion-based work [5] obtains superior results compared to VAEs
>
> We did not intend to claim that our approach leads to better conditional/unconditional generation or coherence than recent/concurrent work using diffusion models [5]. Indeed, for a single modality, our bound becomes a standard VAE, and thus shares its limitations, particularly concerning prior-hole issues that may lead to relatively poor unconditional generation compared to diffusion models, for example. Moreover, our ELBO analysis demonstrates that a mismatch between the aggregated posterior/prior conditioned on a modality subset and the encoding distribution conditioned on said modality subset may likewise lead to sub-optimal conditional coherence. While this mismatch may be smaller for more flexible aggregation functions, we believe this issue applies generally to other multimodal VAEs, unless one incorporates different sampling procedures, such as diffusions, EBMs, etc., in some form. We will clarify this in an updated version.

---

> > ### Author Response · Authors · 2023-11-22
> > **Response to review (cont)**
> >
> > >  Several missing baselines from the multimodal generative models' literature should be included in the experimental comparison: VAE-based models such as Nexus, MMVAE+, MOPOE, MVTCAE.
> >
> >
> > We do agree with the reviewer that our submission lacked a direct comparison and discussion how baseline methods perform on the SVHN-MNSIT-Text dataset. To address this, we have now included the results from previous works using the same dataset. In particular, we report log-likelihoods, conditionally generated coherences and latent classification accuracies for external baselines (MVAE, MMVAE, MoPoE, MMJSD, MVTCAE) in Tables 8-10 and 15. Unlike our initially reported results, different hyperparameter choices in these baselines (different latent dimensions, encoder/decoder architectures, prior/approximate posterior distribution families, etc.) may make a direct comparison more difficult.
> >
> >
> > > The experimental section should be enriched with more challenging datasets, such as Polymnist, MHD, and CUB datasets. The evaluation of the generative quality of the compared models using metrics such as the FID score should be considered.
> >
> > We acknowledge that it is not clear how our approach would perform on more challenging datasets and with respect to different generative measures. Since FID scores can be less suited for the MNIST and Text modality, we only considered log-likelihoods scores. We will definitely consider the application to more challenging datasets, as mentioned by you, although we feel that our submission in its current form already contains enough contributions for a conference proceeding.

---

> > > ### Comment · Reviewer_yUvm · 2023-11-22
> > > **ACK - 2**
> > >
> > > I understand you focus on LLH as a metric, which is in general a reasonable choice, especially for simple datasets. I also understand that results from previous work (which in general publish their code) are not easy to compare to your results, given the choice of hyper parameters and several additional "tricks" (such as importance sampling, peaked priors, ...). Of course, it would not be feasible to do that in the timeframe of a rebuttal, but a possibly sound approach here would have been to re-implement all the baselines, by striping out all bells and whistles, and compare to alternatives with a large number of datasets (not necessarily going for high-res, billion sample datasets, but using those adopted in the literature, which are more manageable).
> > >
> > > So while I really appreciate the effort to add tables to compare to alternative approaches, as you correctly assessed, these results do not really provide strong evidence about the merit of your work, because they are not easy to compare.
> > >
> > > I will remain open in the discussion phase to listen to other reviewers position (some of them are very positive) and will support any decision we will converge to. In the meanwhile, however, I will not modify my score.

---

> > ### Comment · Reviewer_yUvm · 2023-11-22
> > **ACK - 1**
> >
> > Dear Authors,
> > thank you for your answers, which contribute to a more suitable positioning of your work with respect to the literature. Thank you also for the clarifications on the properties of the derived lower bound.
> >
> > The concern that remains for me is that the current state of the art (see references below), already points at the limitations of the previously known bounds, and propose very effective methods (MMVAE+) to overcome them. While I still think your ideas are valuable, and I thank you for the effort in improving their exposition, in its current form, your article does not contain sufficient empirical evidence of their benefit for the main objectives of multimodal (joint and conditional) generation. [See also new comments below]
> >
> > I. Daunhawer, T. M. Sutter, K. Chin-Cheong, E. Palumbo, and J. E. Vogt. On the limitations of multimodal VAEs. In International Conference on Learning Representations, 2022.
> >
> > MMVAE+: Enhancing the Generative Quality of Multimodal VAEs without Compromises. In International Conference on Learning Representations, 2023.

---

### Official Review · Reviewer_vMeL · 2023-10-29

**Soundness:** 3 good
**Presentation:** 2 fair
**Contribution:** 2 fair
**Rating:** 5
**Confidence:** 3

**Summary:**

The article builds upon previous work on Variational Auto-Encoder models designed to work on data with multiple modalities (such as text, images, etc...), on the modelling hypothesis that the different modalities are independently generated from a single common latent representation. The core task that is used to train those models is thus to recover said latent representation from a subset of the observed modalities, and use that to reconstruct the missing modalities.

The paper builds upon a previous bound developed in the restricted context of 2 modalities (Wu and Goodman, 2019), and generalizes to an arbitrary number of them, and an arbitrary observed subset. Alongside this, the paper uses permutation-invariant neural architectures (Zaheer et al, 2017) to build probabilistic encoders that can work on arbitrary subsets of observed modalities, while having more expressive power than the previously-used Mixture-of-Experts and Product-of-Experts constructions.

The resulting architecture is experimentally evaluated on several combinations of encoder architecture and training bounds to evaluate the impact of those two contributions, showing a general improvement to the reconstruction quality.

**Strengths:**

The main contribution of the paper seems theoretically solid: the bound is properly derived with explicit and reasonable assumptions, and provides a promising framework for formulating the training of multi-modal latent variable models such as VAEs. The proposed construction brings together several ideas from the wider literature in a justified and sound way.

The 2 main contributions are extensively experimentally evaluated, both independently and combined, to previous models on the same data and tasks. Detailed quantitative results are provided, covering a large range of tasks and metrics (reconstruction, cross-reconstruction, utilization of the latent representation for downstream classification, ...), illustrating their contextual strengths and weaknesses fairly.

**Weaknesses:**

I find the general structure of the paper to be imbalanced: a large portion of the paper is used to reference and restate the literature in a way that in my opinion does not bring much, while most results and observations are stated and barely discussed, even in the extensive appendix.

Listing the points that should in my opinion be expanded:

1) Proposition 1 relies strongly on the assumption on the structure of $q_\phi$ given at the top of page 4 (without being clearly expressed as an assumption), which amounts to stating that the partial encoder should be identical to the empirical marginalization of the full encoder:
$$
q_\phi(z|x_\mathcal{S}) = \int_{x_{\setminus\mathcal{S}}} p_d(x_{\setminus\mathcal{S}} | x_{\mathcal{S}}) q_\phi(z | x_{\mathcal{S}}, x_{\setminus\mathcal{S}}) \text{d}x_{\setminus\mathcal{S}}
$$
While the text below proposition 1 suggests that the training loss encourages the model to bring these two distributions close to each other, there is no discussion about this assumption.

2) At the end of section 2, it is shown that the mathematical optimal partial encoder $q^\star(z|x_\mathcal{S})$ of the mixture-based objective is not equal to the true Bayesian posterior of the model $p(z|x_\mathcal{S})$ (as opposed to the proposed objective), but instead an additional factor is added to it. Unfortunately, this additional factor and how it may impact the model is not discussed at all.

3) In the experimental results of the MINST-SVHN-Text problem, a lot of quantitative performance measures are provided, but they are barely discussed or interpreted.

Also, I have a few more minor points regarding phrasing and clarity:

4) Section 2 makes heavy use of inline math for rather long expressions. While I understand the space constraint, this makes these paragraphs difficult to parse.

5) Lemma 2 and Corollary 3 use the names $Z_\mathcal{M}$ and $Z_\mathcal{S}$ without them being first defined. While we can guess what they are probably supposed to mean, this hurts readability.

6) The legend of Figure 2 is very small and difficult to read without zooming a lot. Furthermore, it uses the name "Masked" to (I suppose) refer to the proposed bound, while this name is not used anywhere else in the paper.

**Questions:**

My questions directly mirror points that are in my opinion under-discussed:

Regarding the assumption about the structure of $q_\phi$ being compatible with empirical marginalization:
- Is the proposed loss still a proper lower-bound if this assumption is not respected?
- If yes, why is this assumption needed? If not, why is it not a problem?

Regarding the experimental results on MINST-SVHN-Text:
- Why are the model performance so different between the various modalities?
- Why does adding private latent variables seem to overall reduce the performance of the models in terms of conditional coherence (see tables 9-12)?
- Figure 2b suggests that the Mixture loss yields better cross-reconstruction overall, while tables 9-12 suggest the opposite, is there a contradiction?

---

> ### Author Response · Authors · 2023-11-22
> **Response to review**
>
> We thank the reviewer for the detailed review and very insightful comments, which were invaluable in enhancing the clarity and presentation of the submission.
>
> > Proposition 1 relies strongly on the assumption on the structure of $q_\phi$  given at the top of page 4 (without being clearly expressed as an assumption)
>
> We agree that Proposition 1 relies on a strong assumption about the encoding distribution that has not been properly discussed. We have, therefore, removed Proposition 1 in the updated version. Note that the ELBO surgery in Proposition 11 (or Proposition 9 in the updated version) does not make this assumption, which we are now using to show how the proposed objective relates to the multi-modal log-likelihood and under what conditions it constitutes a lower bound. More precisely, see Corollary 1, the variational gap includes, beyond the two KL-divergences between the variational approximation and the true posterior conditioned on $c_{\mathcal{S}}$ and on the full $x$, an additional term
> $\int p_d(x)  q_\phi(z|x )\left[\log \frac{q_\phi(z|x_\mathcal{S})}{p_{\theta}(z|x_\mathcal{S})} \right]  dz dx $
> that is not necessarily negative or negative. As you already commented, in the case where the encoding distribution conditioned on $x_{\mathcal{S}}$ equals the empirical marginalization of the full encoder, or what we called the aggregated encoder conditioned on $x_\mathcal{S}$, the variational gap reduces to the KL conditioned on the full multi-modal data, i.e., we recover the previous Proposition while more clearly stating the required conditions. See Remarks 12 and 13 in Appendix A for details.
>
> We have also added clarifications regarding this to the updated submission. In the experiments with simulated data, we have removed the reported gap between the variational bounds and the log-likelihoods as it may not say much without assumptions on the encoding distributions.
>
> > Is the proposed loss still a proper lower-bound if this assumption is not respected? If yes, why is this assumption needed? If not, why is it not a problem?
>
> The proposed loss is only an approximately lower bound to the multi-modal LLH without additional assumptions on the encoding distributions. Experimentally, this did not lead to divergence issues or so. The ‘approximately lower’ issue is due to the $L_{\setminus \mathcal{S}}$ loss term, which is basically also used in latent Neural Processes, and where we believe it does also not cause optimization issues. Furthermore, our multi-modal ELBO decomposition shows that by maximixing $L_{\setminus \mathcal{S}}$, we minimize the above-mentioned Bayes-consistency matching term between the encoding distribution and its aggregated version.
>
> In the general case with arbitrary $\beta >0$, see Corollary 3, minimizing our bound (properly) minimizes multi-modal rate and distortion measures that are different from those minimized in mixture-based bounds and that provide sandwich bounds for conditional and unconditional mutual information.
>
> > Why are the model performance so different between the various modalities?
>
> We believe the performance differences for the modalities in the MNIST-SVHN-Text data are because SVHN is more noisy than MNIST or Text. Previous work for the same dataset (MVAE, MMVAE, MoPoE, MMJSD, MVTCAE) similarly exhibited differences in the performance measures (see, for example, Table 9 for coherence measures, updated with results as reported in previous work).
>
> > Why does adding private latent variables seem to overall reduce the performance of the models in terms of conditional coherence (see tables 9-12)?
>
> Introducing private latent variables may lead to different inductive biases for the learned latent representations and their conditional reconstruction. In the MNIST-SVHN-Text experiment, we fixed the overall latent dimension to 40. While there were no restrictions for the models with shared variables, for the model with private latent variables, we assumed that both the shared latent variables and the three private latent variables are 10-dimensional each. This creates a bottleneck in how much shared information can be encoded.

---

> > ### Author Response · Authors · 2023-11-22
> > **Response to review (cont)**
> >
> > > Figure 2b suggests that the Mixture loss yields better cross-reconstruction overall, while tables 9-12 suggest the opposite, is there a contradiction?
> >
> > In Figure 2b, we report the cross-reconstruction based on the log-likelihood of each modality. Since mixture-based bounds directly optimize for this objective (which we call cross-distortion), they do, indeed, perform better here. Tables 9-12 measure a cross-coherence measure based on the classification performance of a pre-trained classifier (trained on the true modalities) applied to the generated modalities. The cross-coherence can vary a lot between modalities and may depend on how well the pre-trained classifier generalizes to reconstructed modalities. For example, cross-generations of the MNIST modality appear more averaged with less variation leading to a better coherence for the mixture-based bound.
> >
> > > Lemma 2 and Corollary 3 use the names ZM and ZS without them being first defined.
> >
> > Thank you for pointing this out. We will include their definitions in the main paper.
> >
> >
> > > In the experimental results of the MINST-SVHN-Text problem, a lot of quantitative performance measures are provided, but they are barely discussed or interpreted.
> >
> > We aim to include a more detailed discussion of the quantitative performance measures in an updated version.
> >
> > > The optimal partial encoder $q_\phi(z|x_\mathcal{S})$ of the mixture-based objective is not equal to the true Bayesian posterior [...] and how it may impact the model is not discussed at all.
> >
> > Thank you for pointing out this interesting question. To shed light on how this might impact the model performance, we refer to the toy example in our response to reviewer EVr5 of two modalities $X_1=Z_0+Z_1+U_1$ and $X_1=Z_0+10Z_2+U_2$  driven by a latent variable $Z=(Z_0,Z_1,Z_2) \in \mathbb{R}^3$, where $Z_i$ and $U_j$ are iid $\mathcal{N}(0,1)$. In this setup, we can compute the KL to the true posterior, and we use relatively flexible aggregation schemes (MLP with 64 hidden dimensions of depth 2; possible 64-dim attention; for 3-dim latent variable), hoping to approximate the optimal (partial and full) encoders closely. Indeed, for both bounds, we find that the average KLs between the full encoders and full posterior is small, while the KL between the uni-modal encoders and posteriors becomes small only for our bound.
> >
> > |               | uni-modal KL     |                  | multi-modal KL   |                  |
> > |---------------|------------------|------------------|------------------|------------------|
> > |               | ours             | mixture          | ours             | mixture          |
> > | SumPooling    |           0.008  |           0.789  |           0.013  |           0.015  |
> > | SelfAttention |           0.007  |           0.791  |           0.012  |           0.015  |
> >
> > However, the model performance is different for these two bounds. For example, our bound achieves better LLH (analytically computed using only the learned generative parameters). The information-theoretical measures of the learned representations are very different, see the Table in the response to reviewer EVr5. For the MNIST-SVHN-Text example, where the learned encoders likely differ more from their optimal counterparts, we see similar effects in Figure 2.

---

### Official Review · Reviewer_pr6d · 2023-11-03

**Soundness:** 1 poor
**Presentation:** 2 fair
**Contribution:** 1 poor
**Rating:** 3
**Confidence:** 4

**Summary:**

The authors propose a method to train a multi-modal VAE.
The authors focus on permutation-invariant encoder which is invariant to the modalities order.
The authors demonstrate on a simple dataset that the proposed method works and explore different model configurations.

**Strengths:**

* The authors discusses an important area of research - multi-modal generative models.

**Weaknesses:**

* The novelty is rather limited (the aggregation schemes are marginally novel, and the invariance is trivially achieved using known architectures).
* The experimentation over a single dataset is not satisfactory.
* Comparison to competing models is missing.
* Optimization using MCMC typically limits the scaling of the proposed method from being used for more complex data (e.g., high dimensional) and architectures (i.e., more expressive models).

**Questions:**

* Have you tried scaling the method to higher dimensional and more complex data?
* Did you encounter any issues and challenges during optimization?
* What is the optimization procedure? What is the final loss? Can you provide a clear pseudo code?
* Did you compare the proposed model to external baselines?

---

> ### Author Response · Authors · 2023-11-22
> **Response to review**
>
> We thank the reviewer for the review and comments.
>
> >"Optimization using MCMC typically limits the scaling of the proposed method "
>
> We are unsure why MCMC scaling properties can constitute a weakness of our work. We do not use any MCMC in this submission.
>
> >"The novelty is rather limited (the aggregation schemes are marginally novel, and the invariance is trivially achieved using known architectures)."
>
> We agree with the reviewer that the premutation-invariant architectures are not new and that previous schemes such as MoEs, PoE, or MoPoEs do trivially inherit such an invariance. However, we do not think that permutation-invariant architectures per se represent the contribution of our submission. In contrast, we believe that our contributions entail:
>
> i)            A new multi-modal variational bound, accompanied by detailed analysis (e.g. information-theoretical analysis, multi-modal ELBO surgery).
>
> ii)            The application of flexible fusion/aggregation schemes for multi-modal VAEs. While the network architectures are not new, we believe that our application for multi-modal models with shared or private latent variables is novel.
>
> iii)            Model extensions such as using mixture priors (which are not that easily accommodated with PoE schemes) and identifiable models (where we believe training objectives that maximize the log-likelihood are very beneficial).
>
> > "The experimentation over a single dataset is not satisfactory."
>
> We want to clarify that we included numerical experiments with different bounds and aggregation schemes for various datasets; and not just on a single dataset. In particular, our experimentations include:
>
> i)             Different linear models
>
> ii)            Different non-linear models on synthetic data: 2D toy-example, higher dimensional examples with five modalities, in a complete setup, and with missing modalities. This design allows for access to ground truth values (as required for identifiability analysis)
>
> iii)           MNIST-SVHN-Text data
>
> > "What is the optimization procedure? What is the final loss? Can you provide a clear pseudo code?"
>
> We use Adam as an optimizer. Pseudo-code is given in Appendix K, see Algorithm 1 for models with shared latent variables. The final (negative) loss is the sum of (3) and (4), cf. Algorithm 1.
>
> > "Did you compare the proposed model to external baselines?"
>
> Thank you for pointing this out. We agree with the reviewer that our submission lacked a direct comparison and discussion of how baseline methods perform, for example on the SVHN-MNSIT-Text dataset. To address this, we have now included the results from previous works using the same dataset. In particular, we have updated  Tables 8-10 and 15 with log-likelihoods, conditionally generated coherences, and latent classification accuracies from external baselines (MVAE, MMVAE, MoPoE, MMJSD, MVTCAE). However, unlike our initially reported results, different hyperparameter choices in these baselines (different latent dimensions, encoder/decoder architectures, prior/approximate posterior distribution families, etc.) may make a direct comparison more difficult.

---

### Official Review · Reviewer_dord · 2023-11-08

**Soundness:** 3 good
**Presentation:** 3 good
**Contribution:** 3 good
**Rating:** 6
**Confidence:** 3

**Summary:**

The authors propose a new variational lower bound for VAEs in the multimodal setting.  The lower bound leverages a permutation invariance property and the resulting architecture seems to yield performance improvements in several experimental settings.

**Strengths:**

The paper is clearly written and the limitations of this work and related works are well-articulated.

The contribution appears to be novel, and it nearly always results in models with improved LLs/bounds in the provided experiments.

This application of invariance is novel to me and may be of broader interest in latent variable modeling problems.

**Weaknesses:**

Many of the technical details have been relegated to the discussion section.  While this is somewhat common practice, there are, in my opinion, a lot of details in the supplementary material.  Taking everything into account:  there seems to be more than enough here to constitute a journal submission.   Cramming this into a conference paper makes it difficult for reviewers to adequately assess the correctness.

While this approach does yield improvements with respect to LLs, it comes at a higher computational cost.  In addition, while I know that LL scores are often focused on in these kinds of works, I often find that LL can provide a misleading picture of the model, especially in continuous settings.  From that perspective, it would have been nice to see more task specific results, e.g., inference, etc.

I don't really understand why permutation invariance, specifically, is a good thing here.  Why not other forms of symmetries?  I think I would benefit from some higher level intuition here.  It seems to be central to the approach, but I couldn't tease its importance out of the presentation.

**Questions:**

See above.

---

> ### Author Response · Authors · 2023-11-22
> **Response to review**
>
> Thank you for your thoughtful review and constructive feedback on our paper.
>
> > LL can provide a misleading picture of the model
>
> We completely agree that achieving a high LLH is not always the primary aim for generative models. We think it is central when looking, for instance, at model identifiability. Our new information-theoretic multi-modal rate-distortion analysis suggests - theoretically and experimentally - that different variational bounds are designed to optimize different information quantities. These rationales, along with $\beta$ hyper-parameters, should be kept in mind for a given downstream task. While such rate-distortion analyses have received much attention in the uni-modal setup, we feel that multi-modal extensions of such analyses can provide a complimentary assessment to other multi-modal evaluations (such as coherence measures based on pre-trained classifiers). For example, when one wants to obtain good cross-modal reconstruction, but not necessarily cross-modal generation, we suggest using previous mixture-based bounds from the literature instead of our bound. However, permutation-invariant aggregation schemes can still be beneficial for other variational objectives and can act as replacements for PoEs, MoEs, or MoPoEs therein.
>
> > Why permutation invariance, specifically, is a good thing here. Why not other forms of symmetries?
>
> Our primary motivation for considering permutation-invariant architectures was computational efficiency in that it allows us to model exponentially many functions (for all modality subsets, scaling exponentially in the number of modalities) via a single parametric family based on popular architectures, such as DeepSets or SetAttention. Other neural network architectures allowing for efficient inferences on sets might also be suitable, such as approximately permutation-invariant models via RNNs [1]. Such invariant aggregation schemes are applied after applying a modality-specific encoding functions. These initial modality-specific encoding functions could respect other modality-specific invariances or symmetries. One can also choose modality-specific decoders that respect different invariances. Intuitively, given a set of two modalities $X_1$ and $X_2$, say, we want to learn modality-specific initial encodings $H_1$ and $H_2$ such that the encoding distribution does not change if we reorder $H_1$ and $H_2$. Note that the posterior distribution $p(z|x_1,x_2)$ is invariant with respect to the order of $x_1$ and $x_2$ in the sense that one gets the same posterior when one first updates the prior using the observation $x_1$ to arrive at $p(z|x_1)$ and then updates this with $x_2$, or proceeds first with using $x_2$ and then using $x_1$. However, learning flexible parametric families that satisfy this form of Bayes-consistency is non-trivial, and our architectures may not respect this property exactly, and neither do previous multi-modal aggregation schemes like MoEs, (non-tempered) PoEs, MoPoEs as special cases. However, the exact posterior is proportional to the prior and the (permutation-invariant) product of the likelihoods for each modality. For exponential-family models, this implies that if one uses certain initial encoding functions (based on generative parameters), the optimal natural parameters do solve a non-trivial optimisation problem wherein the aggregation can be represented as a sum, see Appendix G.
>
> Arguably, one could also consider efficient architectures that are not permutation-invariant in the modality dimension (e.g. multi-modal transformer architectures, such as [2]) operating directly on the modalities to build an encoding distribution on a shared latent space. However, we believe using such architectures makes the comparison with previous aggregation schemes more difficult because they would not share the same modality-specific initial encoding architecture.
>
> [1] Cohen-Karlik, Edo, Avichai Ben David, and Amir Globerson. "Regularizing towards permutation invariance in recurrent models." Advances in Neural Information Processing Systems33 (2020): 18364-18374.
>
> [2] Jaegle, Andrew, et al. "Perceiver IO: A General Architecture for Structured Inputs and Outputs." International Conference on Learning Representations. 2021.

---

### Official Review · Reviewer_EVr5 · 2023-11-10

**Soundness:** 3 good
**Presentation:** 2 fair
**Contribution:** 3 good
**Rating:** 6
**Confidence:** 3

**Summary:**

The authors propose a varational bound for multi-modal variational autoencoders. The bound is, to my knowledge novel, and is an extension of previous work on variational bounds defined over subsets of the latent variables. The proposed bound extends these concepts to an arbitrary number of modalities, by defining the bound over all possible subsets of the data distribution (1. Equation after Eq. 4). The paper contains an extensive literature review of related work (with 8 pages of references). Experiments are performed on synthetic data and on rather "small scale" datasets MNIST and SVHN (Street View House Numbers).

**Strengths:**

The biggest strength of this work is the extensive literature review. The paper serves as a good reference for related work on variational bounds for different distributions, especially for multi-modal latent distributions and also discusses related aspects, for example identifiablity.
The proposed bound is derived from previous work which is clearly presented as its basis.

**Weaknesses:**

Writing style: The paper introduces a lot of related work before discussing the actual contributions. It would be beneficial for the camera ready version if the authors could revise the manuscript and dedicate more room to the actual contributions of the paper. Also the conclusion and discussion part is a little to short. The experimental evaluation is perhaps below the current standard (see below).

Experimental evaluation:
- 5.1 Linear Multi-Modal VAES: The bounds presented in Table 1 do not show a big difference between each other. However, this experiment is on synthetic data and perhaps the experimental setup could be improved to show more significant differences between the different approaches (See also the questions below).

5.3. MNIST and SVHN are rather small datasets, and nowadays usually we would expect a more challenging evaluation on larger datasets.

Technical remarks: Please number all the equations (at least for review)

**Questions:**

- 5.1 Linear Multi-Modal VAES: The bounds presented in Table 1 do not show a big difference between each other. However, this experiment is on synthetic data. Could the dataset be modified, e.g. a different number of mixture components, larger (or smaller) variance of the mixture components, such that the differences become more significant?

5.3. MNIST and SVHN are rather small datasets, and nowadays usually we would expect a more challenging evaluation on larger datasets. Is there a larger dataset which could also be used for the rate-distortion analysis?

**Details Of Ethics Concerns:**

No ethical concerns.

---

> ### Author Response · Authors · 2023-11-22
> **Response to review**
>
> We thank the reviewer for the detailed review and helpful suggestions for enhancing the submission in terms of presentation and additional experimental analyses. We will aim to incorporate these in an updated version.
>
> > Can the linear VAE setup be adjusted to show more significant differences between the different approaches?
>
> Thank you for this suggestion. Indeed, one can consider different linear settings that lead to more pronounced differences for linear multi-modal VAEs between different approaches. In particular, we considered a bi-modal setup with a less noisy and more noisy modality. Concretely, for a latent variable $Z=(Z_0,Z_1,Z_2) \in \mathbb{R}^3$, we set $X_1=Z_0+Z_1+U_1$ and $X_1=Z_0+10Z_2+U_2$ where $Z_i$ and $U_j$ are iid $\mathcal{N}(0,1)$. The second modality is thus more noisy than the first. Different aggregation schemes and variational objectives lead to the log-likelihoods below (relative to the true LLH over five repetitions).
>
>
> |                | relative LLH gap    |                    | full reconstruction error |                         | full rates   |            | cross reconstruction error |                         | cross rates   |           |
> |----------------|---------------------|--------------------|---------------------------|-------------------------|--------------|------------|----------------------------|-------------------------|---------------|-----------|
> |                |  ours               |  mixture           |  ours                     |  mixture                |  ours        |  mixture   |  ours                      |  mixture                |  ours         |  mixture  |
> | PoE            |  1.29               |  7.11              |  -2.3E+35                |  -2.2E+35              |  2.1E+35    | 2.0E+35   |  -2.4E+34                 |  -1.9E+34              |  1.4E+35     |  1.7E+35 |
> | MoE            |  0.11               |  0.60              |  -32.07                   |  -30.09                 |  1.02        |  2.84      |  -33.27                    |  -28.52                 |  2.37         |  19.33    |
> | SumPooling     |  3.6E-05           |  0.06              |  -2.84                    |  -3.23                  |  2.88        |  2.82      |  -52.58                    |  -27.26                 |  1.42         |  27.35    |
> | SelfAttention  |  3.4E-05           |  0.06              |  -2.85                    |  -3.23                  |  2.87        |  2.82      |  -52.59                    |  -27.25                 |  1.42         |  27.41    |
>
> In contrast to the previously reported results with more homogeneous modalities, the differences appear more significant. Our variational bound leads to higher LLH for all aggregation schemes. Learnable aggregation schemes yield higher LLH than PoEs or MoEs.
>
> We have also considered a rate-distortion analysis that leads to more significant differences compared with the setting from the initial submission and with overall results appearing in agreement with the MNIST-SVHN-Text experiment. Recall that full reconstruction log-likelihood is the negative full distortion  $-D_\mathcal{M}=\int p_d(x)q_{\phi}(z_{\mathcal{M}}|x) \log p_{\theta}(x|z_{\mathcal{M}})  dz_{\mathcal{M}}  d x$, while the full rate is $R_\mathcal{M} =\int p_d(x)  KL(q_{\phi}(z_{\mathcal{M}}|x)|p_{\theta}(z)) d x$. Similarly, the cross reconstruction log-likelihood is $-D^c_{\setminus \mathcal{S}}=\int p_d(x_\mathcal{S}) q_{\phi}(z|x_{\mathcal{S}}) \log p_{\theta}(x_{\setminus \mathcal{S}}|z_{\mathcal{S}}) d z_{\mathcal{S}} d x_\mathcal{S}$ and the cross rate is  $R_{\setminus \mathcal{S}} =\int p_d(x) KL(q_{\phi}(z|x)|q_{\phi}(z|x_\mathcal{S}))  d x$. Note that mixture-based bounds optimize directly for the cross-reconstruction log-likelihood, see Remark 4, and do not contain a cross-rate term as a regulariser, in contrast to our bound (Lemma 2, Corollary 3).
>
> For the same $\beta=1$, our variational bound leads to higher reconstruction for/given all modalities with increased full rates. Mixture-based bound results in higher cross reconstructions with increased cross rates.
> More flexible aggregation schemes increase the full and cross reconstruction for any given bound, while not necessarily increasing the full or cross rates, i.e., they can result in an improved point within a rate-distortion curve for some configurations.
>
>
>
> > Is there a larger dataset which could also be used for the rate-distortion analysis?
>
> This is a very good question. We will definitely consider the application to a larger dataset (such as PolyMNIST or CUB) and perform a multi-modal rate-distortion analysis there.

---

### Meta-Review · Area_Chair_FpsU · 2023-12-06

**Metareview:**

The paper considers variational autoencoders for multi-modal data. In particular, a new variational lower-bound using flexible aggregation schemes across modalities generalizing some existing works is proposed. In experiments, improvements of the proposed approach, mainly over special-case aggregation schemes, is demonstrated and some aspects regarding learning identifiable models are analyzed.

The paper received mixed recommendations from the reviewers. Reviewers were for instance concerned with the experimental comparison to other state-of-the-art methods and the demonstration of the approach's performance on real-world data. Also, the initially stated contributions were criticized and adjusted by the authors during the rebuttal phase. However, reviewers appreciated, for instance, the importance of the addressed problem, the extensive literature review, and the studies regarding identifiable models.

While some of the presented ideas and parts of the proposed approach are valuable, the paper currently does not meet the bar for acceptance because of its insufficient evaluation against state-of-the-art competitors. Results in this regard added by the authors during the rebuttal phase are incomplete and can partly not be directly compared to state-of-the-art methods in a fair way. The paper must be improved in this regard to clearly demonstrate the advantages of the proposed approach. This requires a substantial update to the conducted experiments. Thus I am recommending rejection of the paper but encourage the authors to improve their work in line with the reviewers' comments for possible future submissions.

**Justification For Why Not Higher Score:**

The paper needs additional experiments and clarification of contributions before it should be accepted.

**Justification For Why Not Lower Score:**

N/A

---

### Decision · Program_Chairs · 2024-01-16

Reject